# INTERPRETABLE 3D NEURAL OBJECT VOLUMES FOR ROBUST CONCEPTUAL REASONING

**Nhi Pham**[1], **Artur Jesslen**[2], **Bernt Schiele**[1], **Adam Kortylewski**[3,*], **Jonas Fischer**[1,*]

[1]Max Planck Institute for Informatics, Saarland Informatics Campus, Germany
[2]University of Freiburg, Germany
[3]CISPA Helmholtz Center for Information Security, Germany
[*]Equal senior advisory

**(a)** Explicit 3D-Aware Concept Representation · **(b)** Robustness vs. Interpretability Trade-off

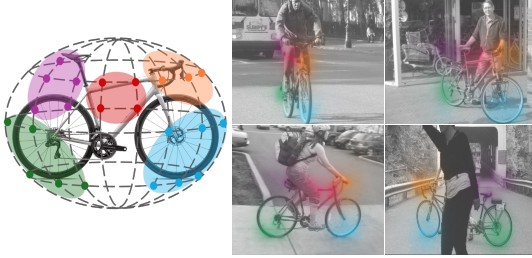
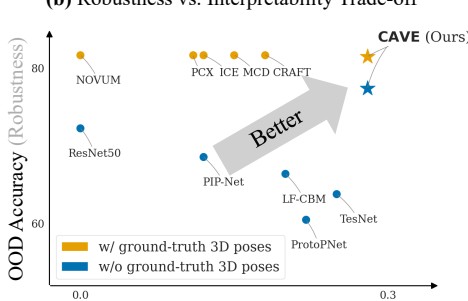

Figure 1: **CAVE - Concept Aware Volumes for Explanations**. **(a)** We learn 3D object volumes (left), here **ellipsoids**, with concept representations. Each concept captures distinct local features of objects (color coded). At inference (right), these concepts are matched with 2D image features, achieving robust and interpretable image classification. **(b) CAVE achieves the best robustness vs. interpretability tradeoff** across methods (higher is better on both axes). Here, we measure robustness with OOD accuracy (%) on Occluded Pascal3D+ (Wang et al., 2020), and interpretability with concept spatial localisation (i.e., how well an explanation generated by a method overlaps with ground-truth human-annotated object parts in Pascal-Part (Chen et al., 2014); defined in App. G.2).

## ABSTRACT

With the rise of deep neural networks, especially in safety-critical applications, *robustness and interpretability* are crucial to ensure their trustworthiness. Recent advances in 3D-aware classifiers that map image features to volumetric representation of objects, rather than relying solely on 2D appearance, have greatly improved robustness on out-of-distribution (OOD) data. Such classifiers have not yet been studied from the perspective of interpretability. Meanwhile, current concept-based XAI methods often neglect OOD robustness. We aim to address both aspects with **CAVE** – **C**oncept **A**ware **V**olumes for **E**xplanations – a new direction that unifies interpretability and robustness in image classification. We design CAVE as a robust and inherently interpretable classifier that learns sparse concepts from 3D object representation. We further propose *3D Consistency (3D-C)*, a metric to measure spatial consistency of concepts. Unlike existing metrics that rely on human-annotated parts on images, 3D-C leverages ground-truth object meshes as a common surface to project and compare explanations across concept-based methods. CAVE achieves competitive classification performance while discovering consistent and meaningful concepts across images in various OOD settings. Code is available at github.com/phamleyennhi/CAVE.

## 1 INTRODUCTION

Deep neural networks (DNNs) have achieved impressive performance in diverse domains ranging from healthcare to autonomous driving. However, their decision-making processes remain largely opaque and often rely on spurious correlations (Noohdani et al., 2024). In high-stake applications,

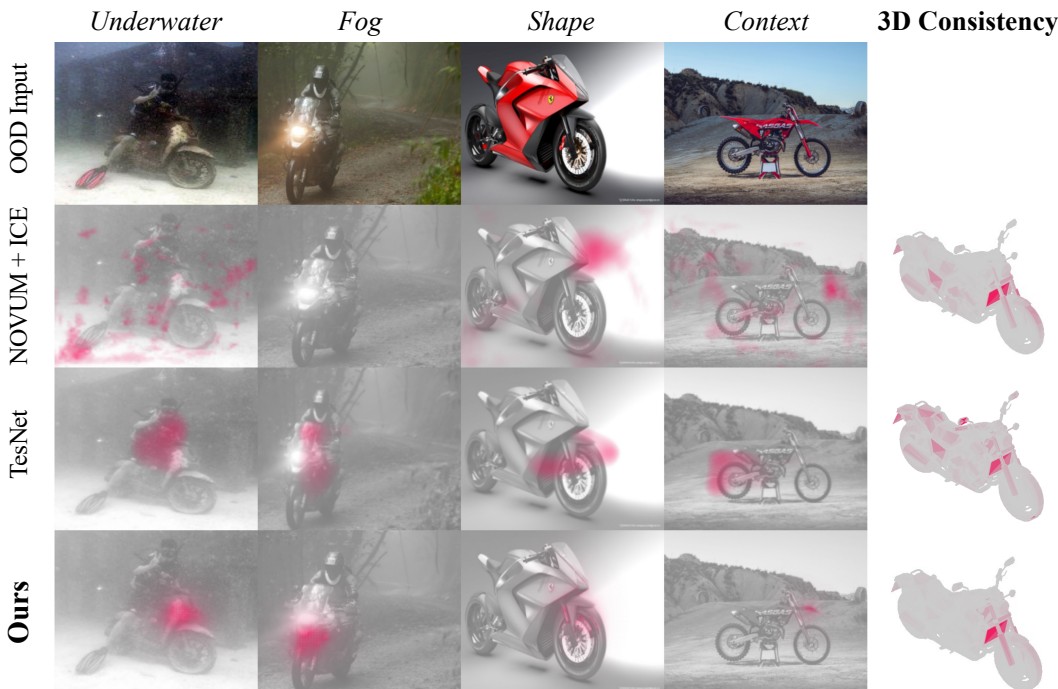

Figure 2: **CAVE (Ours) discovers consistent concept for Motorbike images under challenging OOD nuisances in OOD-CV dataset.** Columns correspond to inputs with different OOD nuisances: *underwater*, *fog*, *shape*, and *context*. Rows show attributions from NOVUM + ICE (best post-hoc), TesNet (best ad-hoc) and Ours. CAD mesh (right) visualises the **class-level 3D consistency** of a concept, where highlighted regions visualise the aggregated concept attributions across test images. **CAVE** produces more consistent and localised explanations, NOVUM + ICE and TesNet detect concepts inconsistently under nuisances. See App. L1 for full qualitative comparison.

for instance, in the medical domain, autonomous systems or judicial justice, ensuring both *interpretability and robustness* is not just desirable – it is essential for safety and trustworthiness.

To overcome such issues and make networks more transparent and interpretable, various approaches have been proposed in the scope of explainable AI (XAI). Notably, post-hoc methods generate concept-based explanations for pre-trained networks, providing insights into their decision-making process without altering the underlying architecture (Fel et al., 2023b; Ancona et al., 2018; Erion et al., 2021; Hesse et al., 2021; Fel et al., 2023a). However, such methods only approximate the model's computations, and thus do not provide a faithful explanation. In contrast, another line of research enforces interpretability directly during training, making aspects of the model inherently interpretable and ensuring that the explanations remain faithful to its computations (Chen et al., 2019; Nauta et al., 2023a; Alvarez Melis & Jaakkola, 2018; Nauta et al., 2021; Oikarinen et al., 2023). Yet, these models are often not designed with robustness in mind (cf. Fig. 1b).

DNNs in real-world scenarios typically encounter distribution shifts over time such as occlusions, or adverse weather conditions in the case of autonomous driving. If the model is not OOD-robust, any explanations extracted from the model representation are unreliable. In fact, under foggy weather or changing contexts, most methods often fail to identify consistent and meaningful explanations (cf. Fig. 2). In an orthogonal line of research, incorporating 3D compositional object representations into the training pipeline significantly improves OOD robustness, yet these classifiers remain inherently opaque, leaving a critical gap in understanding their decision making (Jesslen et al., 2024). They are also restricted to training settings where ground-truth 3D poses are available.

This landscape exposes a key gap in the interpretability and robustness of image classification. We address it with **CAVE** — **C**oncept **A**ware **V**olumes for **E**xplanations — a framework for image classification that is both OOD-robust and inherently interpretable. CAVE builds on the idea of representing each class with a neural object volume (NOV) introduced in NOVUM (Jesslen et al., 2024), where simple shapes such as cuboids or spheres are densely distributed with Gaussian features on the surface. These Gaussian features are then aligned with the latent image features for classification. While this improves OOD robustness, these dense features remain opaque and offer

little semantic insights. CAVE overcomes this by representing objects with ellipsoid NOVs, from which a **sparse dictionary of high-level concepts** is learned (cf. Fig. 1a). Additionally, we leverage zero-shot estimated object orientation from Wang et al. (2025b), thereby alleviating the reliance on pose annotations during training in 3D-aware architectures such as NOVUM. Once concepts are learned, they can be attributed to pixel spaces for explanations. Standard attribution methods such as layer-wise relevance propagation (LRP) however unfaithfully leak relevances in 3D-aware architectures with non-standard layers. We modify LRP to account for volumetric representations such as NOVs in 3D-aware architectures, while also ensuring its relevance conservation property.

For concept evaluation, existing consistency metrics often assume that learned concepts are aligned with human-annotated object parts (Huang et al., 2023; Behzadi-Khormouji & Oramas, 2023), even though good model performance does not require such alignment. We thus propose **3D consistency (3D-C)**, which uses ground-truth 3D object meshes as a common surface to project and compare concepts, allowing consistency to be measured without relying on part annotations.

In summary, our main **contributions** include:

(i) CAVE as a robust and inherently-interpretable image classifier through ellipsoid NOVs. Our concept basis is spatially-aware, and its explanations are model-faithful,

(ii) an adaptation of LRP for concept attribution in classifiers with volumetric representations,

(iii) and a novel part-annotation-free consistency metric 3D-C that captures the spatial coherence of concepts across viewpoints and OOD nuisances.

In comparison to various XAI methods, both post-hoc and inherently interpretable approaches, CAVE shows a favorable balance of OOD robustness and interpretability, with improvements across metrics such as OOD accuracy, object coverage, spatial localisation, and concept consistency.

## 2 RELATED WORK

**Leveraging 3D supervision.** 3D information is useful for 2D feature representations in downstream tasks like segmentation and depth estimation, but these require rich multi-view data (Yue et al., 2024; Hou et al., 2021; Fu et al., 2024). Recently, NOVUM pioneers using 3D information for robust classification, by considering 3D pose information to fit cuboid NOVs to an image (Jesslen et al., 2024). This line of work forms the basis of our approach.

**Learning without 3D supervision.** In 3D-aware image classifiers such as NOVUM, model training requires ground-truth 3D pose annotations to align NOVs with the object in the image. This requirement significantly limits applicability, as such annotations are expensive to obtain and often unavailable in real-world datasets. Recent work proposes zero-shot object orientation estimation models, e.g., Orient-Anything (Wang et al., 2025b), which extract pose information given an input image. CAVE adopts such pose estimators to remove the need for annotated 3D poses in training.

**Concept-based explanations.** A major line of work in XAI focuses on discovering *concept representations*. Post-hoc concept extraction methods such as CRAFT (Fel et al., 2023b; 2024) and ICE (Zhang et al., 2021) factorise model activations to uncover latent concepts, while MCD (Vielhaben et al., 2023) uses sparse subspace clustering to identify concept subspaces, and PCX (Dreyer et al., 2024) learns concepts from relevance maps. These approaches offer *implicit interpretability*, and only approximate its computation (i.e., not model-faithful). A different class of approaches makes the model predictions themselves *explicitly interpretable* by design. Concept Bottleneck Models (CBMs) (Koh et al., 2020; Oikarinen et al., 2023) introduce a dedicated concept layer whose units correspond to human-understandable concepts, thus providing *explicit semantic interpretability*. Similarly, prototype-based networks such as ProtoPNet (Chen et al., 2019) and its follow-up works (Wang et al., 2021; Nauta et al., 2023a; Wang et al., 2025a) learn prototypical image features whose presence is used for prediction, offering *explicit visual interpretability* through prototype projection. CAVE is *faithful-by-design* and *implicitly interpretable*: its internal units arise through unsupervised discovery rather than explicit semantic or prototype supervision. We therefore compare CAVE with both post-hoc concept discovery methods (CRAFT, ICE, MCD, PCX) and faithful-by-design interpretable models (LF-CBM, ProtoPNet, TesNet, PIP-Net, MGProto). This allows us to assess concept quality across implicit versus explicit approaches, while comparing predictive performance among faithful-by-design models.

**Evaluating concept explanations.** Prior works assess concepts along several axes: (i) *spatial localisation* to ground-truth bounding boxes or masks, (ii) *object coverage*, i.e. how well they attend to different parts of the object, and (iii) *consistency* across instances (Huang et al., 2023; Behzadi-Khormouji & Oramas, 2023; Huang et al., 2024; Zhu et al., 2025). Such metrics are limited for two

reasons: they require human-annotated object parts, and model training often optimises for task performance rather than part alignment. To complement these metrics, we propose *3D-C*, a consistency measure of concept across samples without requiring part annotations.

## 3 PRELIMINARIES: NEURAL OBJECT VOLUMES (NOVS) AND NOVUM

In this section, we provide a recap of neural object volumes (NOVs) and NOVUM (Jesslen et al., 2024), as they are essential for defining our method CAVE in Section 4.

**Notations**. In a supervised setting, an image classifier consists of a feature extractor $\mathcal{E}(\cdot)$ and a classification layer $\phi(\cdot)$. Given an input image $x \in \mathbb{R}^{H \times W \times C}$, the feature extractor produces a feature map $F_x = \mathcal{E}(x) \in \mathbb{R}^{H' \times W' \times C'}$. Here, $H' \leq H$ and $W' \leq W$ denote the spatial dimension of the latent representation of $x$ and need not correspond to pixel resolution. $C'$ denotes the number of latent channels. We use $f_i = \mathcal{E}_i(x)$ to indicate the L2-normalised latent feature vector for the $i$-th pixel of feature map $F_x$ in raster order. The classification decision is computed as $y = \phi(F_x)$.

**Neural Object Volumes (NOVs)**. A NOV is a volumetric approximation of an object class $y$ (cf. Fig. 3), and consists of a set of $K$ 3D Gaussians. In NOVUM, the NOVs are instantiated as cuboids with Gaussians evenly distributed on their surface. The $k$-th Gaussian is defined by its center $\mu_y^{(k)} \in \mathbb{R}^3$ with fixed unit variance, and is associated with an L2-normalised feature vector $g_y^{(k)} \in \mathbb{R}^{C'}$. Hence, each Gaussian in a NOV is assigned a fixed 3D coordinate on the surface of a canonical object shape and form a structured volumetric representation. We define the matrix of Gaussian features for the object class $y$ as $\mathcal{G}_y \in \mathbb{R}^{K \times C'}$, which have the same channel dimension $C'$ as latent features $F_x$ from the backbone model and will be later used to match with the $F_x$. Extending this notation, the complete matrix of Gaussian features across all $N$ object classes can be represented as $\mathcal{G} = [\mathcal{G}_1; \mathcal{G}_2; \ldots; \mathcal{G}_N] \in \mathbb{R}^{NK \times C'}$. During training of NOVUM, these NOVs $\mathcal{G}$ are learned to align with latent image features $F_x$, orienting the volumes through ground-truth 3D pose annotations (see also Appendix F for training objectives). Intuitively, matching image latent features to Gaussian features guided by the pose annotations aligns the image representation with the canonical 3D object geometry, allowing Gaussians to correspond to consistent object geometry.

**Classification with NOVs in NOVUM** is done through **feature matching** between the backbone image features $F_x$ and the set of learned 3D object representation NOVs $\mathcal{G}$. This operation aligns each feature $f_i \in F_x$ with the most similar Gaussian feature across $\mathcal{G}$. The logit for class $y$ is computed by summing over all spatial locations where feature $f_i$ is matched to a Gaussian feature of $y$

$$s_y = \phi(F_x, \mathcal{G}_y) = \sum_i \max_k f_i \cdot g_y^{(k)} \quad (1)$$

The class with the highest score $s_y$ is the predicted label $y^*$. This formulation gives rise to 3D-aware classification through a bag-of-words feature matching mechanism, where image features are directly compared against 3D-aware Gaussian features. However, this classification process remains inherently opaque. The number of Gaussian features involved in the matching step in the order of thousands makes it difficult to interpret which features contribute to the final decision (cf. Fig. 3, left). In the next Section 4, we describe our method CAVE, which replace dense Gaussian features and instead operates on a sparse dictionary of representative Gaussian features. We refer to these representations as **concept-based NOVs**, an interpretable concept basis for 3D-aware classification (cf. Fig. 3, right).

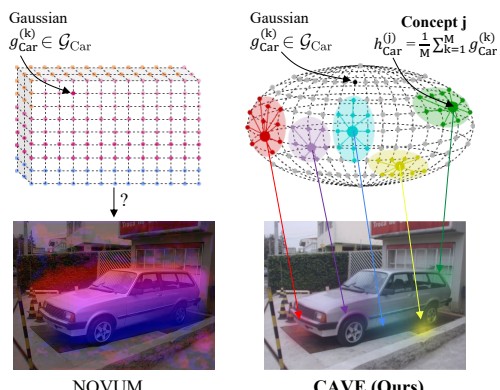

Figure 3: **CAVE** adopts ellipsoid NOVs and produces a sparse set of concepts that replace the dense thousands of Gaussians in NOVUM, thus providing more interpretable explanations.

## 4 CAVE: CONCEPT-AWARE VOLUMES FOR EXPLANATIONS

Our goal is to build an image classifier with two key properties: (i) robust classification in OOD settings, and (ii) inherently interpretable model predictions. While specific solutions exist for each

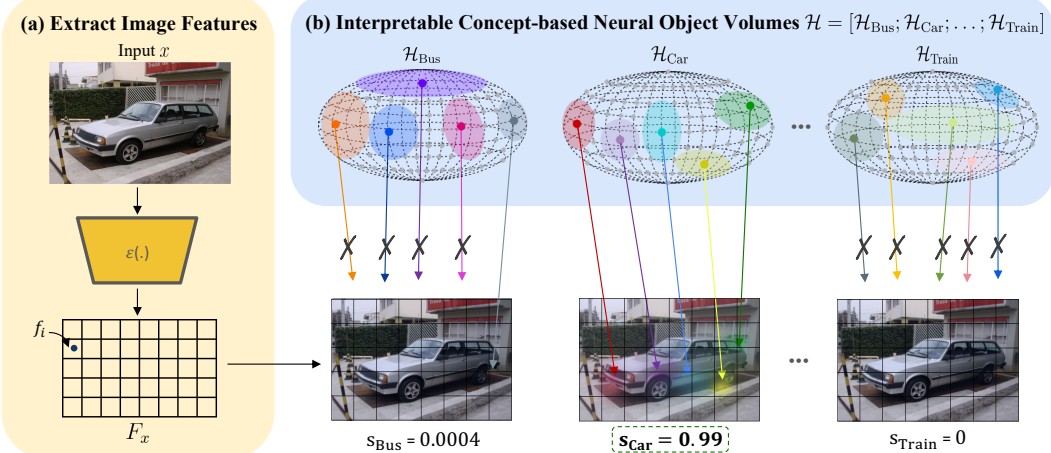

**(a) Extract Image Features**

Input $x$

$\varepsilon(.)$

$f_i$

$F_x$

**(b) Interpretable Concept-based Neural Object Volumes** $\mathcal{H} = [\mathcal{H}_{\text{Bus}}; \mathcal{H}_{\text{Car}}; \dots ; \mathcal{H}_{\text{Train}}]$

$\mathcal{H}_{\text{Bus}}$     $\mathcal{H}_{\text{Car}}$     $\mathcal{H}_{\text{Train}}$

$s_{\text{Bus}} = 0.0004$     $s_{\text{Car}} = 0.99$     $s_{\text{Train}} = 0$

**(c) Classification with Concept Matching Eq. (2) for Each Class**

Figure 4: **CAVE – Concept Aware Volumes for Explanations**, a framework for robust conceptual reasoning and classification through 3D-aware **concept-based neural object volumes (NOVs)**. In this visual illustration, colors indicate the top-5 concepts within each class. For classification, CAVE combines **(a)** extracted image features $F_x$ and **(b)** interpretable concept-aware NOVs $\mathcal{H}$ through a bag-of-word concept matching **(c)** with equation 2, where each feature $f_i \in F_x$ is best aligned with $\mathcal{H}$ by cosine similarity. Correct classification happens when many image features activates Car concepts, while concepts in other classes fail to align with any feature (crossed-out arrows).

property individually, combining them remains far from trivial. Building upon NOVUM, our method leverages volumetric object representations to simultaneously achieve both robustness and interpretability. In Section 4.1, we show how to extract a sparse set of interpretable concepts from dense Gaussian features on NOVs, which then form our concept-based NOVs for inherently interpretable classification. Figure 4 gives an overview of our method CAVE. We further attribute these concepts from the model prediction, through these concept-based NOVs, to the input image for explanations using our modified LRP. Then, in Section 4.2, we discuss how to improve learning NOVs through more expressive shapes and introducing weak 3D supervision with estimated poses for CAVE, thus extending its applicability to settings without ground-truth 3D pose annotations.

## 4.1 Identifying and Attributing Concepts through NOVs

**From NOVs to Concept-Based NOVs**. To achieve an inherently interpretable NOV-based classifier, we identify a meaningful concept basis from each NOV and replace the latter with these concepts (cf. Fig. 4b). Formally, for a NOV $\mathcal{G}_y \in \mathbb{R}^{K \times C'}$ of class $y$, we formulate our class-wise concept extraction problem through the lens of dictionary learning (Mairal et al., 2014; Fel et al., 2024):

$$(\mathcal{W}_y^\star, \mathcal{H}_y^\star) = \arg \min_{\mathcal{W}_y, \mathcal{H}_y} \| \mathcal{G}_y - \mathcal{W}_y \mathcal{H}_y^\top \|_F^2$$

where the weight matrix $\mathcal{W}_y^* \in \mathbb{R}^{K \times D}$ and the dictionary of $D$ concept vectors $\mathcal{H}_y^* = [h_y^{(1)}, \dots, h_y^{(D)}]^T \in \mathbb{R}^{D \times C'}$ minimize the element-wise distance between our Gaussian features $\mathcal{G}_y$ and $\mathcal{W}_y \mathcal{H}_y^T$. In the case of hard clustering, the weight matrix $\mathcal{W}_y^*$ reduces to a discrete assignment matrix, where each row is a one-hot encoding that corresponds to only one concept. This allows clustering to be much more interpretable than methods with less sparse weight matrices. We adopted K-Means clustering for its balance of accuracy, concept sparsity, and alignment to the learned NOVs. We refer to our ablation on concept extraction methods in Appendix J.

The extracted concept dictionary $\mathcal{H}_y^*$ is now seen as a *sparse and interpretable* **concept-based NOV** to replace the original dense NOV $\mathcal{G}_y$ (cf. Fig. 3). We modulate the original feature matching $\phi(F_x, \mathcal{G})$ in NOVUM with **concept matching** $\phi(F_x, \mathcal{H})$ that establishes correspondences between $F_x$ and new volumetric representation $\mathcal{H} = [\mathcal{H}_1^*; \mathcal{H}_2^*; \dots ; \mathcal{H}_N^*] \in \mathbb{R}^{ND \times C'}$. Eq. (1) thus becomes

$$s_y = \phi(F_x, \mathcal{H}_y) = \sum_i \max_{j \leq D} f_i \cdot h_y^{(j)} \qquad (2)$$

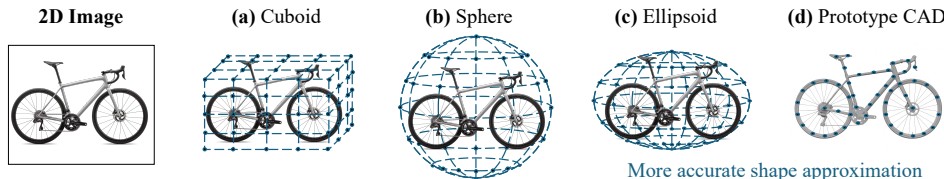

Figure 5: **NOV shapes for approximating object class Bicycle**. Here Gaussians are evenly distributed on the surface of each volumetric type.

This reformulation, illustrated in Fig. 4b-c, enables feature matching against a compact and interpretable concept set instead of thousands of Gaussians, yielding sparser representations, stronger robustness, and more confident predictions compared to NOVUM (cf. Fig. 7). Both $f_i$ and $h_y^{(j)}$ are L2-normalised. Thus each dot product $f_i \cdot h_y^{(j)}$ in Eq. (2) is a cosine similarity in $[-1, 1]$ which defines latent feature–concept alignment. Importantly, the score in Eq. (2) is computed exactly from activations of these volumetric concepts, preserving the faithfulness that NOVUM (Jesslen et al., 2024) also has while adding interpretability through sparse concept representations. These concepts emerge implicitly as geometrically-grounded units through unsupervised clustering of Gaussian features, while prototype-based methods and CBMs learn concepts explicitly.

**Attributing concepts with NOV-aware LRP.** We aim to provide interpretable explanations on the input-level for our NOV-based concepts $\mathcal{H}$, thereby demonstrating the model's reasoning through neural volumetric concepts. To achieve this, we build on LRP, a well-established attribution method that traces relevances from the model's prediction back to the input pixels (Bach et al., 2015; Otsuki et al., 2024). A key principle of LRP is the conservation property, which requires the total relevance to remain constant throughout the network (Otsuki et al., 2024). However, we empirically find that when directly applied to NOV-based architectures, LRP fails to uphold this property and instead unfaithfully leaks relevances (cf. Appendix Fig. I1). We address this by introducing a redistribution rule that preserves the conservation property through the concept-matching operator $\phi(F_x, \mathcal{H})$, ensuring that the total relevance assigned to input pixels equals that at the concept level $\sum_{f_i \in F_x} R_{f_i} = \sum_{h \in \mathcal{H}} R_\phi(h) = R_{y^*}$. This NOV-aware extension allows us to correctly attribute predictions through volumetric concepts with LRP, enabling robust and reliable concept explanations even under challenging OOD conditions. Full derivation is provided in Appendix C.

### 4.2 EXTENDING NOVs: WEAK 3D SUPERVISION AND MORE EXPRESSIVE SHAPES

**Learning with weak 3D supervision**. One notable limitation of NOV-based classifiers is that they assume access to ground-truth 3D pose annotations during training (Jesslen et al., 2024). Here, we relax this requirement by training CAVE with estimated object orientations from Orient-Anything (Wang et al., 2025b). While the weaker supervision introduces some performance drop, especially under OOD nuisances (cf. Appendix E2), it shows that NOV-based classifiers can operate without explicit pose annotations, allowing for better scalability. Unless stated otherwise, we use CAVE with estimated poses for a fair comparison to ad-hoc baselines in our setting.

**More accurate shape approximation.** Typically, NOV-based classifiers such as NOVUM use simple shapes such as cuboids and spheres, which provide a coarse volumetric approximation of objects. We broaden the scope by adapting NOVs to more expressive geometries: ellipsoids and prototype CADs (cf. Fig. 5), which serve as basis for our concept extraction. We adopt **ellipsoid NOVs** in our setup, given their favorable trade-off between OOD accuracy and interpretability (cf. Appendix H).

## 5 3D CONSISTENCY OF CONCEPTS

Evaluating concept consistency is challenging. Prior works often rely on part annotations (Huang et al., 2023; Behzadi-Khormouji & Oramas, 2023; Huang et al., 2024), which do not necessarily reflect what models actually learn, since training optimises task accuracy rather than alignment with pre-defined parts. Object geometry, however, provides a natural reference: if a concept is meaningful, it should consistently map to the same semantic region of the object under different poses or OOD factors. We call this property **3D consistency** (3D-C) (cf. Fig. 6, in-distribution). For instance, the concept "front part of a motorbike" is consistent if its attributions refer to the same part under distribution shifts such as weather, shape, or context (cf. Fig. 2, OOD). Our 3D-C thus complements existing metrics with a principled alternative independent of part annotations.

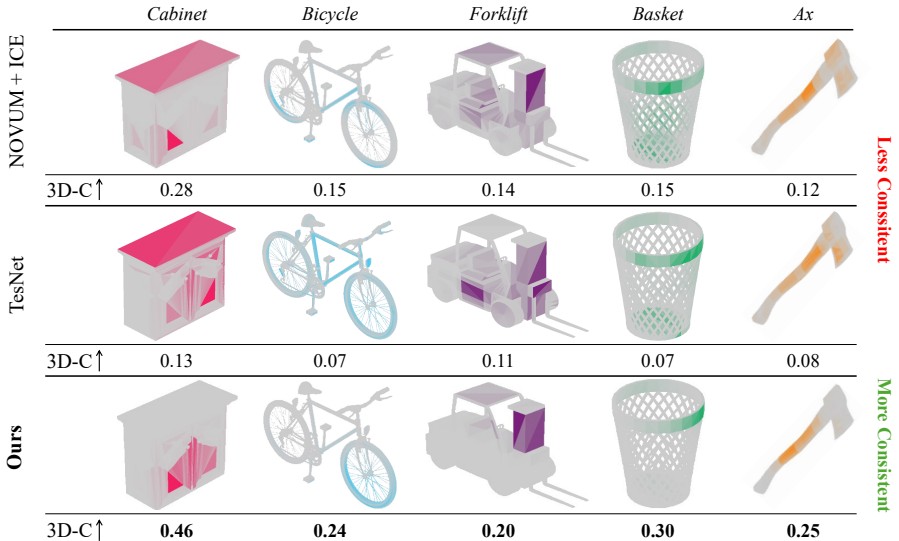

Figure 6: **CAVE (Ours) produces most consistent concepts across different classes**, compared to NOVUM + ICE (best post-hoc) and TesNet (best ad-hoc). We highlight how consistent a concept by aggregating concept relevance scores from all class-wise test images onto object mesh, and report **class-wise 3D-C scores** below each row. Higher 3D-C means more consistent mapping to the same region. See Tab. 1 for full quantitative comparison with baselines.

To compute the 3D-C score of a concept $h$ in class $y$, we project its positive attributions $A^+(x, h) \in \mathbb{R}^{H \times W}$ from test images $x \in \mathcal{X}_y$ onto the CAD model of class $y$. This projection uses ground-truth 3D poses when available, and estimated poses from Orient-Anything (Wang et al., 2025b) otherwise. Formally, we define the mapping $\Omega_y : \mathbb{R}^{H \times W} \to \mathbb{R}^{|\mathcal{Q}_y|}$, where $\mathcal{Q}_y$ denotes the set of triangles in the CAD model of class $y$ that represents the object's surface. Given a concept $h$ and an input image $x$ of class $y$, $\Omega_y$ maps the positive attribution map $A^+(x, h)$, onto the corresponding oriented CAD model and aggregates, for each triangle $q \in \mathcal{Q}_y$, the sum of all projected attributions falling onto it. We denote by $(i, j)$ a pixel position in $A^+(x, h)$ and by $A^+(x_{ij}, h)$ the positive attribution given to this pixel. Concretely, we define the output vector $\Omega_y(A^+(x, h))$ component-wise as:

$$\Omega_y\big(A^+(x, h)\big)_q := \sum_{(i,j) \in \mathcal{P}_y^{(q)}} A^+(x_{ij}, h), \tag{3}$$

where $\mathcal{P}_y^{(q)}$ is the set of pixel positions $(i, j)$ whose projection falls onto triangle $q \in \mathcal{Q}_y$ of the CAD model of class $y$. For each concept $h$, we normalise the concept attribution $A^+(x, h)$ such that $\sum_{(i,j) \in \{1,...,H\} \times \{1,...,W\}} A^+(x_{ij}, h) = 1$. The 3D-C score for concept $h$ across $\mathcal{X}_y$ is defined as:

$$\textbf{3D-C}(\mathcal{X}_y, h) = 1 - \frac{1}{2}\left[ \frac{1}{n_y^2} \sum_{x \neq x' \in \mathcal{X}_y} \big\| \Omega_y(A^+(x, h)) - \Omega_y(A^+(x', h)) \big\|_1 \right] \tag{4}$$

which is normalised to $[0, 1]$, where $n_y$ is the number of test images in $\mathcal{X}_y$ in which concept $h$ is present. We exclude concepts that occur in fewer than $\tau\%$ of test images of class $y$ (here, we choose $\tau = 50$), as they may appear spuriously consistent when evaluated on too few test samples. See Appendix D for further details on visualisation of concept consistency on object meshes.

## 6 EXPERIMENTS

**Datasets and metrics.** We evaluate CAVE with weak 3D supervision on *classification accuracy* and *3D-C* in two settings: (i) *in-distribution* on Pascal3D+ (Xiang et al., 2014) and large-scale ImageNet3D (Ma et al., 2024), and (ii) *OOD* on OccludedP3D+ (Wang et al., 2020) (3 occlusion levels on Pascal3D+) and OOD-CV (Zhao et al., 2022) (nuisances in pose, shape, context, texture, and weather). We further assess *spatial localisation* (weighted IoU with attributions) and *object*

| | Models | Localise. ↑ | Coverage ↑ | 3D Consistency (3D-C.) ↑ | | | |
|---|---|---|---|---|---|---|---|
| | | Pascal-Part | Pascal-Part | Pascal3D+ | ImageNet3D | OccludedP3D+ | OOD-CV |
| Post-hoc | NOVUM + CRAFT (Fel et al., 2023b) | 0.18 | 0.42 | 0.28 | 0.26 | 0.15 | 0.15 |
| | NOVUM + MCD (Vielhaben et al., 2023) | 0.15 | 0.34 | 0.16 | 0.25 | 0.11 | 0.14 |
| | NOVUM + ICE (Zhang et al., 2021) | 0.12 | 0.44 | 0.28 | 0.27 | 0.15 | 0.15 |
| | NOVUM + PCX (Dreyer et al., 2024) | 0.11 | 0.33 | 0.10 | 0.21 | 0.08 | 0.11 |
| Ad-hoc | LF-CBM (Oikarinen et al., 2023) | 0.20 | 0.56 | 0.15 | 0.14 | 0.13 | 0.11 |
| | ProtoPNet (Chen et al., 2019) | 0.22 | 0.43 | 0.19 | 0.13 | 0.21 | 0.09 |
| | TesNet (Wang et al., 2021) | 0.25 | 0.44 | 0.20 | 0.18 | 0.18 | 0.12 |
| | PIP-Net (Nauta et al., 2023a) | 0.12 | 0.13 | 0.09 | 0.09 | 0.07 | 0.00 |
| | MGProto (Wang et al., 2025a) | 0.25 | 0.35 | 0.19 | 0.16 | 0.16 | 0.07 |
| | CAVE (Ours) | **0.28** (± 0.001) | **0.80** (± 0.002) | **0.40** (± 0.001) | **0.40** (± 0.001) | **0.23** (± 0.006) | **0.24** (± 0.002) |
| | CAVE (with full 3D supervision) | 0.28 (± 0.001) | 0.87 (± 0.002) | 0.42 (± 0.001) | 0.43 (± 0.0003) | 0.23 (± 0.010) | 0.26 (± 0.001) |

Table 1: **Concept interpretability evaluation** using *spatial localisation* (whether concepts align with human-annotated parts), *object coverage* (extent of concept coverage over the object), and *3D consistency* (3D-C) (concept stability across 3D viewpoints, independent of part annotations). CAVE trained with full 3D supervision (ground-truth 3D poses) are shown in gray text. Our CAVE produces concepts that are spatially localised, sufficiently diverse to cover the object, and robustly consistent across in-distribution and OOD settings. We report our results across 10 random seeds.

| Models | W/o Ground-truth 3D Pose | In-distribution | | Out-of-distribution (OOD) | |
|---|---|---|---|---|---|
| | | Pascal3D+ | ImageNet3D | Occluded P3D+ | OOD-CV |
| LF-CBM (Oikarinen et al., 2023) | **Yes** | 98.4 | 83.3 | 66.4 | 73.5 |
| ProtoPNet (Chen et al., 2019) | **Yes** | 97.4 | 74.0 | 60.5 | 71.2 |
| TesNet (Wang et al., 2021) | **Yes** | 97.6 | 77.9 | 63.8 | 70.1 |
| PIP-Net (Nauta et al., 2023a) | **Yes** | 95.7 | 51.0 | 68.6 | 60.0 |
| MGProto (Wang et al., 2025a) | **Yes** | 97.2 | 64.2 | 73.8 | 72.3 |
| CAVE (Ours) | **Yes** | **99.0** (± 0.03) | **84.6** (± 0.02) | **76.8** (± 0.51) | **80.3** (± 0.27) |
| CAVE (with full 3D supervision) | No | 99.4 (± 0.02) | 88.5 (± 0.03) | 81.3 (± 0.30) | 84.0 (± 0.21) |
| NOVUM (with full 3D supervision) | No | 99.5 | 88.3 | 81.7 | 81.3 |

Table 2: **Classification accuracy** (%, ↑) **comparison**. We compare CAVE (Ours) trained with no 3D supervision (using Orient-Anything (Wang et al., 2025b)) against inherently interpretable models across both in-distribution and OOD datasets. **Best** and second best are highlighted. CAVE and NOVUM with full supervision, i.e., ground-truth 3D poses, are shown in gray text. CAVE achieves consistently higher accuracy, especially in OOD settings. CAVE with weak supervision delivers competitive accuracy without ground-truth 3D pose, with only a modest gap to full supervision. We report our results as the mean (±$\sigma$, where $\sigma$ is standard deviation) across 10 random seeds.

*coverage* (concept comprehensiveness) on Pascal-Part (Chen et al., 2014), and *concept faithfulness* to model's predictions (Wang et al., 2024; Böhle et al., 2022; Rudin, 2019; Adebayo et al., 2018). See also Appendix G for dataset and metric details.

**Baselines.** We apply common post-hoc concept-based methods CRAFT (Fel et al., 2023b), MCD (Vielhaben et al., 2023), ICE (Zhang et al., 2021), and PCX (Dreyer et al., 2024) on NOVUM to make it concept-interpretable. We also consider LF-CBM (Oikarinen et al., 2023), and the prototype learning approaches ProtoPNet (Chen et al., 2019), TesNet (Wang et al., 2021), PIP-Net (Nauta et al., 2023a), and MGProto (Wang et al., 2025a), which are all inherently interpretable. We use ResNet-50 backbone for all methods. Post-hoc baselines extract concepts from NOVUM activations. Unless explicitly stated, CAVE is learned with weak 3D supervision. The number of concepts per class is fixed to $D = 20$ across methods. See further implementation details in Appendix G.

## 6.1 CAVE DISCOVERS SPATIALLY CONSISTENT CONCEPTS

Both NOVUM (Jesslen et al., 2024) and CAVE are faithful-by-design, since their predictions decompose exactly over internal units (Gaussian features in NOVUM, region-level concepts in CAVE). CAVE achieves implicit interpretability by learning sparse, structured concept units from NOVUM's fine-grained Gaussian features, effectively grouping them into geometrically-meaningful concepts. We evaluate concepts on: (i) spatial localisation, (ii) object coverage to measure the extent of concept comprehensiveness, and (iii) 3D-C to assess concept consistency across settings (Tab. 1).

On Pascal-Part, CAVE with weak 3D supervision still provides stronger concept localisation and coverage than both ad-hoc baselines and post-hoc methods applied to NOVUM with full 3D supervision. In particular, CAVE discovers diverse concepts that sufficiently cover on average $\sim 80\%$ of the object, whereas the next best method LF-CBM reaches only $\sim 56\%$. We hypothesise that this higher coverage also supports CAVE in identifying concepts, for example, under occlusion.

For 3D-C, post-hoc methods ICE and CRAFT benefit from the 3D supervision in NOVUM and extract more spatially consistent concepts compared to inherently interpretable baselines on in-distribution data. Their consistency, however, remains lower than CAVE's. In OOD scenarios, all methods show a decline in consistency. CAVE maintains the highest scores and thus comparatively shows that its concepts are more stable under distribution shifts. While only roughly approximated through an ellipsoid, CAVE's concepts still consistently map to meaningful and diverse regions on the object mesh, even under OOD shifts in Fig. 2 and complex structures in Fig. 6.

## 6.2 CAVE MAINTAINS COMPETITIVE CLASSIFICATION ACCURACIES

The goal of CAVE is to be **both** robust and interpretable. We measure robustness in terms of OOD accuracy on OccludedP3D+ and OOD-CV, and further report accuracy on in-distribution Pascal3D+ and ImageNet3D (cf. Tab. 2). We compare only to inherently interpretable methods, omitting post-hoc methods as they do not modify the underlying classification of NOVUM.

Across all datasets, CAVE with ground-truth 3D poses achieves performance competitive with NOVUM, even slightly surpassing it on large-scale ImageNet3D ($+0.2\%$) and OOD-CV ($+2.7\%$), while using much sparser representations. With weak supervision (no ground-truth 3D poses), CAVE shows comparatively mild drops in performance on ImageNet3D and OOD-CV relative to ground truth pose supervision. In the following, we report CAVE *without* ground-truth 3D poses. On in-distribution Pascal3D+, all methods perform relatively well. However, when scaling to ImageNet3D, prototypical networks sharply degrade, with even the strongest TesNet almost $8\%$ lower than CAVE (vs. $1.4\%$ gap on the comparably small Pascal3D+). Under occlusion ranging $20 - 80\%$ of the image in OccludedP3D+, CAVE outperforms the competitors by around $10\%$. Similarly, on OOD-CV, CAVE performs best (80.4% acc) with LF-CBM a distant second (73.5% acc) and other methods performing much worse. In summary, CAVE provides a unique combination of inherent interpretability and robustness to OOD data unmatched by existing work.

## 6.3 ABLATIONS

**Consistency across concept count.** We study how the spatial consistency of concepts varies with the number of class-wise concepts $D$. As shown in Tab. 3, our 3D-C scores improve with more concepts under heavy occlusion, but overall remain stable across concept counts across settings.

| Concept count | Pascal3D+ | ImageNet3D | OccludedP3D+ | | | | OOD-CV |
|---|---|---|---|---|---|---|---|
| | in-dist. | in-dist. | $[20 - 40\%]$ occ. | $[40 - 60\%]$ occ. | $[60 - 80\%]$ occ. | avg. | OOD nuisances |
| $D = 5$ | 0.38 ($\pm$ 0.004) | 0.42 ($\pm$ 0.002) | 0.27 ($\pm$ 0.002) | 0.22 ($\pm$ 0.01) | 0.17 ($\pm$ 0.01) | 0.22 ($\pm$ 0.005) | 0.24 ($\pm$ 0.003) |
| $D = 10$ | 0.39 ($\pm$ 0.003) | 0.41 ($\pm$ 0.001) | 0.28 ($\pm$ 0.003) | 0.21 ($\pm$ 0.005) | 0.19 ($\pm$ 0.02) | 0.23 ($\pm$ 0.007) | 0.24 ($\pm$ 0.001) |
| $D = 20$ | 0.40 ($\pm$ 0.001) | 0.40 ($\pm$ 0.001) | 0.29 ($\pm$ 0.002) | 0.21 ($\pm$ 0.002) | 0.20 ($\pm$ 0.02) | 0.23 ($\pm$ 0.006) | 0.24 ($\pm$ 0.002) |
| $D = 40$ | 0.41 ($\pm$ 0.004) | 0.40 ($\pm$ 0.001) | 0.29 ($\pm$ 0.001) | 0.22 ($\pm$ 0.001) | 0.21 ($\pm$ 0.01) | 0.24 ($\pm$ 0.003) | 0.23 ($\pm$ 0.001) |

Table 3: **3D-C across concept count per class $D \in \{5, 10, 20, 40\}$ of our CAVE with weak 3D supervision.** The results are reported as mean ($\pm\sigma$) across 10 random seeds.

**Sparsity–accuracy tradeoff.** We study the effect of varying the number of class-wise concepts $D$ in CAVE (5–90) compared to NOVUM's fixed $\sim$1130 Gaussians per class (cf. Fig. 7a, b). CAVE achieves competitive accuracy with far fewer concepts, with a knee around $D = 20$, yielding $\sim$98% sparser representations that match or even slightly exceed NOVUM's performance, especially under OOD shifts. CAVE also produces more confident predictions with clearer class separation (Fig. 7c).

**NOV-aware LRP.** Our NOV-aware LRP yields spatially coherent attributions, even under OOD conditions such as snow and heavy occlusion, whereas vanilla LRP (Bach et al., 2015) and Grad-CAM (Selvaraju et al., 2017) produce scattered explanations (cf. Fig 8). Empirically, vanilla LRP unfaithfully leaks relevance compared to our formulation (cf. Appendix Fig. 11). Our full ablation in Appendix I shows that our NOV-aware LRP is essential for reliable concept attribution.

## 7 DISCUSSION

CAVE preserves NOVUM's faithfulness while providing *implicit interpretability*. Its geometrically-grounded concepts emerge through unsupervised clustering and are directly used for prediction. While this differs from explicitly interpretable models, e.g., CBMs and prototype networks, which enforce semantic or visual grounding through supervision, all these approaches are faithful-by-design, concept-based models. Thus, we compare CAVE against these baselines for assessing concept quality and predictive performance across explicit and implicit forms of interpretability.

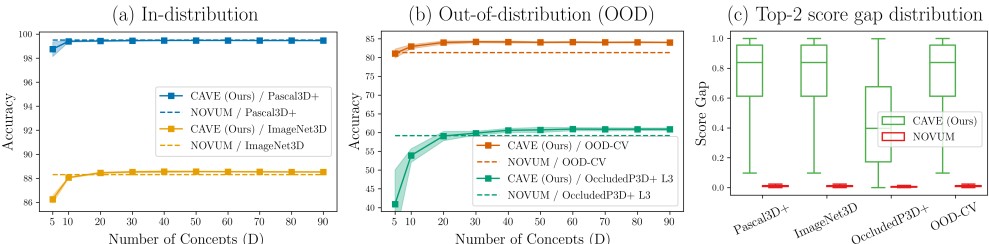

Figure 7: CAVE **replace 1130 dense Gaussians in NOVUM with a compact concept dictionary**, yielding $\sim 98\%$ **sparser representations** that **match or slightly exceed the performance** of NOVUM especially in OOD settings. Both are trained with 3D supervision for a fair comparison. We report mean accuracy in (a), (b) across 10 random seeds, with shaded regions as $\pm 2\sigma$. (c) shows improved model prediction; more confident predictions indicate a clearer class separation, which improves reliability (Hendrycks & Gimpel, 2017) and explanation confidence (Nauta et al., 2023b).

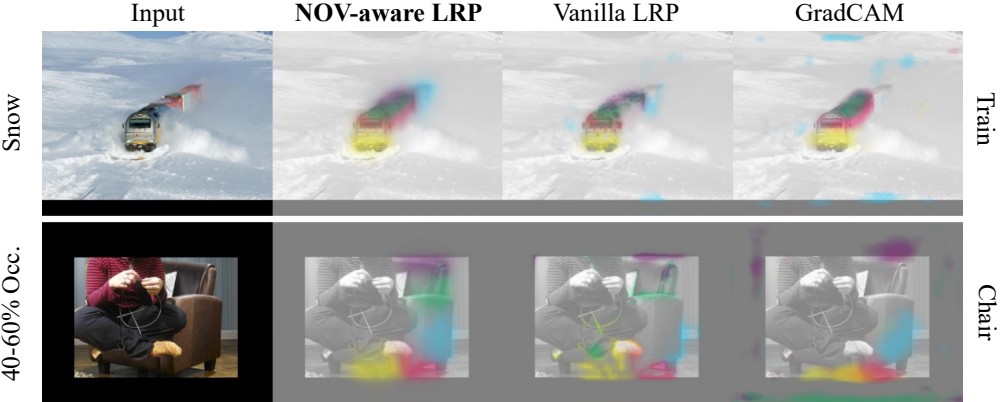

Figure 8: **Our NOV-aware LRP correctly attributes concepts and yields localised explanations, even under different OOD settings**: snow and 40–60% occlusion. Colors indicate the top-5 class-wise concepts per row and are not comparable across rows. See full ablation in Appendix Fig. I2.

Our 3D-C metric requires reference object meshes to assess the spatial consistency of concepts. While this limits evaluation to datasets with reliable CAD models such as Pascal3D+ (Xiang et al., 2014) and ImageNet3D (Ma et al., 2024), advances in large-scale object meshes (Deitke et al., 2023) and mesh generation from text (Siddiqui et al., 2024) or image (Yan et al., 2025) are making high-quality proxies increasingly accessible. We expect paired image–mesh benchmarks to become more common, which in turn enables wider practical use of 3D-C in XAI evaluation.

While our experiments focus on single-object settings, CAVE is not inherently limited to this regime. The method can be extended by first detecting object candidates and then applying our concept matching to each detected region, following standard pipelines in pose estimation and object-centric 3D understanding (Khirodkar et al., 2022). We consider this a promising direction for future work. Our current formulation assumes a fixed canonical shape, which is effective for rigid and moderately varying categories but does not directly capture highly non-rigid classes such as humans. We leave this for future work. Furthermore, our weak supervision relies on Orient-Anything (Wang et al., 2025b) for pose estimation, which, although effective, is not perfect across all object categories (cf. Fig. E3). Nevertheless, we expect the fast progress in foundation models to further strengthen this component. Finally, we discuss challenging failure cases in Appendix K.

## 8 CONCLUSION

We proposed **CAVE**, a 3D-aware image classifier that introduces concept-based NOVs to **jointly achieve OOD robustness and interpretability**, while also removing the need for ground-truth 3D poses. This enables faithful, concept-based explanations while retaining strong task performance across OOD settings. We further complement existing XAI metrics with our novel 3D-C to measure concept consistency, relaxing prior assumptions on alignment with pre-defined part annotations.

ACKNOWLEDGEMENT

Adam Kortylewski acknowledges support via his Emmy Noether Research Group funded by the German Research Foundation (DFG) under Grant No. 468670075.

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

# Interpretable 3D Neural Object Volumes
# for Robust Conceptual Reasoning

## Appendix

This supplement provides additional technical details, experimental setup, ablations, and qualitative results supporting our work on robust and interpretable 3D-aware classification with CAVE. We **strongly encourage** readers to review NOV-aware LRP in Section C, and its corresponding ablation in Section I which highlights the stability of our proposed LRP adaptation, as well as the additional *randomly sampled* qualitative examples from our method CAVE in Section L.

## A  CLASS- AND NUISANCE-WISE ANALYSIS UNDER OOD SETTINGS

We analyse class-wise performance of CAVE (weak 3D supervision) under different occlusion levels (Sec. A.1) and OOD nuisances (Sec. A.2). We further analyse whether pose errors introduced by Orient Anything estimates affect our model performance in Sec. A.3.

### A.1  CLASS-WISE ACCURACY AND 3D CONSISTENCY UNDER OCCLUSION

Overall, CAVE maintains strong class-wise accuracy and 3D consistency under mild occlusion (L1), with performance degrading gradually as occlusion increases. The drop is most pronounced for classes such as Boat, Chair, and Train. A similar trend appears in the 3D-C scores.

| Occlusion Level | Aeroplane | Bicycle | Boat | Bottle | Bus | Car | Chair | Dining Table | Motorbike | Sofa | Train | TV Monitor | all |
|---|---|---|---|---|---|---|---|---|---|---|---|---|---|
| L0 [0%] (in-distribution) | 99.67 | 99.86 | 99.26 | 99.85 | 98.48 | 99.72 | 91.78 | 99.53 | 98.79 | 99.57 | 95.90 | 100.00 | 99.0 (±0.03) |
| L1 [20 − 40% occluded] | 96.78 | 98.20 | 92.41 | 98.06 | 86.32 | 97.99 | 85.58 | 96.48 | 94.45 | 98.32 | 78.58 | 99.55 | 94.8 (± 0.12) |
| L2 [40 − 60% occluded] | 83.51 | 90.44 | 73.30 | 93.88 | 62.38 | 87.63 | 71.49 | 85.08 | 84.02 | 94.89 | 57.10 | 91.03 | 82.8 (± 0.40) |
| L3 [60 − 80% occluded] | 40.60 | 62.60 | 41.05 | 81.35 | 21.44 | 52.81 | 44.81 | 60.24 | 52.41 | 83.71 | 32.35 | 57.41 | 52.7 (± 0.96) |

Table A1: **Class-wise Accuracy on Pascal3D+ (in-distribution) and OccludedP3D+ (L1, L2, L3 occlusion)**. We report overall accuracy (not the average of class-wise accuracies) in the last column. See Fig. A1 for class-wise statistical variance across seeds.

| Occlusion Level | Aeroplane | Bicycle | Boat | Bottle | Bus | Car | Chair | Dining Table | Motorbike | Sofa | Train | TV Monitor | avg |
|---|---|---|---|---|---|---|---|---|---|---|---|---|---|
| L0 [0%] (in-distribution) | 0.220 | 0.235 | 0.174 | 0.694 | 0.343 | 0.299 | 0.430 | 0.456 | 0.248 | 0.561 | 0.385 | 0.799 | 0.404 (± 0.0014) |
| L1 [20 − 40% occluded] | 0.153 | 0.150 | 0.115 | 0.497 | 0.219 | 0.186 | 0.325 | 0.383 | 0.152 | 0.336 | 0.241 | 0.681 | 0.287 (± 0.0018) |
| L2 [40 − 60% occluded] | 0.105 | 0.104 | 0.085 | 0.380 | 0.149 | 0.117 | 0.255 | 0.288 | 0.094 | 0.204 | 0.169 | 0.581 | 0.211 (± 0.0015) |
| L3 [60 − 80% occluded] | — | 0.075 | — | 0.268 | — | 0.069 | 0.103 | 0.186 | 0.076 | 0.118 | — | 0.436 | 0.200 (± 0.0166) |

Table A2: **Class-wise 3D Consistency Scores on Pascal3D+ (L0, in-distribution) and OccludedP3D+ (L1, L2, L3 occlusion)**. We report the average 3D-C scores as mean (± std) and class-wise mean 3D-C scores only for readability. We denote — for classes that do not have consistent concepts. See Fig. A2 for class-wise statistical variance across seeds.

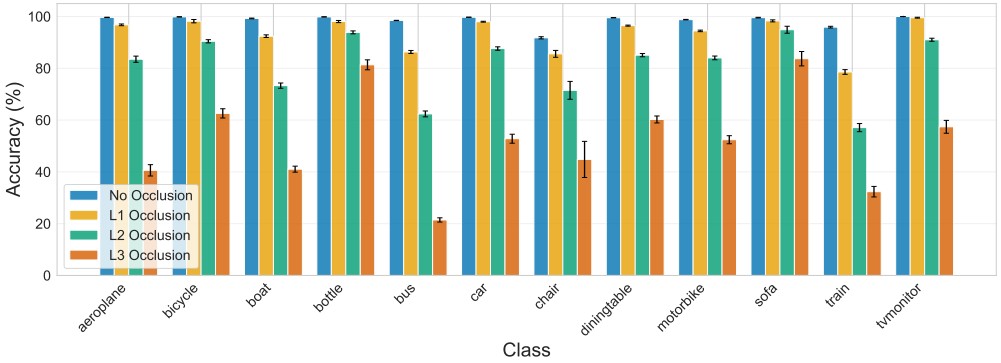

Figure A1: **Class-wise accuracy on Pascal3D+ (L0, in-distribution) and OccludedP3D+ (L1, L2, L3 occlusion) across 10 random seeds**. For precise values, refer Tab. A1.

### A.2  OOD ATTRIBUTE (NUISANCE)-WISE ACCURACY AND 3D CONSISTENCY

We show class-wise and attribute (OOD nuisance)-wise performance of our CAVE with weak 3D supervision on OOD-CV dataset (Zhao et al., 2022). Across OOD attributes, we observe that *pose* is the most challenging nuisance factor, leading to the largest drop in both accuracy and 3D-C across classes. *Weather* is the second most difficult attribute, likely due to reduced visibility and contrast in rainy or foggy conditions, as we have seen in qualitative examples in Fig. 2. In contrast, *context*, *shape*, and *texture* shifts result in comparatively moderate changes, with performance remaining relatively stable across most classes. These trends hold consistently across 10 random seeds and are reflected in both accuracy and 3D consistency metrics.

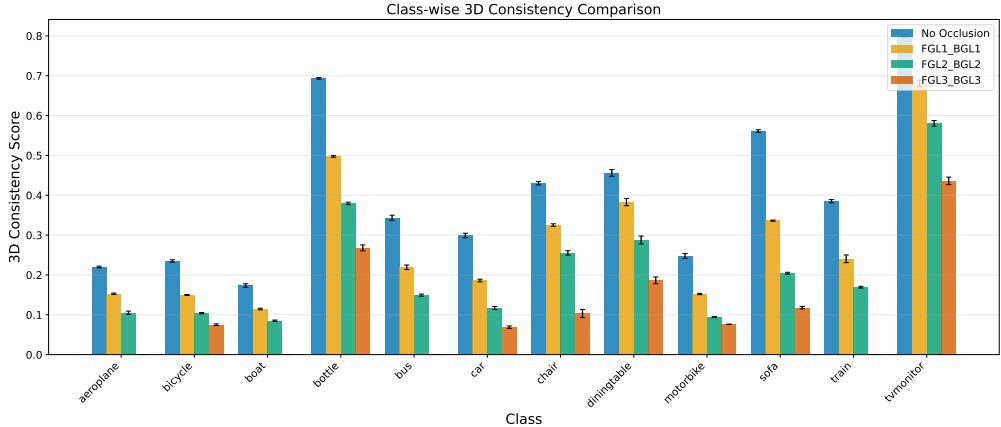

Figure A2: **Class-wise 3D-C on Pascal3D+ (L0, in-distribution) and OccludedP3D+ (L1, L2, L3 occlusion) across 10 random seeds**. For precise values, refer Tab. A2.

| OOD Attribute | Context | Pose | Shape | Texture | Weather | **all** |
|---|---|---|---|---|---|---|
| Accuracy (%, ↑) | 82.95 (± 0.20) | 72.77 (± 0.77) | 81.22 (±0.65) | 83.27 (± 0.61) | 77.61 (± 0.39) | 80.3 (± 0.27) |
| 3D-C (↑) | 0.236 (± 0.006) | 0.227 (± 0.018) | 0.239 (± 0.004) | 0.238 (± 0.004) | 0.234 (± 0.009) | 0.235 (± 0.002) |

Table A3: **Attribute-wise performance, both accuracy and 3D-C across 10 random seeds**, where *context*, *pose*, *shape*, *texture*, and *weather* constitute $25\%$, $8\%$, $15\%$, $22\%$, and $30\%$ of the OOD-CV dataset. The last column reports the overall score computed over all attributes. We further break down the scores class-wise in Tab. A4 & Tab. A5 .

| OOD Attribute | ✈ Aeroplane | 🚲 Bicycle | 🚗 Boat | 🚌 Bus | 🚗 Car | 🪑 Chair | ■ Dining Table | 🏍 Motorbike | 🚚 Sofa | 🖥 Train | all |
|---|---|---|---|---|---|---|---|---|---|---|---|
| **Context** | 81.10 (± 1.06) | 88.92 (± 0.42) | 89.83 (± 1.34) | 66.40 (± 0.92) | 69.86 (± 0.91) | 74.05 (± 0.85) | 85.19 (± 0.67) | 84.66 (± 0.97) | 94.62 (± 0.43) | 75.81 (± 1.20) | 82.95 (± 0.20) |
| **Pose** | 87.59 (± 1.78) | 86.77 (± 1.38) | 79.39 (± 2.05) | 21.91 (± 1.75) | 65.56 (± 2.17) | 31.54 (± 2.43) | 33.33 (± 0.00) | 92.86 (± 0.00) | 75.00 (± 0.00) | 89.50 (± 3.69) | 72.77 (±0.77) |
| **Shape** | 93.88 (± 0.57) | 90.89 (± 1.45) | 87.50 (± 1.96) | 90.00 (± 0.00) | 65.52 (± 2.30) | 71.17 (± 1.20) | 62.33 (± 2.59) | 93.49 (± 1.47) | 85.47 (± 0.44) | 63.33 (± 2.87) | 81.22 (± 0.65) |
| **Texture** | 97.43 (± 1.16) | 84.05 (± 1.00) | 89.19 (± 1.56) | 90.17 (± 0.91) | 90.62 (± 1.77) | 58.35 (± 0.89) | 83.68 (± 1.32) | 91.11 (± 0.65) | 88.62 (± 1.01) | 69.50 (± 4.08) | 83.27 (± 0.61) |
| **Weather** | 84.69 (± 0.85) | 92.63 (± 1.22) | 81.83 (± 1.62) | 55.82 (± 1.52) | 54.74 (± 1.25) | 87.14 (± 2.02) | 30.00 (± 0.00) | 91.28 (± 0.64) | 75.00 (± 0.00) | 84.31 (± 1.46) | 77.61 (± 0.39) |

Table A4: **Attribute-wise accuracy per class of OOD-CV dataset averaged across 10 random seeds.** We report the accuracy for each Pascal3D+ class under each OOD attribute (context, pose, shape, texture, weather), averaged across seeds. For each OOD attribute, we additionally report the overall accuracy computed over the union of all classes (rather than averaging class-wise accuracies), along with its statistical variance across seeds (last column).

| OOD Attribute | ✈ Aeroplane | 🚲 Bicycle | 🚗 Boat | 🚌 Bus | 🚗 Car | 🪑 Chair | ■ Dining Table | 🏍 Motorbike | 🚚 Sofa | 🖥 Train | avg |
|---|---|---|---|---|---|---|---|---|---|---|---|
| **Context** | 0.152 | 0.159 | 0.129 | 0.235 | 0.164 | 0.239 | 0.394 | 0.175 | 0.401 | 0.309 | 0.236 (± 0.006) |
| **Pose** | 0.155 | 0.164 | 0.127 | 0.247 | 0.169 | 0.212 | 0.375 | 0.169 | 0.338 | 0.307 | 0.227 (± 0.018) |
| **Shape** | 0.157 | 0.162 | 0.131 | 0.246 | 0.148 | 0.242 | 0.392 | 0.176 | 0.422 | 0.310 | 0.239 (± 0.004) |
| **Texture** | 0.150 | 0.173 | 0.136 | 0.235 | 0.163 | 0.235 | 0.393 | 0.174 | 0.416 | 0.300 | 0.238 (± 0.004) |
| **Weather** | 0.153 | 0.156 | 0.131 | 0.228 | 0.156 | 0.229 | 0.411 | 0.176 | 0.387 | 0.305 | 0.234 (± 0.009) |

Table A5: **Attribute 3D-C (↑) scores per class averaged across 10 random seeds.** We report mean 3D-C attribute scores for each Pascal3D+ class and OOD factor: context, pose, shape, texture, and weather. For class-wise attribute-wise statistical variance, refer Fig. A3. We further report average score across classes for each attribute along with its statistical variance across seeds (last column).

## A.3   CLASS-WISE SENSITIVITY TO POSE ESTIMATION ERROR

We further examine class-wise accuracy and 3D-C to training pose estimation error in Fig. A4. We find that accuracy does not strongly degrade for classes with substantial pose ambiguity (e.g., Dining Table, Boat with azimuth error > 20°). In contrast, 3D-C exhibits a mild downward trend as azimuth error increases, which is expected because 3D consistency directly measures geometric alignment. We hypothesise that, since CAVE is trained on Orient Anything's estimated poses, it inherits symmetric ambiguities present in those estimates. At inference time, symmetric objects (e.g., a left–right symmetric boat) still activate the correct class concepts even if the pose is mirrored; a

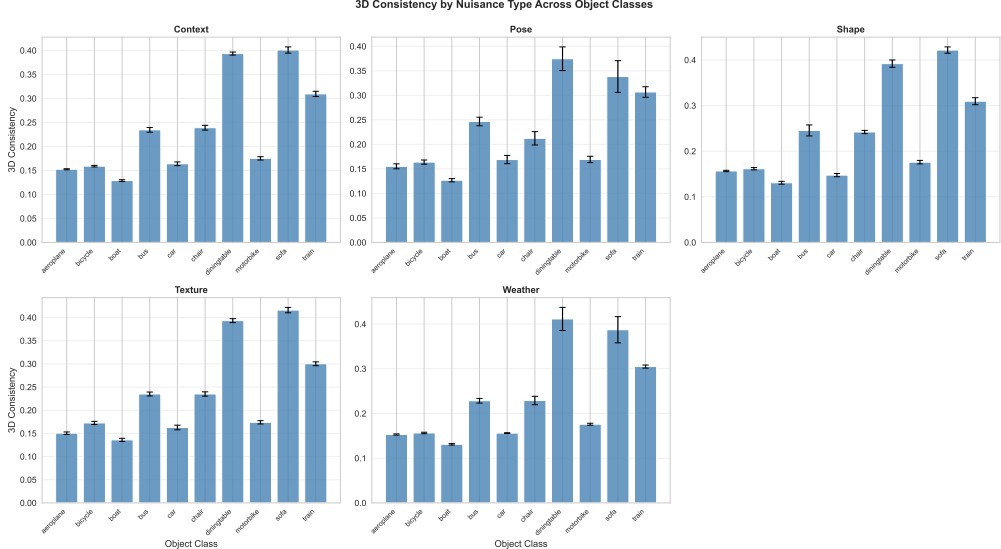

Figure A3: **Class-wise 3D-C for each OOD attribute in OOD-CV dataset across 10 random seeds**. For precise values, refer Tab. A5.

concept trained to fire on the left may instead fire on the right, without affecting the final classification. This symmetry-induced "flip" effect naturally explains why classification accuracy remains stable while 3D-C scores decrease for classes with higher pose error.

| Metric | ✈ Aeroplane | 🚲 Bicycle | 🚤 Boat | 🍼 Bottle | 🚌 Bus | 🚗 Car | 🪑 Chair | ⬛ Dining Table | 🏍 Motorbike | 🛋 Sofa | 🚆 Train | 🖥 TV Monitor |
|---|---|---|---|---|---|---|---|---|---|---|---|---|
| Azimuth error (°) | 10.14 | 12.92 | 26.94 | 6.86 | 7.25 | 6.71 | 10.09 | 20.15 | 12.03 | 8.58 | 8.46 | 9.53 |

Table A6: **Class-wise Mean Azimuth Pose Error (in degrees °)** between Orient-Anything predictions and ground-truth pose on Pascal3D+.

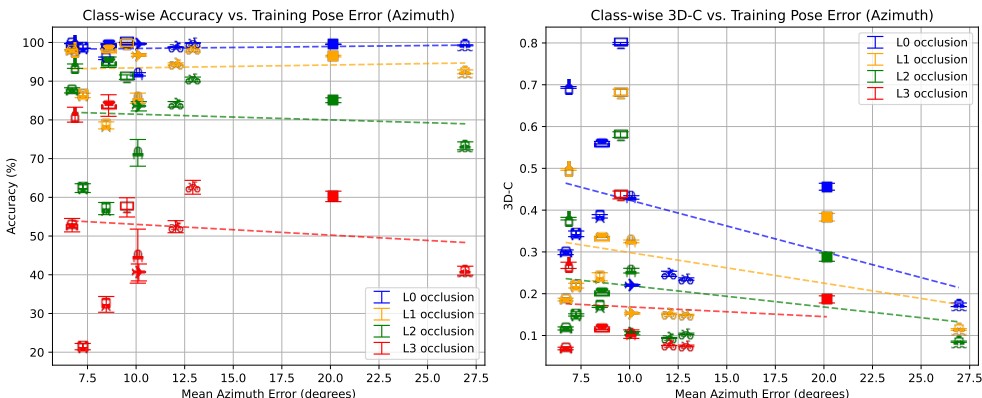

Figure A4: **Class-wise accuracy (left) and 3D-C (right) as a function of training pose errors (azimuth) in degree (°) across occlusion levels.** The icons and their corresponding class names are defined in Tab. A6.

## B    ON IMPLICIT VERSUS EXPLICIT INTERPRETABILITY

Interpretability methods differ not only in whether they provide post-hoc explanations or are faithful-by-design, but also in *how* the underlying concepts acquire meaning. Two complementary paradigms have emerged in the literature.

**Explicit interpretability.**    Models in this category impose architectural or training-time constraints that bind internal units directly to human-understandable notions. CBMs (Koh et al., 2020; Oikari-nen et al., 2023) explicitly supervise units to correspond to semantic concepts such as object attributes or part labels. Prototype-based networks (e.g., ProtoPNet (Chen et al., 2019) and its variants (Wang et al., 2021; Nauta et al., 2023a; Wang et al., 2025a)) explicitly ground units by enforcing a prototype layer and projecting prototypes onto representative input patches, yielding "this looks like that" explanations. In both cases, the interpretability is *explicit* because the model is guided toward semantic or visually grounded concepts during training. These approaches are therefore faithful-by-design and offer direct, easily inspectable explanations.

**Implicit interpretability.**    In contrast, implicitly interpretable approaches do not impose semantic supervision or dedicated prototype objectives. Instead, they rely on structure that *emerges* from the model's learned representation. Post-hoc concept discovery methods such as CRAFT (Fel et al., 2023b), ICE (Zhang et al., 2021), MCD (Vielhaben et al., 2023), and PCX (Dreyer et al., 2024) extract interpretable structure from trained models by clustering or factorising latent activations. These methods provide implicit interpretability but are not faithful, as their concepts do not participate in the model's forward computation.

CAVE occupies a distinct position within this paradigm: although its concepts are obtained implicitly through unsupervised clustering rather than semantic or prototype supervision, these units are geometrically grounded and integrated directly into the model's decision pathway. This yields *faithfulness-by-design* while still achieving interpretability implicitly from the geometric and feature structure of the NOV representation. The resulting region-level concepts differ from the atomic Gaussian features in NOVUM (Jesslen et al., 2024) by providing a coarser, more coherent, and easier-to-inspect basis for prediction that can be visualised post-hoc with our adapted LRP. Importantly, each CAVE centroid (concept) is a linear combination of NOVUM's Gaussian features and therefore lies within the span of the original predictive units. CAVE's concept dictionary thus remains in the same predictive subspace while offering a more compact and interpretable representation, preserving both accuracy and faithfulness.

**Inherent interpretability of CAVE.**    To contextualise CAVE's interpretability within the literature, we draw on Molnar's taxonomy of intrinsic (inherent) interpretability. Molnar highlights that models can be inherently interpretable even when they "mix both interpretability by design and post-hoc interpretability", with his examples being models whose structure makes their computation transparent and complexity appropriately constrained, even if individual components require post-hoc visualisation to be inspected (Molnar, 2025). CAVE thus fits the same paradigm of inherent interpretability: its prediction is transparent by design, decomposing exactly over a small dictionary of region-level concepts derived from Gaussian features, while our adapted LRP is used for post-hoc concept visualisation.

## C  LAYER-WISE RELEVANCE PROPAGATION FOR 3D-AWARE CLASSIFIERS

As mentioned in Section 4.1, layer-wise relevance propagation (LRP) defined for standard architectures unfaithfully leaks relevances when attributing NOV-based concepts to image pixels. To enable tracing relevance from the model's prediction backward through the concept-based NOVs $\mathcal{H}$ to the input image (Bach et al., 2015; Otsuki et al., 2024), we extend LRP to our 3D-aware setting with volumetric object representation. In doing so, we also ensure that the key conservation property of LRP is preserved, i.e., total relevance remains constant throughout the network (Otsuki et al., 2024).

In the following, we briefly explain vanilla LRP with $\epsilon-$rule that is defined for standard feedforward network (cf. Sec. C.1), and formulate **NOV-aware LRP** for NOVUM and CAVE-like architectures (cf. Fig. C1, Sec. C.2). Specifically, this is tailored to two layers: (i) upsampling by concatenation, and (ii) volume concept matching, that do not exist in standard architectures. We further show how to estimate concept-wise importance scores and visualise concepts in Section C.3.

### C.1  VANILLA LRP WITH $\epsilon-$RULE

Standard LRP with $\epsilon-$rule propagates relevances backward through network layers $l+1$ to layer $l$:

$$R_i = \sum_j \frac{a_i w_{ij}}{\sum_{i'} a_{i'} w_{i'j} + \epsilon \operatorname{sign}(a_{i'} w_{i'j})} R_j \tag{C.1}$$

where $a_i$ is the activation of neuron $i$ in layer $l$, $w_{ij}$ is the weight connecting neuron $i$ in layer $l$ to neuron $j$ in layer $l+1$, $R_j$ is the relevance of neuron $j$ and $R_i$ is the relevance to propagate back to neuron $i$. Here the $\epsilon - rule$ is introduce to dampen relevance when the denominator gets arbitrarily small (Springenberg et al., 2015). An important property of LRP is its relevance conservation, meaning:

$$\sum_i R_i = \sum_j R_j \tag{C.2}$$

which is often violated if relevances are not attributed faithfully through the network.

### C.2  LRP WITH CONSERVATION FOR CAVE

As described, vanilla LRP does not handle non-standard operation such as upsampling by concatenation (no weight matrix defines a simple mapping from input to output channels), or concept matching via NOVs, which introduces structured multiplicative interactions between image features and NOV-based concepts. We thus formulate the LRP redistribution rule for these layers.

*(i) Upsampling by concatenation.* Similar to NOVUM (Jesslen et al., 2024), the basic CAVE contains a feature extractor which consists of a ResNet-50 backbone followed by three upsampling layers with concatenation. In this design, each upsampling layer combines feature maps from earlier layers, preserving fine-grained details important for 3D-aware classification. Let us consider an upsampling layer $U$, which concatenates in the channel dimension feature maps $A_v \in \mathbb{R}^{H_1 \times W_1 \times C_1}$ and $A_{v+l} \in \mathbb{R}^{H_2 \times W_2 \times C_2}$ from two non-consecutive layers. $A_{v+l}$ is padded to $A'_{v+l} \in \mathbb{R}^{H_1 \times W_1 \times C_2}$ to maintain spatial consistency. We denote this concatenation operation as $A_U = A_v \oplus A'_{v+l}$. Let us further denote $R_U, R_v$, and $R_{v+l}$ as the relevance scores at the upsampling layer and two non-consecutive layers, respectively. $R'_{v+l}$ is the padded relevance of $R_{v+l}$. By conservation property, it should hold that $R_U = R_v \oplus R'_{v+l}$. We define a relevance-preserving splitting as follows:

$$R'_{v+l} = R_U[: H_1, : W_1, : C_2], \quad R_{v+l} = R'_{v+l} \cdot \mathbb{1}(A_{v+l})$$

$$R_v = R_U[H_1 : 2H_1, W_1 : 2W_1, C_2 : (C_1 + C_2)]$$

where $\mathbb{1}(.)$ is the indicator function that is 1 for original non-padded elements in $A_{v+l}$ and 0 otherwise. After three upsampling layers, we obtain our feature map $F_x$ for 3D-aware concept matching.

*(ii) Volume concept matching.* For the concept matching $\Phi(F_x, \mathcal{H})$ between NOVs-based concepts $\mathcal{H}$ and image features $F_x \in \mathbb{R}^{H \times W \times C}$, let the output be $s_\Phi \in \mathbb{R}^{H \times W}$. We further denote $R_\Phi \in \mathbb{R}^{H \times W}$ as the relevance for the feature matching layer, and $R_{F_x} \in \mathbb{R}^{H \times W \times C}$ as the relevance of the feature map $F_x$. To ensure spatial consistency, a relevance score $r_i \in R_\Phi$ is first directly mapped to

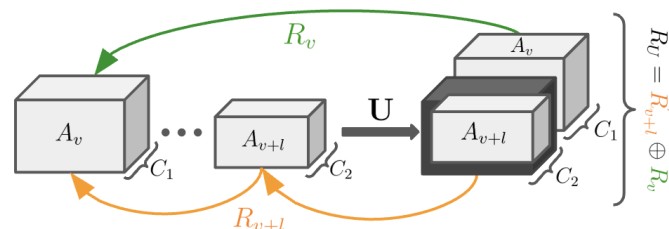

*Relevance splitting at Upsampling layer* **U**

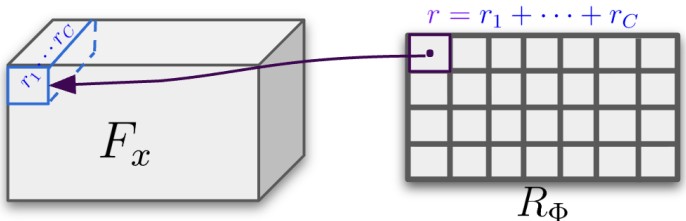

*Relevance splitting at Matching layer* $\Phi$

Figure C1: **NOV-aware relevance propagation in CAVE**. **Top**: At upsampling layer $U$, feature maps $A_v$ and $A_{v+l}$ from non-consecutive layers are concatenated along channel dimension after padding for consistency. Relevance score $R_U$ is split into $R_v$ and $R'_{v+l}$, where $R'_{v+l}$ is padded $R_{v+l}$. **Bottom**: We ensure spatial consistency by mapping relevance $R_\Phi$ at matching layer to corresponding feature $f_i \in F_x$, then distributing channel-wise with NOV-weighted feature importance.

the corresponding feature $f_i \in F_x$, and then further distributed *channel-wise* for each channel $c$ (cf. Fig. C1). We get

$$R_{F_x}^{\text{spatial}}(i) = R_\Phi(i) = r = \sum_{j=1}^{C} r_j = \sum_{j=1}^{C} (f_i \odot \mathcal{H}_{f_i})(j),$$

$$R_{F_x}(i,c) = R_{F_x}^{\text{spatial}}(i) \cdot \frac{(f_i \odot \mathcal{H}_{f_i})(c)}{\sum_j (f_i \odot \mathcal{H}_{f_i})(j)}$$

where $\mathcal{H}_{f_i}$ denotes the matching NOV-based concept for $f_i$, and $\odot$ denotes element-wise multiplication. We integrate our formulation with LRP with conservation for ResNet-50 (Otsuki et al., 2024). In Section I, we provide the full comparisons between our NOV-aware LRP and other common attribution methods, including vanilla LRP.

## C.3 EXPLANATIONS THROUGH VOLUME ALIGNMENT

**Concept Importance through NOVs.** Our next goal is to estimate the importance of all class-wise concepts in $\mathcal{H}_y^*$ using the NOV-aware LRP attributions. The intuition behind this is straightforward: starting from the model's softmax output, we trace back relevance to each concept. The concept importance score is then determined using its $x\%$ quantile (e.g., 90th percentile) across the training dataset to capture the most representative high-relevance values while being robust to outliers. We denote the matching NOV-based concept in $\mathcal{H}$ for the feature map $F_x$ as $\Delta_{F_x \to \mathcal{H}}$. We compute the relevance for a concept $h_y^{(j)} \in \mathcal{H}$ of class $y$ by aggregating the relevance scores, denoted $R_\phi$, at all spatial locations $i$ where $\Delta_{F_x \to \mathcal{H}}(i) = h_y^{(j)}$. Formally, it is defined as:

$$R_{h_y^{(j)}} = \sum_{i \in H \times W} R_\phi \cdot \mathbb{1}_{\Delta_{F_x \to \mathcal{H}}(i) = h_y^{(j)}}$$

**Concept visualisation.** We visualise class-wise NOV concepts $h_y^{(j)} \in \mathcal{H}$ by redistributing their relevance scores $R_{h_y^{(j)}}$ (Sec. C.2) to pixel space and highlighting locations with positive contributions.

# D    FURTHER DETAILS ON 3D CONSISTENCY OF CONCEPTS

As introduced in Section 5, our 3D consistency (3D-C) metric evaluates whether a concept remains consistent across test images of the same class. Unlike prior metrics (Huang et al., 2023; Behzadi-Khormouji & Oramas, 2023; Huang et al., 2024), our 3D-C removes the need for part annotations and avoids assuming that concepts must align with human-defined parts, an assumption often not enforced explicitly during training. Instead, 3D-C complements existing measures such as spatial localisation and object coverage by leveraging object geometry to project concept attributions onto a common 3D space (cf. Fig D1). In this section, we further describe 3D-C visualisation details.

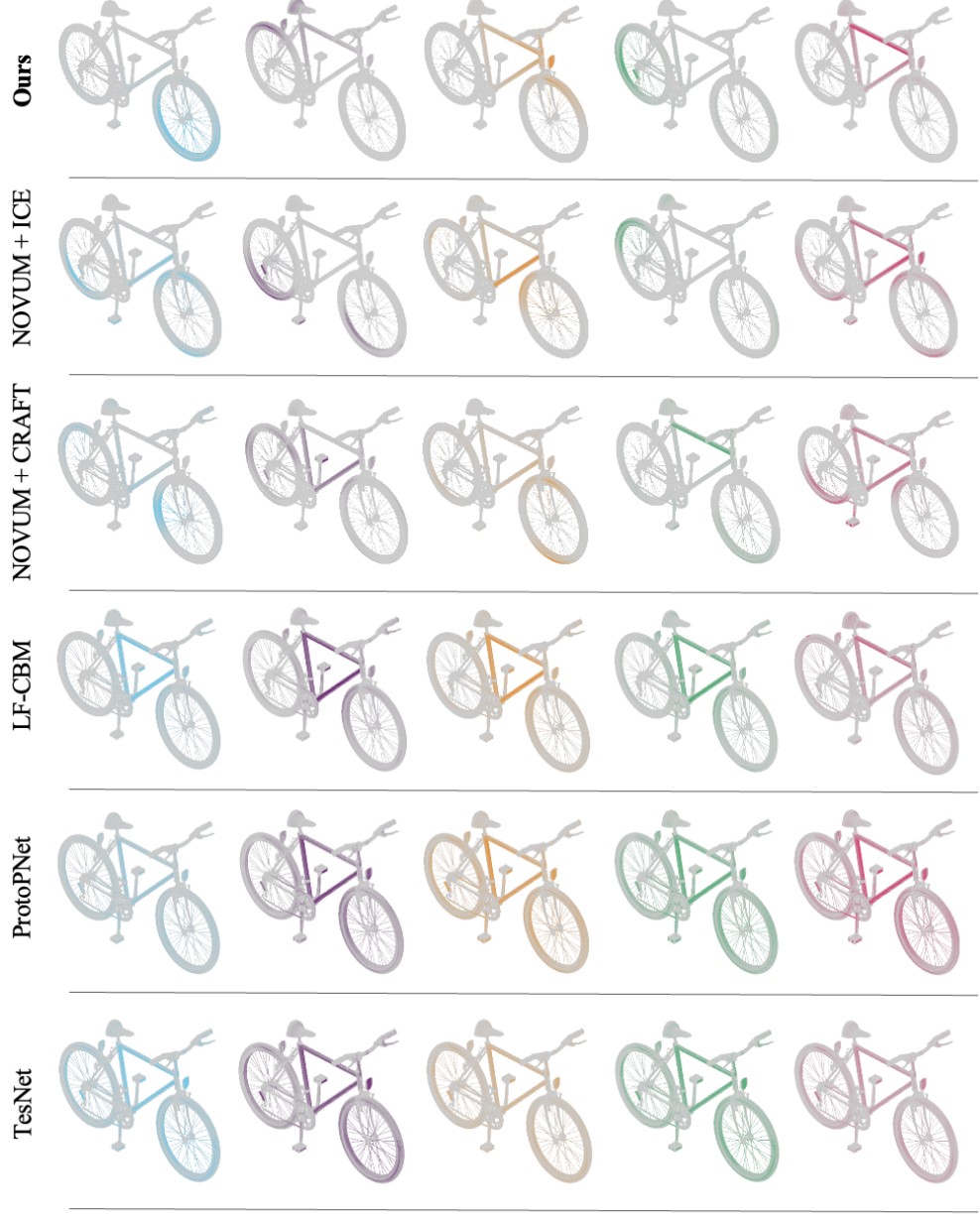

Figure D1: **CAVE (Ours) shows more consistent concepts compared to both post-hoc and ad-hoc methods**. Here, we show top-5 most important (left to right, color-coded) concepts of class Bicycle for each method. CAVE visually shows better concept consistency in comparision to strong baselines (see also quantitative evaluation in Tab. 1).

**Object mesh.** For each object class $y$, we load the canonical CAD mesh available in the datasets such as Pascal3D+ (Xiang et al., 2014) and ImageNet3D (Ma et al., 2024). Since these meshes follow a CAD-centric coordinate convention that is inconsistent with the camera system of PyTorch3D, we apply a pre-processing step to re-align the axes, i.e.:

$$\begin{pmatrix} x \\ y \\ z \end{pmatrix}_{\text{CAD}} \mapsto \begin{pmatrix} x \\ z \\ -y \end{pmatrix}_{\text{camera}}$$

**Rendering.** Rendering is performed with PyTorch3D. We use an image canvas of $640 \times 800$, and adjust camera intrinsics such that focal lengths are scaled to preserve field of view, and the principal point is placed at the image center. Addtionally, we adopt a spherical camera at radius 15, elevation $30°$, and azimuth $-40°$. Rasterisation is done with a single face per pixel and no blur radius.

**Concept attribution projections onto object mesh.** Concept attributions are then projected onto the mesh. For each concept $h$, we use its positive attributions $A^+(x, h)$ defined as:

$$A^+(x, h) \ = \ \max\left(0, A(x, h)\right)$$

to highlight regions that contribute to this concept $h$. We assign these values as per-face textures on the CAD mesh and render with PyTorch3D, where barycentric interpolation distributes per-face attributions to pixels, yielding a dense per-pixel attribution map.

**3D-C visualisation.** For visualisation, we first render a neutral base mesh to provide geometric context and occlusion boundaries. Positive attributions $A^+(x, h)$ for a concept $h$ are then overlaid as a heatmap on this render, given the pixel-to-face mapping from the projection step. To characterise the overall spatial footprint of the concept $h$, we aggregate attributions across all test images $x \in \mathcal{X}_y$ of class $y$, i.e.:

$$A^+_{\text{agg}}(h) \ = \ \sum_{x \in \mathcal{X}_y} A^+(x, h)$$

followed by min–max normalisation to $[0, 1]$ for the final visualisation. This naturally provides a direct visual illustration to the 3D-C metric scores, showing concept consistency across instances.

**Limitations and scope.** While our 3D-C metric relies on class-level CAD meshes and face-level attribution aggregation, which may smooth out very fine-grained details, this design ensures scalability across large datasets and consistency of evaluation. By not enforcing alignment with human-annotated parts, 3D-C sidesteps annotation biases and instead directly reflects the model's own concept structure. Finally, canonical meshes do not capture all intra-class shape variations. Yet, they provide a stable reference geometry that enables meaningful comparisons of concept consistency across object instances.

# E    LEARNING WITH WEAK SUPERVISION

Recent advance in object orientation estimation, such as Orient-Anything (Wang et al., 2025b), allows us to relax the constraints on pose annotations during training 3D-aware classifiers. We study how accurate Orient-Anything estimated poses are compared to ground-truth poses in Pascal3D+ (Xiang et al., 2014) and ImageNet3D (Ma et al., 2024) in Tab. E1. With the tolerance $X = 60°$, Orient-Anything poses achieves near-perfect accuracy on polar and rotation estimation on both Pascal3D+ and ImageNet3D. For azimuth estimation, Orient-Anything performs better on Pascal3D+ ($\sim 96\%$) while struggles on larger-scale dataset like ImageNet3D ($\sim 72\%$). A stricter tolerance thresholds introduce performance drops on both datasets. Our inspection shows that failures of Orient-Anything in predicting object pose often occur with symmetric objects such as tables and boats, where correct orientation is often ambiguous. See also Fig. E2 and Fig. E3 for qualitative visualisations.

| Class Label | Azimuth Estimation ↑ | | | Polar Estimation ↑ | | | Rotation Estimation ↑ | | |
|---|---|---|---|---|---|---|---|---|---|
| | $60°(\frac{\pi}{3})$ | $30°(\frac{\pi}{6})$ | $10°(\frac{\pi}{18})$ | $60°(\frac{\pi}{3})$ | $30°(\frac{\pi}{6})$ | $10°(\frac{\pi}{18})$ | $60°(\frac{\pi}{3})$ | $30°(\frac{\pi}{6})$ | $10°(\frac{\pi}{18})$ |
| Pascal3D+ | 96.70 | 93.92 | 71.58 | 99.77 | 98.82 | 86.25 | 99.57 | 99.14 | 93.09 |
| ImageNet3D | 72.32 | 48.06 | 27.63 | 94.69 | 79.92 | 51.31 | 96.92 | 95.10 | 90.11 |

Table E1: **Accuracy ($\%$, ↑) of zero-shot orientation estimation by Orient–Anything (Wang et al., 2025b) on training images of Pascal3D+ and ImageNet3D**. Accuracy is reported within different tolerances of $\pm X°$ with $X° = \{60°, 30°, 10°\}$ (i.e., $\{\pi/3, \pi/6, \pi/18\}$ radians).

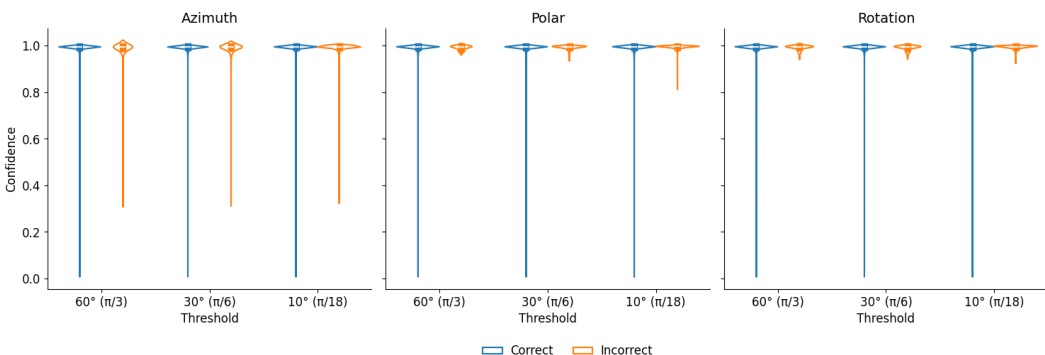

Figure E1: **Prediction Confidence on Pascal3D+ of Orient–Anything (Wang et al., 2025b) across Azimuth, Polar and Rotation** with respect to different tolerences of $\pm X°$ with $X° = \{60°, 30°, 10°\}$ (i.e., $\{\pi/3, \pi/6, \pi/18\}$ radians). Orient–Anything produces consistently high-confidence predictions across both correctly and incorrectly classified samples.

Orient-Anything poses can be treated as pseudo ground-truth, enabling weakly supervised training of CAVE without requiring manual pose annotations. Importantly, CAVE maintains competitive performance across all datasets, even under weaker supervision (see Tab. E2), while being substantially more parameter-efficient by representing each class with only $D = 20$ concepts (roughly $98\%$ fewer than NOVUM's 1130 Gaussians per class).

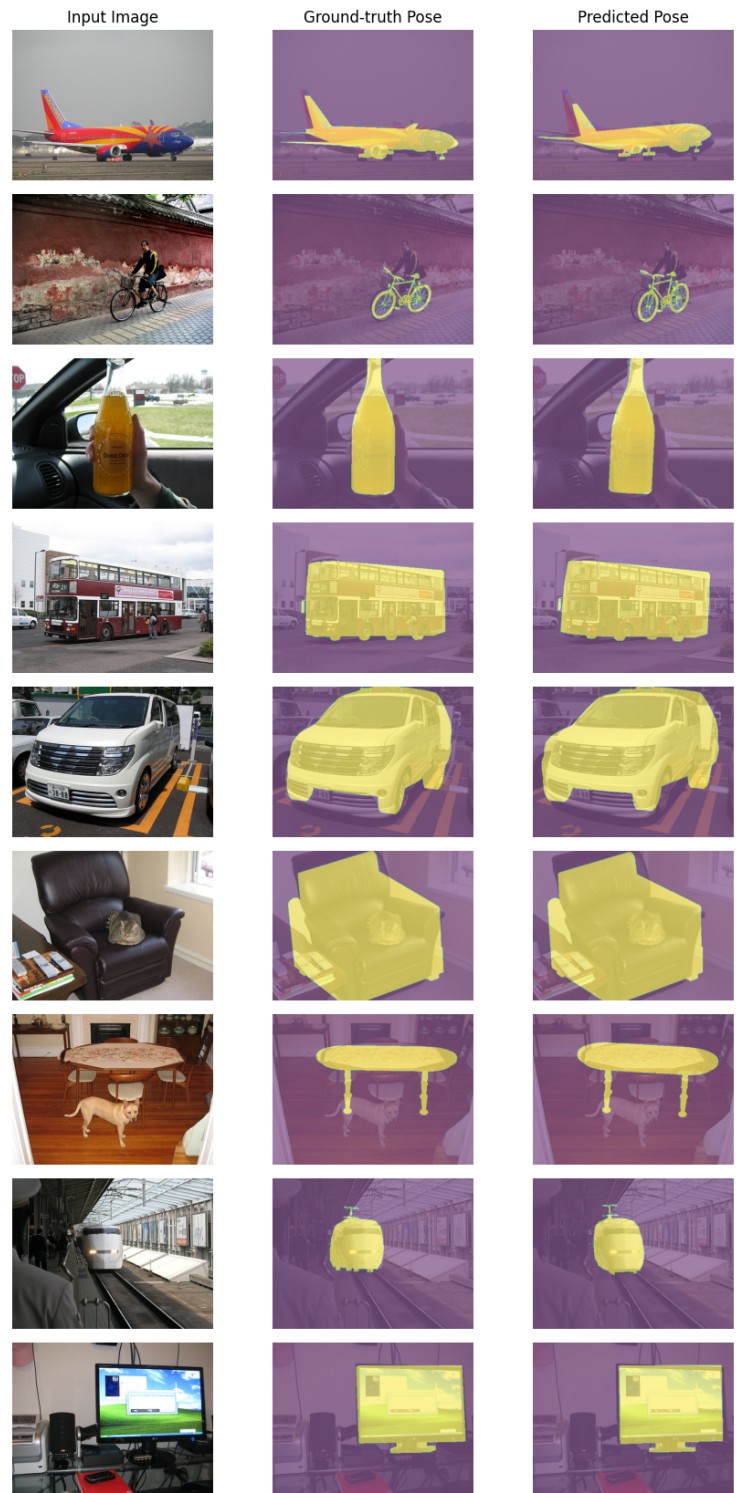

Figure E2: **Qualitative visualisation of accurate pose predictions** (azimuth error $< 10°$) by Orient–Anything (Wang et al., 2025b).

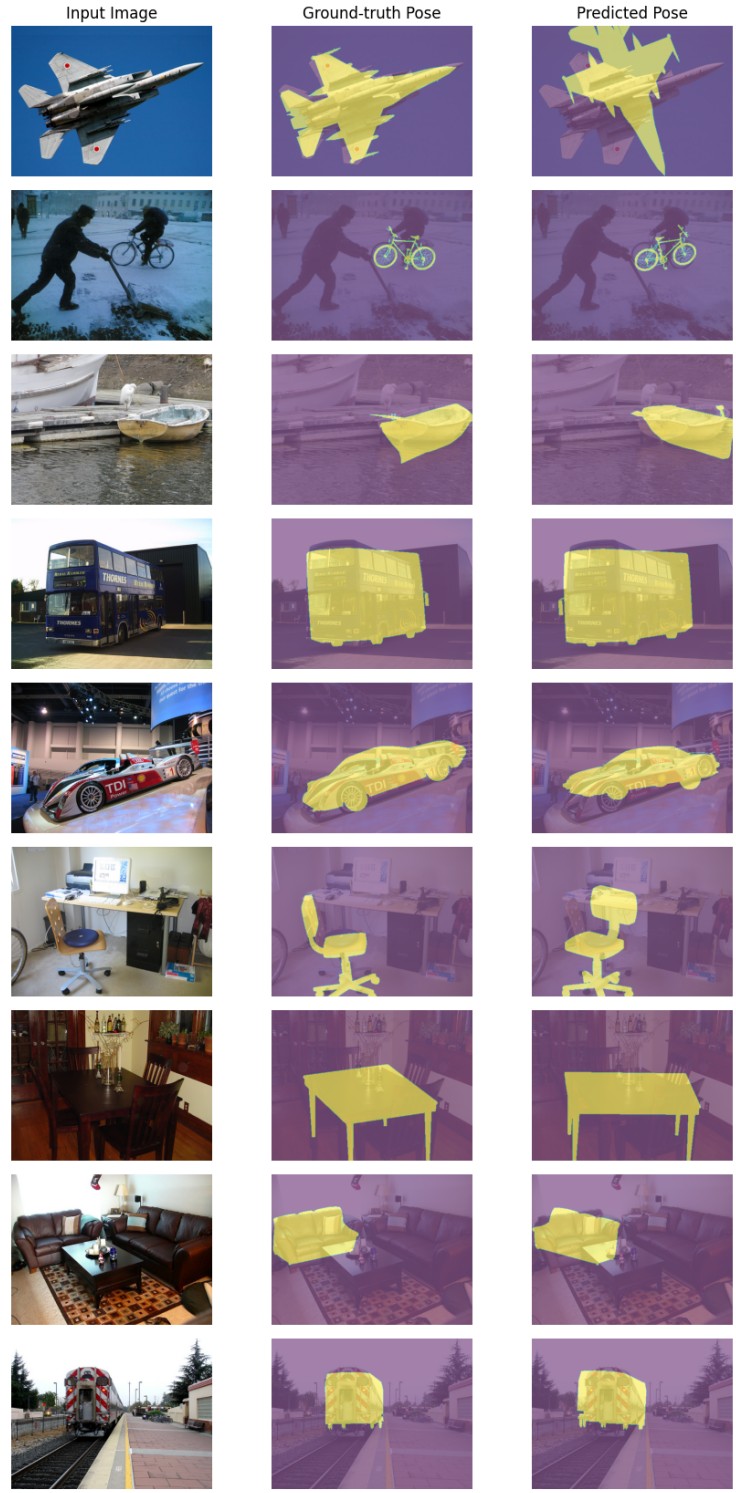

Figure E3: **Qualitative visualisation of inaccurate pose predictions** (azimuth error $> 60°$) by Orient–Anything (Wang et al., 2025b).

| Models | Ground-truth 3D Pose | | | | Orient-Anything 3D Pose | | | |
|---|---|---|---|---|---|---|---|---|
| | P3D+ | IN-3D | Occ-P3D+ | OOD-CV | P3D+ | IN-3D | Occ-P3D+ | OOD-CV |
| NOVUM | **99.5** | 88.3 | **81.7** | 81.3 | 98.4 | **85.7** | 77.3 | 79.7 |
| CAVE (ours) | 99.4 | **88.5** | 81.5 | **84.0** | **99.0** | 84.7 | **77.4** | **80.4** |

Table E2: **Accuracy** ($\%$, $\uparrow$) **of NOVUM (Jesslen et al., 2024) and CAVE (ours)** under full supervision (ground-truth pose) and less supervision (generated pose via Orient-Anything (Wang et al., 2025b)). Results are reported on two in-distribution datasets Pascal3D+ (P3D+) and ImageNet3D (IN-3D), and two OOD datasets OccludedP3D+( Occ-P3D+) and OOD-CV. Best scores are in **bold**. CAVE uses $D = 20$ concepts/class, roughly $98\%$ sparser than NOVUM (1130 Gaussians/class).

# F   3D-AWARE CLASSIFICATION WITH NOVUM

**Training.** NOVUM (Jesslen et al., 2024) is trained using a feature-extractor backbone such as ResNet-50 and, for each class, a neural object volume (NOV) composed of 3D Gaussian primitives that emit feature vectors, as defined in Section 3. Full details of the training pipeline can be found in the original paper, but we provide here an overview of the different objectives that come into play during the training of NOVUM. During training, three main objectives are used:

- *Intra-object discriminative loss:* Gaussians $g_l \in \mathcal{G}_y$, also can be written as $g_y^{(l)}$ to be consistent with the main paper, of a class NOV are trained to produce distinguishable features, thus different object regions are represented by different primitives.

- *Discriminative feature loss across classes:* features produced by Gaussians of one class $\bar{y}$, i.e., $g_m \in \mathcal{G}_{\bar{y}}$, are encouraged to be distinct from those of other classes $y$.

- *Background contrastive loss:* the model is trained to separate object features $f_i \in F_x$ from background features $\mathcal{B}$, ensuring focus on object-specific regions.

Formally, we define the constrastive learning

$$\mathcal{L}(\mathcal{G}, \mathcal{B}) = -\sum_{y}\sum_{k=1}^{K} o_k \cdot \log \frac{e^{\kappa f_{k\to i} \cdot g_k}}{\displaystyle\sum_{\substack{g_l \in \mathcal{G}_y \\ g_l \notin \mathcal{N}_k}} e^{\kappa f_{k\to i}\cdot g_l} + \omega_\beta \sum_{\beta_n \in \mathcal{B}} e^{\kappa f_{k\to i}\cdot \beta_n} + \omega_{\bar{y}} \sum_{g_m \in \mathcal{G}_{\bar{y}}} e^{\kappa f_{k\to i}\cdot g_m}}, \quad \text{(F.1)}$$

where $o_k \in \{0,1\}$ indicates the Gaussian visibility, $\kappa$ is a concentration hyperparameter which determines the spread of the von-Mieses-Fisher distribution of the Gaussians $g_k$, $\omega_{\mathcal{B}}$ and $\omega_{\bar{y}}$ indicate the probability that an image feature $f_i \in F_x$ corresponds to the background features $\mathcal{B}$ or Gaussians of a class $\bar{y}$, and $\mathcal{N}_k = \{g_r : \|\mu_k - \mu_r\| < \delta, k \neq r\}$ is the neighborhood of a Gaussian $g_k$ with mean $\mu_k$ defined within a radius $\delta$. Furthermore, $f_{k\to i}$ denotes the extracted feature $f_i$ that Gaussian $g_k$ projects to.

Importantly, all these losses require 3D pose supervision, which is provided by the dataset. Since each Gaussian primitive must be projected from 3D to the 2D image plane, the knowledge of the object pose is necessary. We relax this constraint in our CAVE with weak supervision from Orient-Anything (Wang et al., 2025b) estimated pose, and find that replacing ground-truth poses with noisy pseudo ground-truth still yields strong performance in in-distribution and OOD settings (cf. Sec. E). Additionally, we note that during training, the intra-object discriminative loss encourages nearby Gaussians to remain close in feature space while pushing apart those that are spatially distant. This reduces the influence of their spatial arrangement on the learned representation. To support this, NOVUM demonstrates that replacing the underlying cuboid shape with a sphere, an ellipsoid or a prototype CAD has minimal effect on model accuracy (cf. Appendix Tab. H1).

**Inference.** At inference time, NOVUM classifies images by matching backbone features against the dense set of Gaussian features learned for each class. Each image feature is compared to the $\sim 1130$ Gaussians per class, and the class score is obtained by aggregating the maximum similarity responses across spatial locations. As shown in Fig. 7, this dense matching yields only a thin decision margin, making predictions less robust and, more importantly, difficult to interpret. In contrast, our approach drastically reduces the number of Gaussians, producing clearer decision boundaries and enabling a more transparent understanding of how predictions arise.

**Visualisation of NOVUM's feature matching.** Given NOVUM's explicit volumetric object representations, i.e., NOVs, the visualisation of its feature matching between the learned Gaussian features and image features may give some insights into which parts of the images contribute the most to the classification decision (cf. Fig. 3). Nevertheless, its classification relies on thousands of Gaussian matches do not align with semantic parts or human-understandable concepts.

## G    ADDITIONAL EXPERIMENTAL DETAILS

In this section, we provide a comprehensive review of our experimental setting to ensure reproducibility for future research. We will release the codebase and relevant checkpoints upon publication. We first describe the datasets (Sec. G.1) and interpretability metrics G.2 in our experiments. Finally, we detail training hyperparameters used in our CAVE and baselines in Section G.3.

### G.1    DATASETS

We follow the setup in NOVUM (Jesslen et al., 2024), which evaluates on three datasets Pascal3D+ (Xiang et al., 2014), OccludedP3D+ (Wang et al., 2020) and OOD-CV (Zhao et al., 2022), and extend this setting to include the large-scale ImageNet3D (Ma et al., 2024). We further evaluate concept interpretability on Pascal-Part (Chen et al., 2014).

**Pascal3D+.** The Pascal3D+ dataset  (Xiang et al., 2014) is a common benchmark for 3D object understanding. It augments 12 object class from the PASCAL VOC detection dataset (Everingham et al., 2010) with 3D annotations, and further includes images from ImageNet (Deng et al., 2009) for these categories. In total, Pascal3D+ consists of roughly 30000 images, with 8505 Pascal images and 22394 ImageNet images. Each class consists of about 3000 images on average, covering a wide range of viewpoints and intra-class shape variations.

**ImageNet3D.** The ImageNet3D dataset (Ma et al., 2024) is a large-scale dataset of natural images that extends the ImageNet dataset Deng et al. (2009) with pose annotations. It contains 200 diverse object classes, and consists of more than 86000 images in total. Only 189 object classes are currently available in ImageNet3D with sufficient annotation quality (Ma et al., 2024).

**OclcudedP3D+.** The OccludedP3D+ dataset (Wang et al., 2020) is a test benchmark that builds upon Pascal3D+ (Xiang et al., 2014) and evaluates OOD robustness against three different levels of simulated occlusion, ranging from mild (20-40%) to heavy (60-80%) occlusion. Similar to Pascal3D+ (Wang et al., 2020), this dataset contains 12 rigid object classes. Respectively, OccludedP3D+ consists of more than 10400 images for L1 (20-40%) occlusion, 10200 images for L2 (40-60%) occlusion and 9900 images for L3 (60-80%) occlusion.

**Out-of-Distribution-CV (OOD-CV).** The OOD-CV dataset (Zhao et al., 2022) is a test benchmark that introduces OOD examples for 10 different object classes. In specific, it has diverse real-world OOD nuisance factors including *shape*, *3D pose*, *texture*, *context* and *weather*. Overall, the OOD-CV consists of 2632 images with these aforementioned OOD nuisances collected from the internet and additionally 2133 test images from Pascal3D+ (Xiang et al., 2014).

**Pascal-Part.** The Pascal-Part dataset (Chen et al., 2014) introduces additional human-annotated part labels for the PASCAL VOC 2010 (Everingham et al., 2010) dataset. It covers 20 object classes, consisting of 10103 training and validation images and 9637 test images. Additionally, it provides silhouette annotations for classes without consistent parts such as *Boat*. We filter the dataset to include 12 object classes in Pascal3D+ and cross-reference them to identify test images that also appear in Pascal3D+. We use this dataset to evaluate two metrics introduced by prior works, namely *spatial localisation* and *object coverage*, that will be described formally in Section G.2 below.

### G.2    METRICS

In the following, we further describe interpretability metrics used in our evaluation.

**Model faithfulness** evaluates whether a concept truly reflects what the model uses to make a prediction (Rudin, 2019; Böhle et al., 2022). By design, inherently-interpretable models are model-faithful, while post-hoc methods, which only approximate model computations, are not. Ensuring model faithfulness is important, since it determines whether an explanation can be trusted, for example, in safety-critical and high-stake downstream applications.

**Spatial localisation** evaluates whether an explanation of a concept is spatially well-localised within a ground-truth object part (Huang et al., 2023; Behzadi-Khormouji & Oramas, 2023; Schulz et al., 2020). Localisation can be measured by Intersection-over-Union (IoU) between concept explanation and semantic part annotations, but this does not differentiate between pixels with varying attribution

strengths, treating all contributing pixels equally. Therefore, we use IoU weighted with attributions, defined similarly to Dice-Sørensen coefficient that has been previously used in XAI evaluation (He et al., 2023). Specifically, given an input image $x$, we denote the attribution of a concept $h$ at each pixel location $(i, j)$ as $A_h(i, j)$. Then, the spatial localisation score of concept $h$ with respect to part $b_k$ in image $x$ is computed as:

$$SL_{h,k}(x) = \frac{\sum_{i,j} A_h^+(i,j) \cdot \mathbb{1}_k(i,j) + \mathbb{1}_h(i,j) \cdot \mathbb{1}_k(i,j)}{\sum_{i,j} \mathbb{1}_h(i,j) + \sum_{i,j} \mathbb{1}_k(i,j)} \quad \text{(G.1)}$$

where $A_h^+(i, j)$ is the positive attribution given at pixel $(i, j)$, and score $SL_{h,k}(x) \in [0, 1]$ with higher being better. A concept $h$ with a good spatial localisation not only has high attributions in ground-truth region $b_k$ but also covers it well. As described, this metric requires human-annotated parts and will be evaluated on Pascal-Part (Chen et al., 2014). For each image and each concept, we identify the ground-truth part that the concept is most aligned with and compute the spatial localisation score with respect to that part. The spatial localisation for concept $h$ in image $x$ is thus defined as:

$$SL_h(x) = \max_k SL_{h,k}(x)$$

These per-image scores for a concept $h$ are then averaged across all test images of the same class to obtain a concept-level localisation score. These concept-level scores are then averaged per class. The dataset-level score is then obtained by averaging these per-class scores, and reported in Tab. 1.

**Object coverage** measures the extent of concept comprehensiveness, i.e., how well discovered concepts cover the object (Zhu et al., 2025). This is especially crucial in case of occlusion, where a well-covered set of concepts improves robustness as it can still identify other visible, unoccluded parts. We first normalise the attributions such that the total attribution across all concepts for a given input image $x$ sums to 1. This means that if the union of all concepts perfectly collides with the ground-truth object mask, the coverage score is 1. This also ensures a fair and consistent comparison across methods. The normalisation is computed as follows:

$$\tilde{A}_h^+(i,j) = \frac{A_h^+(i,j)}{\sum_{h'} \sum_{i,j} A_{h'}^+(i,j)} \quad \text{(G.2)}$$

For an input image $x$, we define the object coverage score as

$$\text{Cov}_{\text{object}}(x) = \sum_{i,j} \sum_h \tilde{A}_h^+(i,j) \cdot \mathbb{1}_{\text{bbox/object mask}}(i,j) \quad \text{(G.3)}$$

Naturally, $\text{Cov}_{\text{object}}(x) \in [0, 1]$ for both cases, with a higher score means better coverage, and thus a more comprehensive set of concepts. This metric also requires ground-truth object masks, and is here evaluated on Pascal-Part (Chen et al., 2014). Each test image yields an object-coverage score. We average these scores over all images of a class to obtain a class-level value, and then average the class-level values across all classes to obtain the dataset-level score reported in Tab. 1.

**3D consistency (3D-C).** We propose the 3D-C metric to complement existing evaluation metrics that relies heavily on pre-defined part labels. We describe this metric in detail in Section D. Our 3D-C allows for evaluation across 4 datasets: Pascal3D+ (Xiang et al., 2014), ImageNet3D (Ma et al., 2024), OccludedP3D+ (Wang et al., 2020), and OOD-CV (Zhao et al., 2022).

### G.3 IMPLEMENTATION DETAILS

Our experimental setting requires training our CAVE as well as baseline methods on two datasets: Pascal3D+ (Xiang et al., 2014) and ImageNet3D (Ma et al., 2024). Our evaluation on the aforementioned metrics in Section G.2 is then done on five datasets: Pascal3D+ (Xiang et al., 2014), ImageNet3D (Ma et al., 2024), and two OOD datasets Occluded-P3D+ (Wang et al., 2020) and OOD-CV (Zhao et al., 2022), and part-annotated Pascal-Part (Chen et al., 2014).

In the following, we detail the training setup for our CAVE as well as for the baselines. For a fair comparision, our CAVE and all baselines use ResNet-50 backbone. In NOVUM (Jesslen et al., 2024), the ResNet-50 backbone was extended with two upsampling layers by concatenation for NOV alignment. Our CAVE adopts the same architecture.

| Training Hyperparameter | Value |
|---|---|
| Input resolution | $640 \times 800$ |
| Backbone | ResNet-50 |
| Training batch size | 38 |
| Total epochs | 200 |
| Learning rate | $1 \times 10^{-4}$ |
| Learning rate decay | $\times 0.2$ every 10 epochs |
| Momentum update of Gaussian features | 0.9 |
| Weight decay | $1 \times 10^{-4}$ |
| Gradient accumulation | 10 steps |
| Temperature $T$ | 0.07 |
| Noise weight | 0.005 |
| Distance threshold | 48 |
| Number of background features for momentum update | 5 |

Table G1: Training hyperparameters used in CAVE (Ours) and NOVUM (Jesslen et al., 2024).

**CAVE (Ours).** Our model training, similar to NOVUM (Jesslen et al., 2024), uses input images of size $640 \times 800$. The ResNet-50 backbone produces the output feature map $F_x$ by downsampling the input spatial resolution by a factor of 8. Training on Pascal3D+ takes approximately 24 hours, while training on ImageNet3D requires about 72 hours on a single NVIDIA H100 GPU. As mentioned in section 4, we use ellipsoid as volumetric surface of Gaussian features, computing its vertices and faces based on parametric sampling where a mesh is returned with roughly 1000 vertices. We summarise our training hyperparamters in the Tab. G1. Once the model is trained, we filter out Gaussian features with low visibility ($< 0.1$ of training data), and apply K-Means clustering for concept extraction with 20 clusters per class (i.e., $D = 20$ following notation in Sec. 4).

**Baselines**. For baseline methods, we follow the original implementation, training details, and concept visualisation provided by the authors:

1. For NOVUM (Jesslen et al., 2024), the training follows the same hyperparameters in Tab. G1. We obtained the model checkpoint for Pascal3D+ from the NOVUM's codebase (Jesslen et al., 2024), and trained NOVUM from scratch for ImageNet3D.

2. For post-hoc methods CRAFT (Fel et al., 2023b), ICE (Zhang et al., 2021), and PCX Dreyer et al. (2024), we used the exact setting provided in their codebases. We set the number of class-wise concepts to 20. For MCD-SSC, we used completeness threshold of 0.9, which determines the extent to which the discovered set of concepts jointly cover model prediction. The concepts were extracted from NOVUM's activations for each method.

3. For LF-CBM (Oikarinen et al., 2023), we followed the codebase instructions to generate a concept set for each dataset from GPT-3 model, resulting in 441 concepts for Pascal3D+ and 710 concepts for ImageNet3D. We trained a ResNet-50 label-free CBM using their hyperparameters: input resolution $224 \times 224$, learning rate 0.1, and batch size 512, CLIP model ViT-B/16, and interpretability cutoff 0.45. Unlike post-hoc approaches, where the number of concepts is fixed at 20, label-free CBM learns this number dynamically. For Pascal3D+, we observed 297 non-zero weights out of 5292, yielding roughly 5% sparsity. For ImageNet3D, we observed 4058 non-zero weights out of 128709, yiedling 3% sparsity.

4. For prototypical networks such as ProtoPNet (Chen et al., 2019) and TesNet (Wang et al., 2021), we used the same training hyperparameters from their codebases: input resolution $224 \times 224$, log as prototype activation function, with regular add-on layers, a training batch size of 80, and a training push batch size of 75. The learning rates for features, add-on layers, and prototype vectors are set to $1e-4, 3e-3, 3e-3$ respectively, with a joint learning rate step size of 5. The projection (push) step was performed every 10 epochs. The network was trained for 200 epochs without a warm-up period or extensive data augmentation to ensure fairness across baselines.

5. For PIP-Net (Nauta et al., 2023a), we trained the network using the hyperparameter settings from its codebase, with input resolution $224 \times 224$. During pre-training, we used batch size

of 128 and trained for 10 epochs. For the main training phase, we used a batch size of 64 and trained for 200 epochs. The learning rate was set to 0.05 with no weight decay.

We also find that post-hoc methods such as Fel et al. (2023b) threshold their attributions to reduce noise in their concept visualisations. For a fair qualitative comparision, we apply the same $90\%$-th quantile threshold across baselines that requires it for concept visualisation. Our method CAVE does not require such thresholding, as its attributions are correctly propagated for each concept with our NOV-aware LRP to input pixels.

# H  ABLATION: SHAPE OF NEURAL OBJECT VOLUMES

For fair comparision, we use full 3D supervision, i.e., with ground-truth 3D poses, for both NOVUM (Jesslen et al., 2024) and CAVE (Ours) in this ablation across different shapes of the neural volumes, including simpler shapes in NOVUM such as cuboids and spheres, and more expressive shapes such as ellipsoid and prototype CADs.

At first glance, CAVE is able to match or even slightly exceeding NOVUM's performance with roughly $D = 20$ class-wise concepts, yielding 98% sparser representation compared to NOVUM's 1130 Gaussians per class (cf. Fig H1).

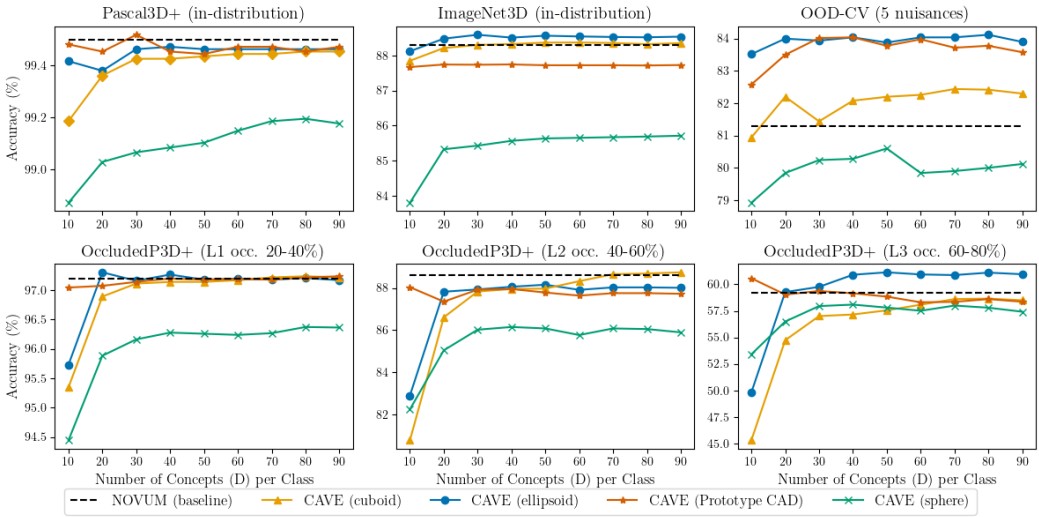

Figure H1: **CAVE with ellipsoid NOVs matches or slightly exceeding NOVUM** in most cases with only $D = 20$ concepts per class, compared to NOVUM with roughly 1130 Gaussians per class. Compared to other shapes, ellipsoid NOVs give an advantage in large-scale dataset ImageNet3D, heavy occlusion in OccludedP3D+ and OOD-CV with challenging nuisances.

| Dataset | P3D+ | ImageNet3D | | Occluded P3D+ | | OOD-CV |
|---|---|---|---|---|---|---|
| Occlusion | L0 (0%) | L0 (0%) | L1 (20 − 40%) | L2 (40 − 60%) | L3 (60 − 80%) | 5 Nuisances |
| Sphere | 99.2 | 85.8 | 96.7 | 86.3 | 57.9 | 81.6 |
| Ellipsoid | 99.4 | **88.6** | 97.0 | 87.5 | 58.9 | 82.4 |
| Prototype CAD | **99.5** | 87.7 | **97.3** | 88.1 | 58.6 | **83.3** |
| Cuboid (default) | **99.5** | 88.3 | 97.2 | **88.6** | **59.2** | 81.3 |

Table H1: **Accuracy (%, ↑) of 3D-aware classifier NOVUM** on two in-distribution dataets Pascal3D+ (P3D+) and ImageNet3D, and on two OOD datasets Occluded-P3D+ and OOD-CV, evaluated across different neural volume shapes. **Best** and second best are highlighted.

| Dataset | P3D+ | ImageNet3D | | Occluded P3D+ | | OOD-CV |
|---|---|---|---|---|---|---|
| Occlusion | L0 (0%) | L0 (0%) | L1 (20 − 40%) | L2 (40 − 60%) | L3 (60 − 80%) | 5 Nuisances |
| Cuboid | 99.4 | 88.2 | 96.9 | 86.6 | 54.7 | 82.2 |
| Sphere | 99.0 | 85.3 | 95.9 | 85.0 | 56.5 | 79.8 |
| Prototype CAD | **99.5** | 87.7 | 97.1 | 87.4 | 59.0 | 83.5 |
| Ellipsoid (default) | 99.4 | **88.5** | **97.3** | **87.8** | **59.3** | **84.0** |

Table H2: **Accuracy (%, ↑) of CAVE (ours)** with $D = 20$ class-wise concepts on two in-distribution dataets Pascal3D+ (P3D+) and ImageNet3D, and on two OOD datasets Occluded-P3D+ and OOD-CV, evaluated across different neural volume shapes. **Best** and second best are highlighted.

| Shape | IoU ↑ | Spatial Localisation ↑ | Coverage ↑ | | 3D-C ↑ |
| | | | ∪ Parts | Object | |
|---|---|---|---|---|---|
| Cuboid | 0.150 | 0.264 | 0.368 | 0.873 | 0.402 |
| Sphere | 0.148 | 0.263 | 0.367 | 0.870 | 0.394 |
| Ellipsoid | 0.156 | 0.277 | 0.368 | 0.865 | 0.417 |
| Prototype CAD | **0.169** | **0.305** | **0.378** | **0.874** | **0.426** |

Table H3: **Interpretability evaluation of CAVE across different neural volume shapes** with $D = 20$ concepts per object category on Pascal-Part with different benchmark metrics. In specific, *Part IoU*, *Spatial Localisation* (weighted IoU with attributions), *Global Coverage* assessing both object coverage and union of annotated parts, and our proposed metric *3D Consistency* (3D-C) to assess spatial consistency of concepts across images. **Best** and second best are highlighted.

In terms of performance, while there is no consistent gain in NOVUM with a particular NOV shape (cf. Tab. H1), CAVE with more expressive shapes like ellipsoid and prototype CAD performs relatively better in heavy occlusion (OccludedP3D+ L3), and OOD-CV, improving roughly 2-3% point in OOD accuracy (cf. Tab H2). We hypothesise that since Gaussians are densely populated in NOVUM which are then used for classification, having a more accurate shape approximation has limited effects. Whereas for CAVE, our concept-based NOV representation is approximately 98% **sparser** than NOVUM's 1130 Gaussians per class, and thus the shape matters more in our design.

Additionally, we evaluate how NOV shape influences CAVE's concept interpretability (cf. Tab. H3). Prototype CADs score highest across all metrics, consistent with their closer match to the underlying object geometry. Ellipsoids are only slightly worse yet require no CAD assets and are faster to fit. Given this empirical trade-off between OOD accuracy and interpretability, we **adopt ellipsoid NOVs** in CAVE, instead of using cuboids as in prior work (Jesslen et al., 2024).

Taken together, these results show that CAVE delivers strong OOD accuracy with a highly sparse and compact concept-based representation across shapes, with more expressive shapes such as ellipsoids and prototype CADs are slightly better. Among the tested shapes, ellipsoid NOVs offer the best balance between interpretability and robustness, making them a scalable choice for CAVE, where 3D-aware classification is both inherently interpretable and robust to distribution shifts.

## I    ABLATION: ATTRIBUTING CONCEPTS WITH NOV-AWARE LRP

We provide here a more comprehensive quantitative and qualitative evaluation of our proposed NOV-aware LRP, described in Sec. 4.1 & Sec. C, against common attribution methods for attributing concepts in 3D-aware architectures like CAVE. These include Vanilla LRP  (Bach et al., 2015), Guided Backpropagation (Springenberg et al., 2015), Smooth Gradients (Smilkov et al., 2017), GradCAM (Selvaraju et al., 2017), and Integrated Gradients (Sundararajan et al., 2017).

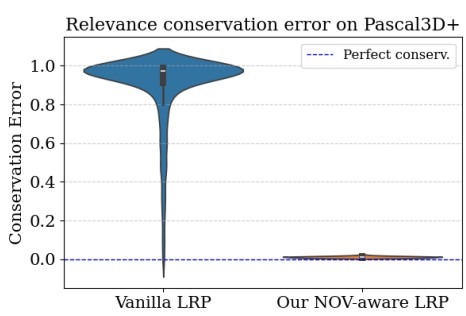

In a direct comparison to vanilla LRP (Bach et al., 2015) that we base our method on, we show that our NOV-aware LRP enables faithful attribution of concepts through the NOVs back to the input pixels, achieving near-perfect relevance conservation. In contrast, vanilla LRP unfaithfully leaks relevances, where relevance is either lost or incorrectly redistributed during propagation (cf. Fig. I1). Qualitatively in Fig. I2, we observe spurious relevances visibly appear in background regions or occlusion masks in vanilla LRP. On the other hand, our NOV-aware LRP consistently produces sharp, spatially coherent attributions that remain well-aligned with the underlying object structure. This demonstrates not only the technical benefits of adapting LRP to 3D-aware classifiers with volumetric representations, but also the

Figure I1: **Violin plot on relevance conservation** of Ours vs. Vanilla Layer-wise Relevance Propagation (LRP), where 0 indicates perfect conservation. Our NOV-aware LRP achieves near-perfect conservation compared to Vanilla LRP.

practical interpretability gains, where concepts are now more reliably traced back to the input image. Taken together, our results demonstrate the strong effect of NOV-aware LRP in improving the fidelity of attribution, i.e., by ensuring that relevance is conserved properly during propagation.

| | Attribution Methods | Localise. ↑ | Coverage ↑ | 3D Consistency ↑ | |
| --- | --- | --- | --- | --- | --- |
| | | Pascal-Part | | Pascal3D+ | OccludedP3D+ |
| CAVE | Vanilla LRP ($\epsilon$−rule) | 0.26 | 0.73 | 0.29 | 0.16 |
| | Guided Backprop. | 0.25 | 0.61 | 0.32 | 0.19 |
| | SmoothGrad | 0.23 | 0.69 | 0.31 | 0.16 |
| | Grad-CAM | 0.12 | 0.47 | 0.27 | 0.13 |
| | Integrated Gradients | 0.19 | 0.57 | 0.35 | 0.19 |
| | NOV-aware LRP (Ours) | **0.28** | **0.80** | **0.40** | **0.23** |

Table I1: **Quantitative comparison of different attribution methods for CAVE**, evaluated on *spatial localisation*, *object coverage*, and *3D consistency*

We further study the effectiveness of our NOV-aware LRP compared to common attribution methods (cf. Tab. I1). Ours produces concepts with higher spatial localisation, object coverage (i.e., concept comprehensiveness), and 3D consistent across in-distribution Pascal3D+ (Xiang et al., 2014) and OOD OccludedP3D+ (Wang et al., 2020). This means that our LRP reformulation allows us to generate not only highly localised but also semantically comprehensive visualisation of concepts, covering the object well rather than focusing on a few discriminative fragments (cf. Fig. I2). Furthermore, NOV-aware LRP maintains strong consistency across 3D views, outperforming all baselines on Pascal3D+ (0.40 vs. 0.35 from Integrated Gradients) and OccludedP3D+ (0.23 vs. 0.19 from Guided Backpropagation and Integrated Gradients). The gains are particularly pronounced under occlusion, where reliable attribution is more challenging.

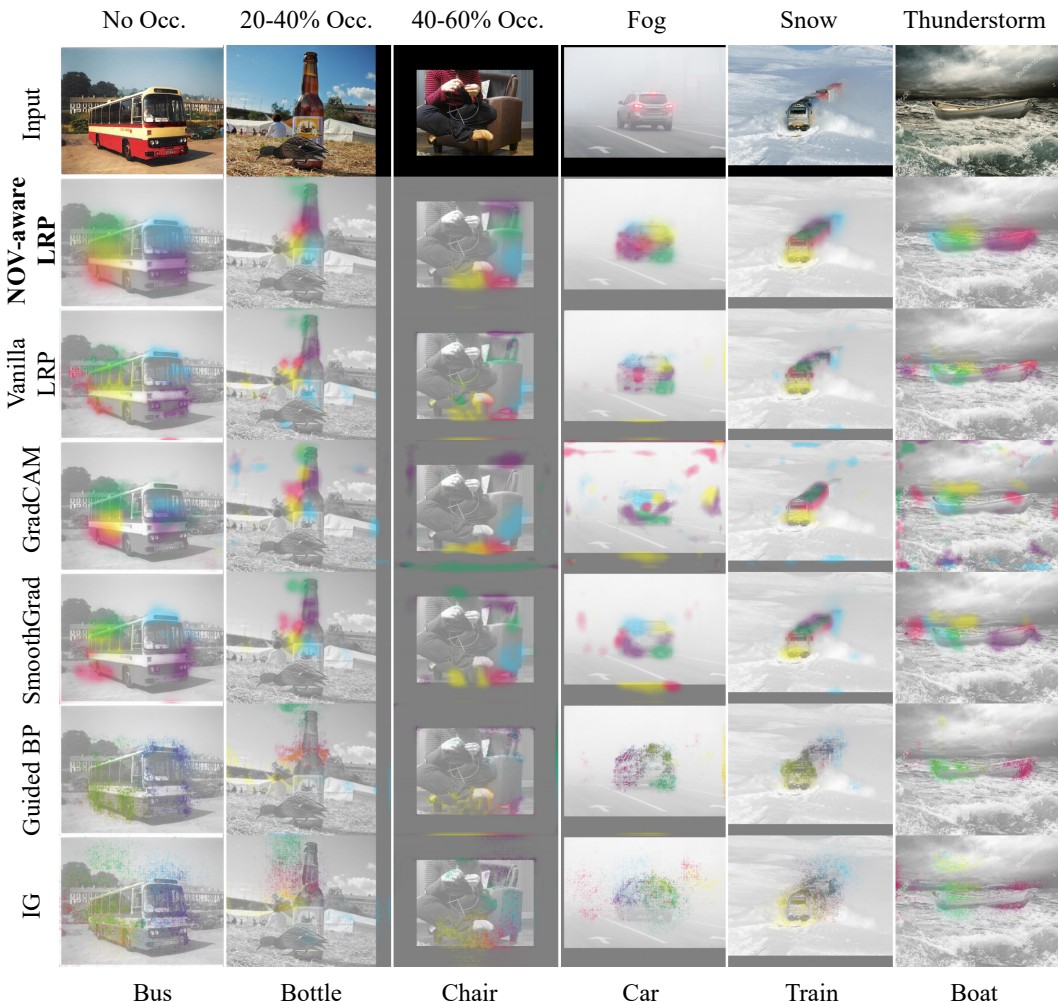

Figure I2: **Our NOV-aware LRP (second row) yields better-localised, more stable visualisation of top-5 concepts across both in-distribution and OOD settings**, while vanilla LRP and other common attribution methods show scattered explanations on the image background or occlusion masks. From left → right, the first input image comes from in-distribution Pascal3D+ (Xiang et al., 2014), the next two from OccludedP3D+ (Wang et al., 2020) with different levels of occlusion, and the last three from OOD-CV (Zhao et al., 2022).

## J    ABLATION: CONCEPT EXTRACTION ON NEURAL OBJECT VOLUMES (NOVS)

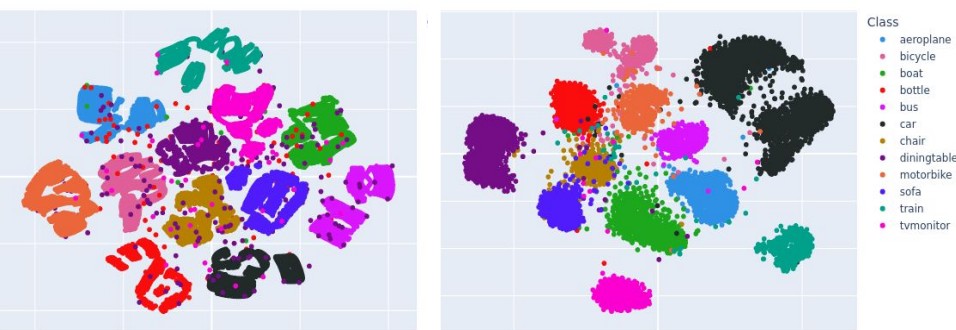

Figure J1: **Representation spaces** derived from neural object volumes $\mathcal{G}$ (left) and activations of features (right) of all classes in Pascal3D+.

An advantage of an image classifier with NOVs is that it enhances the separation of the representation space, ensuring that Gaussian features from different classes remain well-distinguished (Jesslen et al., 2024). Naturally, this is also the case for activations of features within the same classifier (cf. Fig. J1), which then leads us to examine where concept extraction is most effective. Our CAVE framework extracts class-wise concepts directly from neural object volumes $\mathcal{G}$, whereas traditional methods rely on activations from training features. We explore whether $\mathcal{G}$ provides a more meaningful concept space by following three key steps:

1. extracting concepts using low-rank approximations (e.g., NMF). For a fair comparison, we use NMF as our extraction method because it is known to effectively extract meaningful and interpretable concepts from activations (Fel et al., 2024). Note that NMF is not guaranteed to be the optimal choice for neural volumes $\mathcal{G}$,

2. projecting activations into the concept space, and

3. train a K-means clustering model on training image activations and evaluate it on test activations across different occlusion levels for concept separation.

A well-structured concept space should produce well-separated and compact clusters, which can be evaluated using the Silhouette Score and Davies-Bouldin Index (DBI). The Silhouette Score ranges from $[-1, 1]$, where 1 indicates optimal separation and values near 0 suggest overlapping clusters. In contrast, the Davies-Bouldin Index measures the ratio of intra-cluster dispersion to inter-cluster separation, with lower values indicating better-separated, more compact clusters. We also show 2D low-dimensional representations comparing the concept spaces of neural volumes and training activations across different classes in Fig. J2.

| Occlusion | 0 | $[20, 40]$ | $[40, 60]$ | $[60, 80]$ |
|---|---|---|---|---|
| *NOVs* $\mathcal{G}$ | **0.343** | **0.333** | **0.335** | **0.338** |
| *Activations* | 0.034 | 0.040 | 0.037 | 0.033 |

Table J1: Silhouette scores ($\uparrow$) averaged across all classes in Pascal3D+, comparing concept spaces derived from neural object volumes $\mathcal{G}$ and traditional activations. Higher scores indicate more compact clusters, with values close to 1 representing well-structured concept spaces. Results are reported across different occlusion levels in %.

As we extract concepts from NOVs, there are different established approaches to obtain the decomposition $\mathcal{G}_y \approx \mathcal{W}_y \mathcal{H}_y^T$, specifically K-Means, Principal Component Analysis (PCA) and Non-negative Matrix Factorisation (NMF). We aim to strategically select the concept extraction method that encourages disentangled, part-based concepts while maintaining competitive performance and faithfulness. To do so, we conduct a small ablation and compare in terms of three key metrics:

| Occlusion | 0 | [20, 40] | [40, 60] | [60, 80] |
|---|---|---|---|---|
| *NOVs $\mathcal{G}$* | **1.154** | **1.155** | **1.162** | **1.137** |
| *Activations* | 2.365 | 2.348 | 2.306 | 2.318 |

Table J2: Davies-Bouldin Index ($\downarrow$) averaged across all classes in Pascal3D+, comparing concept spaces derived from neural object volumes $\mathcal{G}$ and traditional activations. Lower score means better cluster separation.

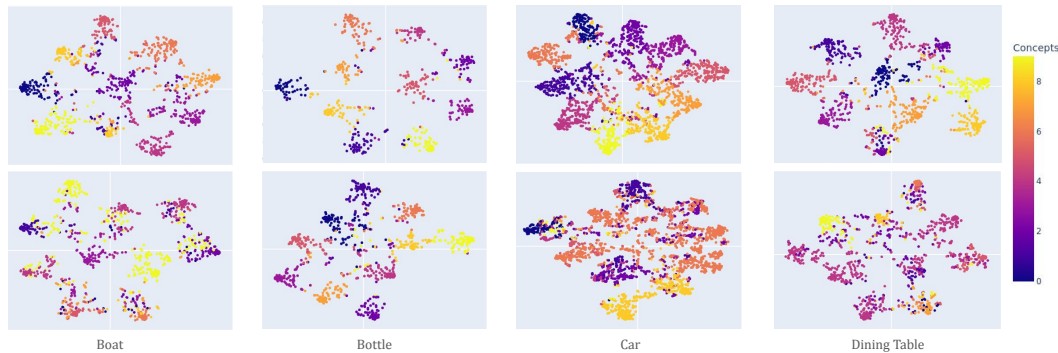

Figure J2: **t-SNE projection of concept embeddings in neural volume space (top) vs. activation space (bottom) for four classes in Pascal3D+**. Concepts in neural volume space are more well-separated and compact compared to those extracted from activation space.

(i) *accuracy*, (ii) *sparsity*, and (iii) *feature distribution distance (FDD)*. Here, *sparsity* is measured following the approach in Fel et al. (2024), which quantifies how sparse the weight matrix $\mathcal{W}_y^*$ is. *FDD* measures the divergence between the original distribution of $\mathcal{G}_y$ and its new representation $\mathcal{H}_y$, thereby quantifying how *faithful* the latter preserves the former. After evaluating these three metrics, we select K-Means clustering as concept identification in CAVE.

| | *Accuracy* (%, $\uparrow$) | *Sparsity* ($\uparrow$) | *FDD* ($\downarrow$) |
|---|---|---|---|
| K-Means | 99.36 | **0.95** | **0.32** |
| PCA | **99.39** | 0.00 | 0.39 |
| NMF | 99.25 | 0.63 | 0.83 |
| NOVUM | 99.5 | – | – |

Table J3: **Ablation on concept extraction methods.** The concept extraction methods are applied on the neural volumes $\mathcal{G}_y \in \mathcal{G}$ in image classifier NOVUM with $D = 20$ concepts. Each result is averaged across all classes from Pascal3D+.

## K    DISCUSSION OF FAILURE CASES

While CAVE achieves significant improvement in both robustness and interpretability, some inherent limitations of 3D-aware classification remain.

First, CAVE is trained on object-centric images and orients its ellipsoid NOVs to the *primary object* to learn Gaussian features for concept dictionary extraction. This in turn allows for spatially consistent and localised explanations. In multi-object classification tasks where multiple instances of the same objects are present, CAVE surprisingly can still identify concepts across objects, even though its training is not optimised for multi-object scenes. In some cases, however, CAVE identifies concepts on the central object and abstains from others (cf. Fig. K1). While this behavior is consistent with object-centric learning of 3D-aware classifiers, improving CAVE further in multi-object scenes would be an interesting future direction.

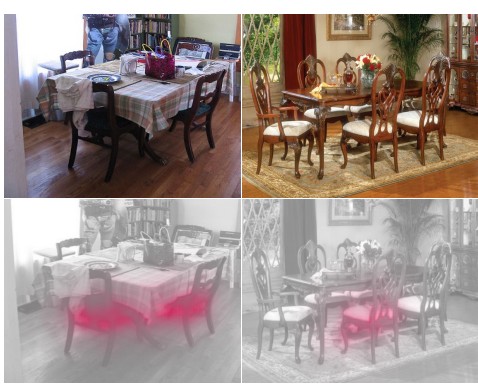

Figure K1: **CAVE** identifies concept across multiple objects (left) but occasionally misses (right).

Second, we study cases where CAVE fails to find spatially consistent concepts in most test images. For example, in class `Boat` of ImageNet3D (Ma et al., 2024), the same concept only localises on the stern in some images, while to both bow and stern in others. This can be attributed to the fact that boats do not have a consistent set of parts across subcategories, as also observed in the Pascal-Part dataset (Chen et al., 2014), as well as because of object symmetries. In fact, when the two ends are visually similar, the concept generalises to *hull ends*, rather than only on *stern* (cf. Fig. K2).

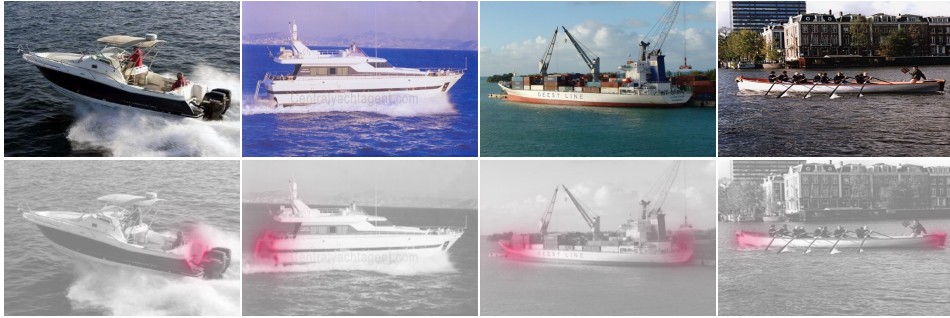

Figure K2: For the same concept in the `Boat` class of ImageNet3D, **CAVE** detects only *stern* in some cases (first two columns), while both hull ends in others (last two columns).

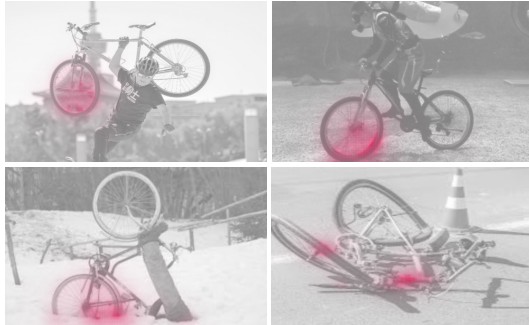

Figure K3: Under OOD factors, **CAVE** correctly detects *front wheel* (above) but can occasionally activates back wheel or other parts (below) under strong perspective shift or object deformation.

We further investigated how OOD nuisances affect concept consistency. In in-distribution settings such as Pascal3D+ (Xiang et al., 2014) and ImageNet3D (Ma et al., 2024), CAVE's 3D-C scores of Bicycle are roughly $0.24$. Under distribution shift in OOD-CV (Zhao et al., 2022), the score drops to about $0.16$, albeit still substantially higher than all baselines, which fail to detect any consistent concepts. We investigate examples where CAVE shows inconsistency, and find that for example concept such as *front wheel* activates also on back wheel under strong perspective shift, or when the bicycle is severely deformed (cf. Fig. K3).

## L    ADDITIONAL QUALITATIVE EXAMPLES

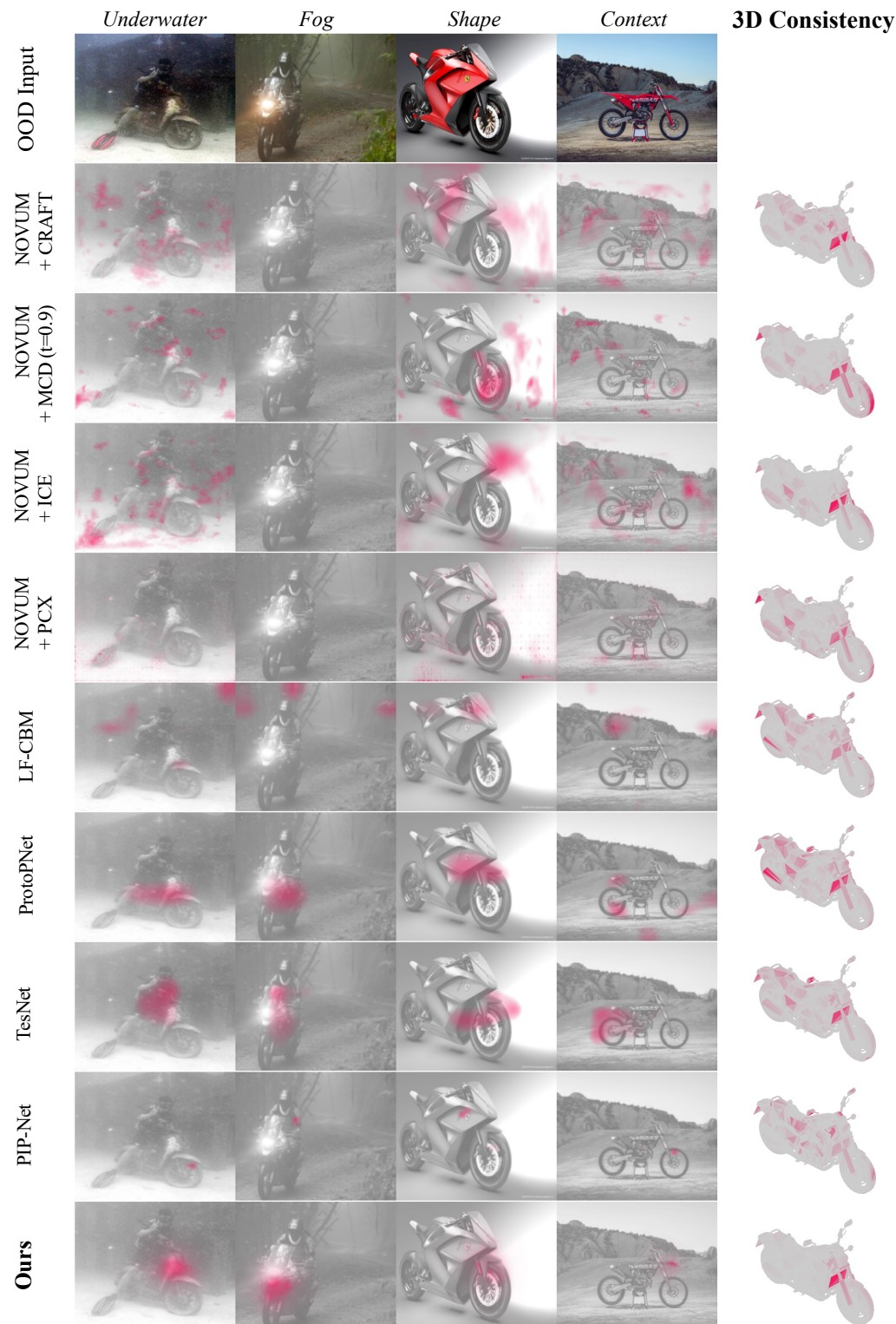

Figure L1: **CAVE (Ours) produces more spatially consistent and localised explanations for OOD images of class Motorbike compared to all baselines.**

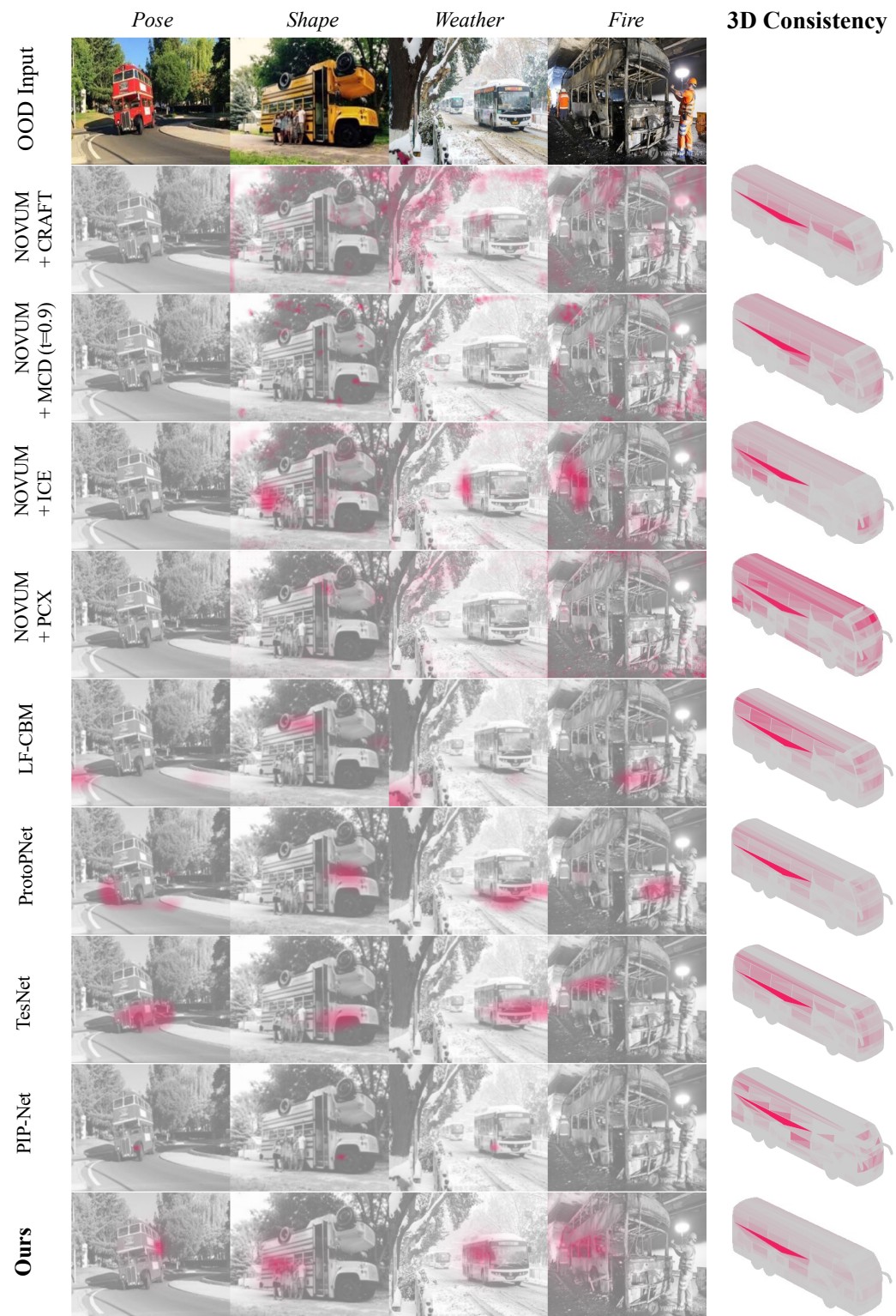

Figure L2: **CAVE (Ours) produces more spatially consistent and localised explanations for OOD images of class Bus compared to all baselines.**

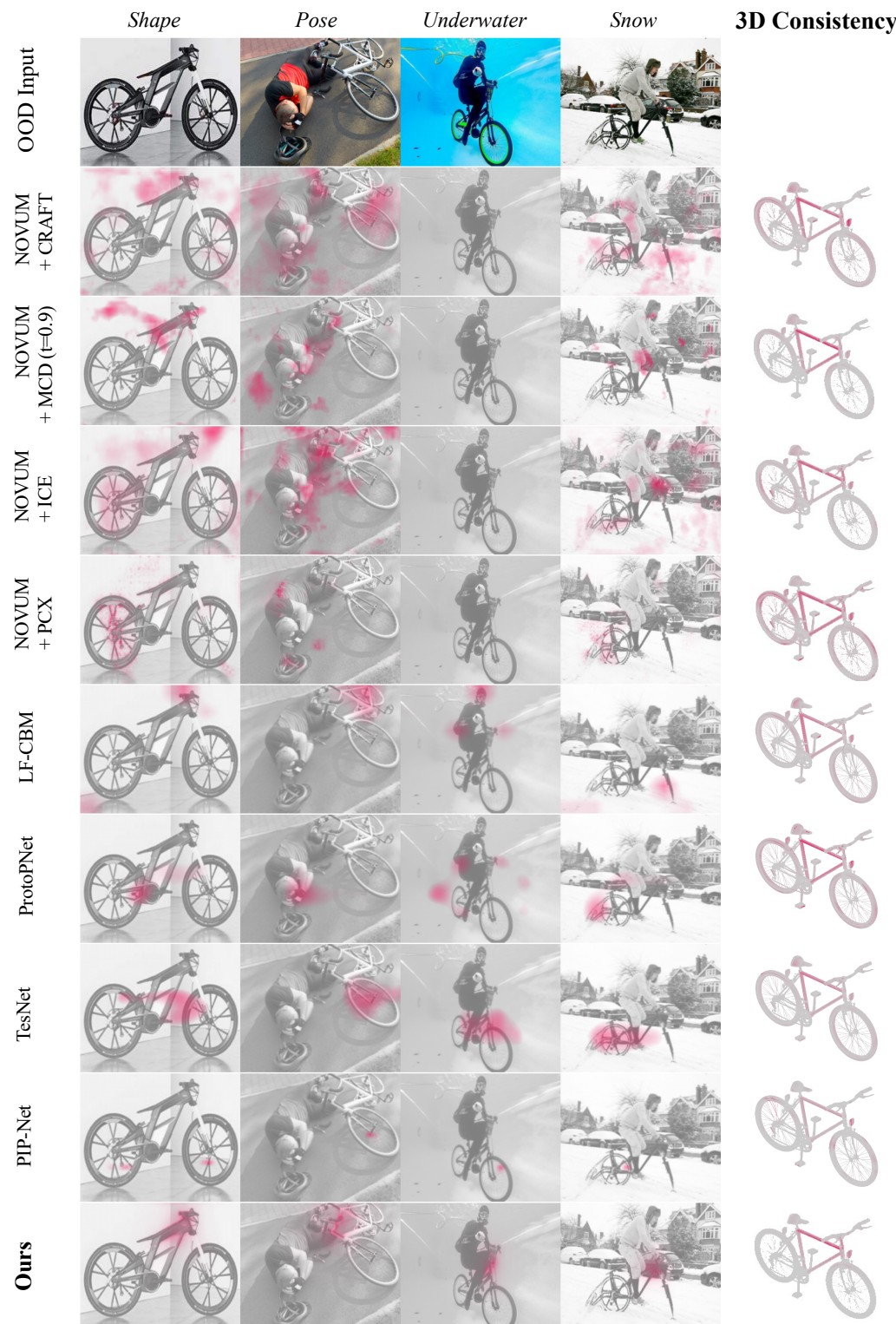

Figure L3: **CAVE (Ours) produces more spatially consistent and localised explanations for OOD images of class Bicycle compared to all baselines.**

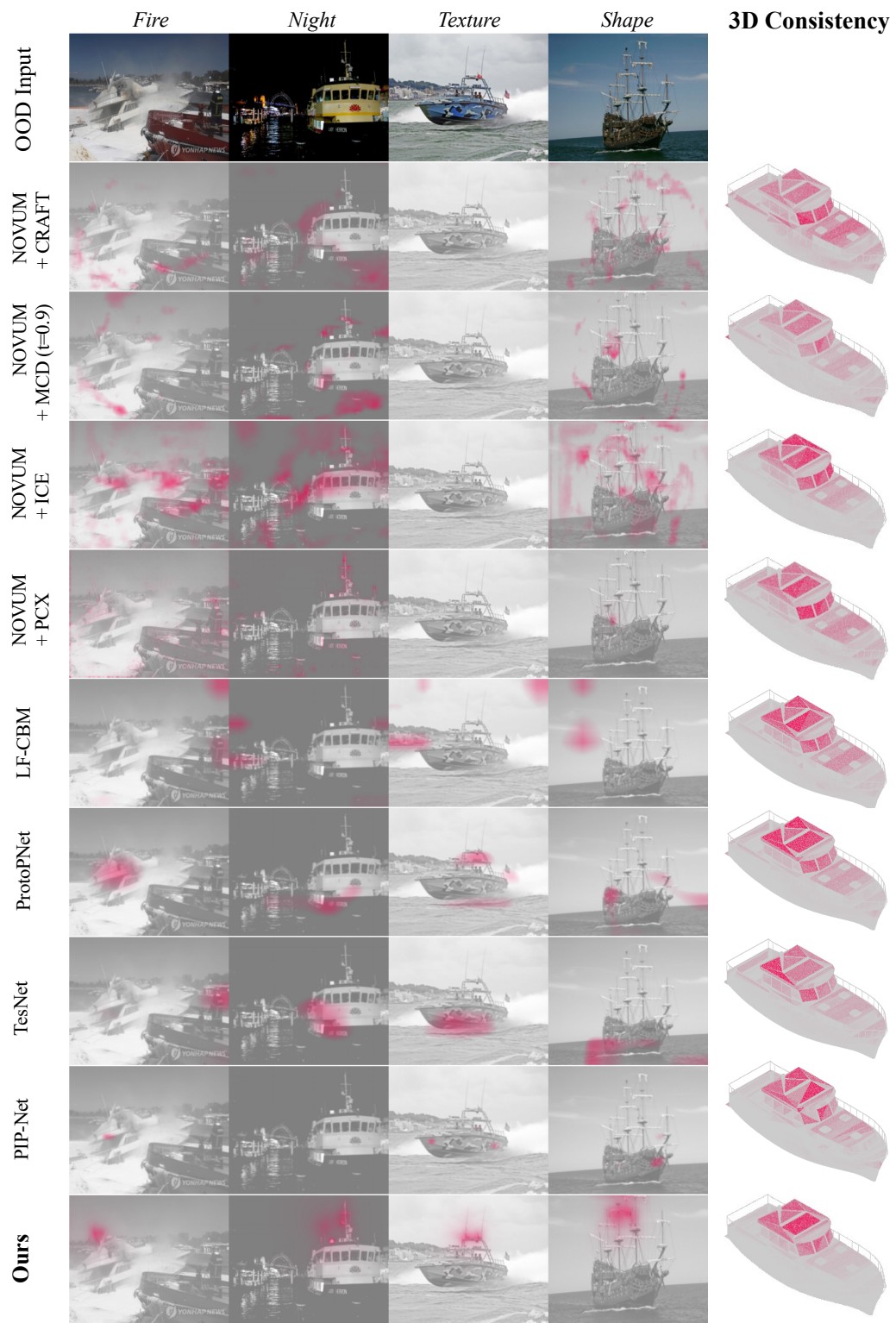

Figure L4: **CAVE (Ours) produces more spatially consistent and localised explanations for OOD images of class Boat compared to all baselines.**

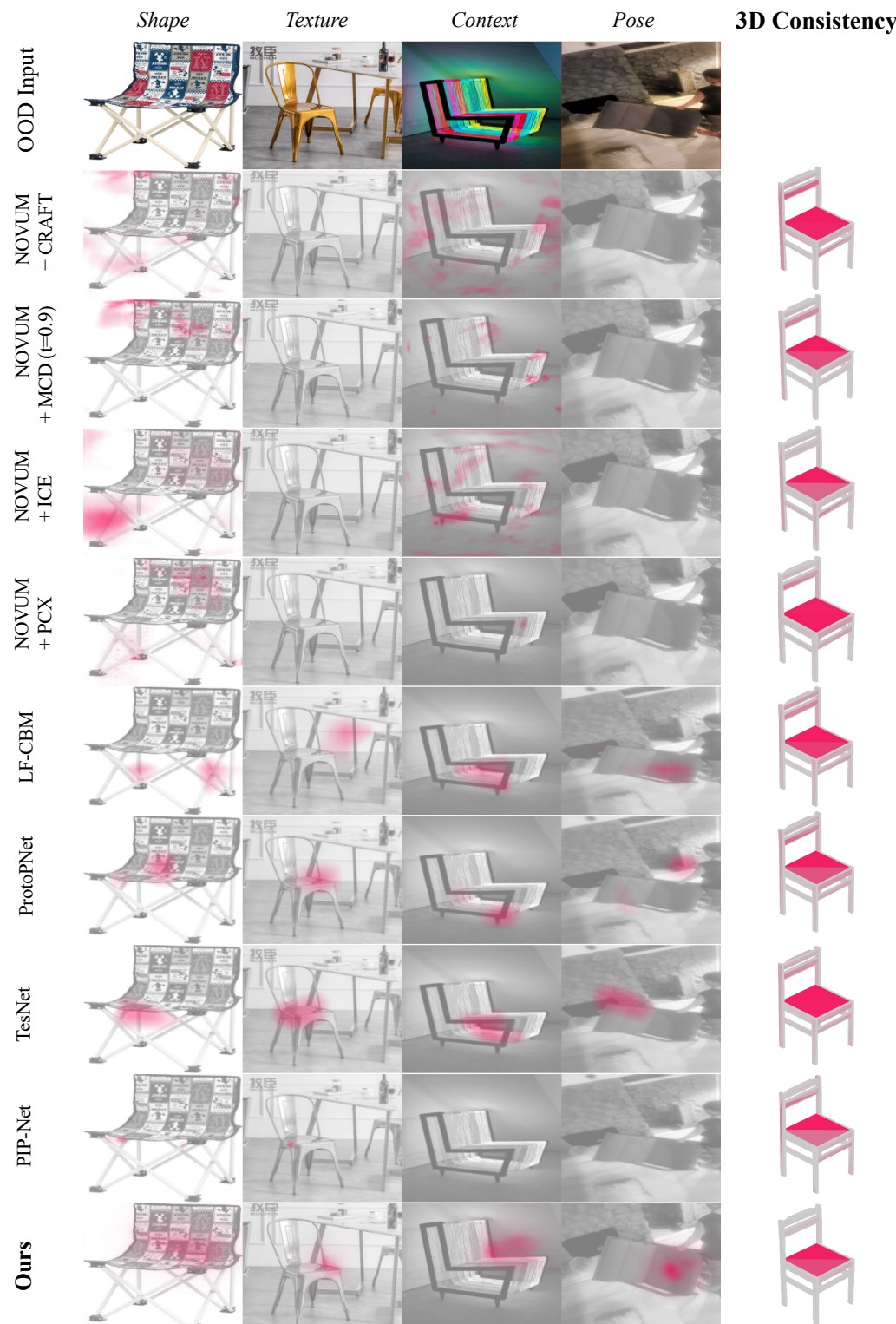

Figure L5: **CAVE (Ours) produces more spatially consistent and localised explanations for OOD images of class Chair compared to all baselines.**

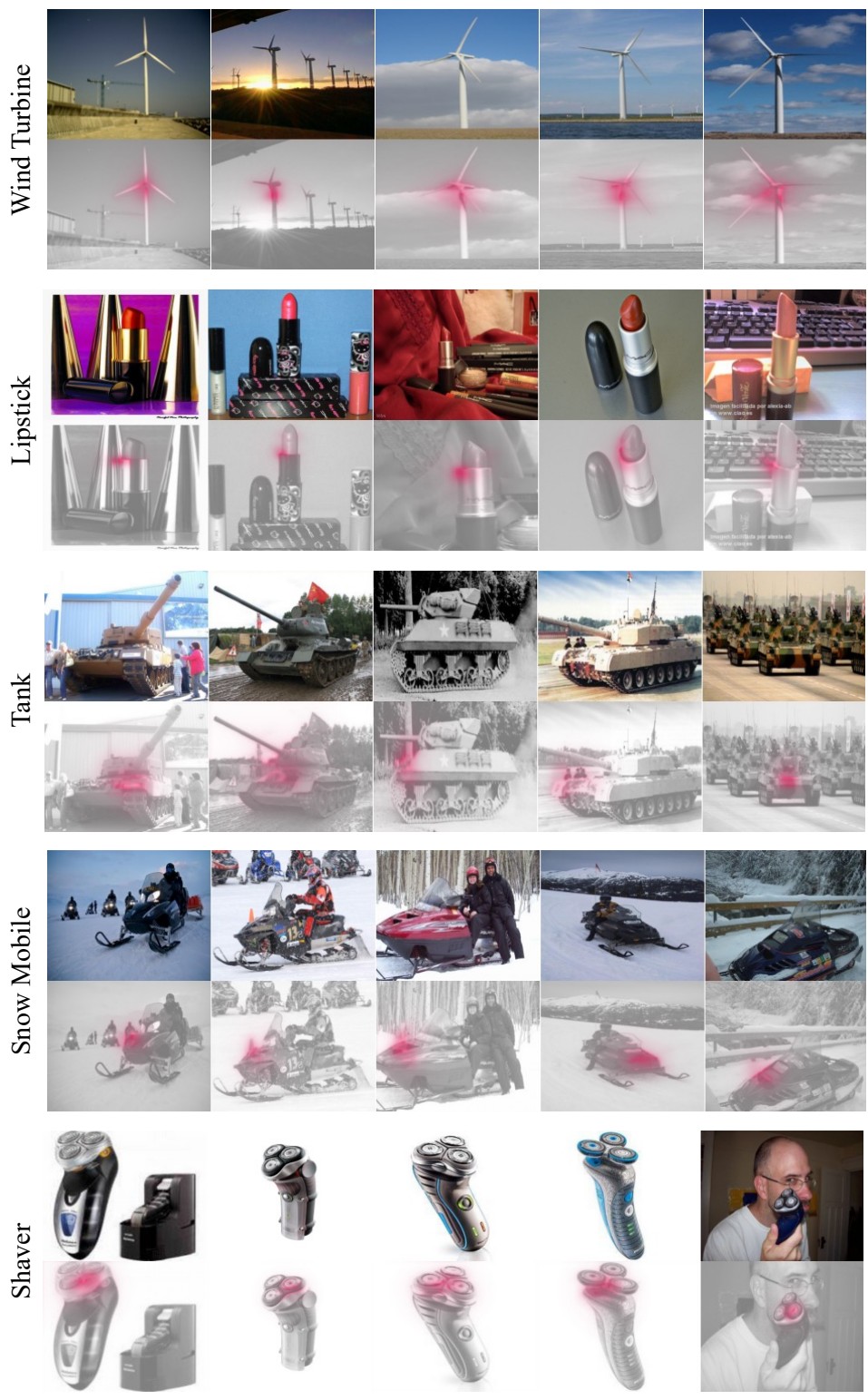

Figure L6: **CAVE (Ours) detects consistent concepts across different ImageNet3D classes**. For each class, the concept is *sampled randomly*, and the images associated with that concept are also *sampled randomly*, both with seed 42.

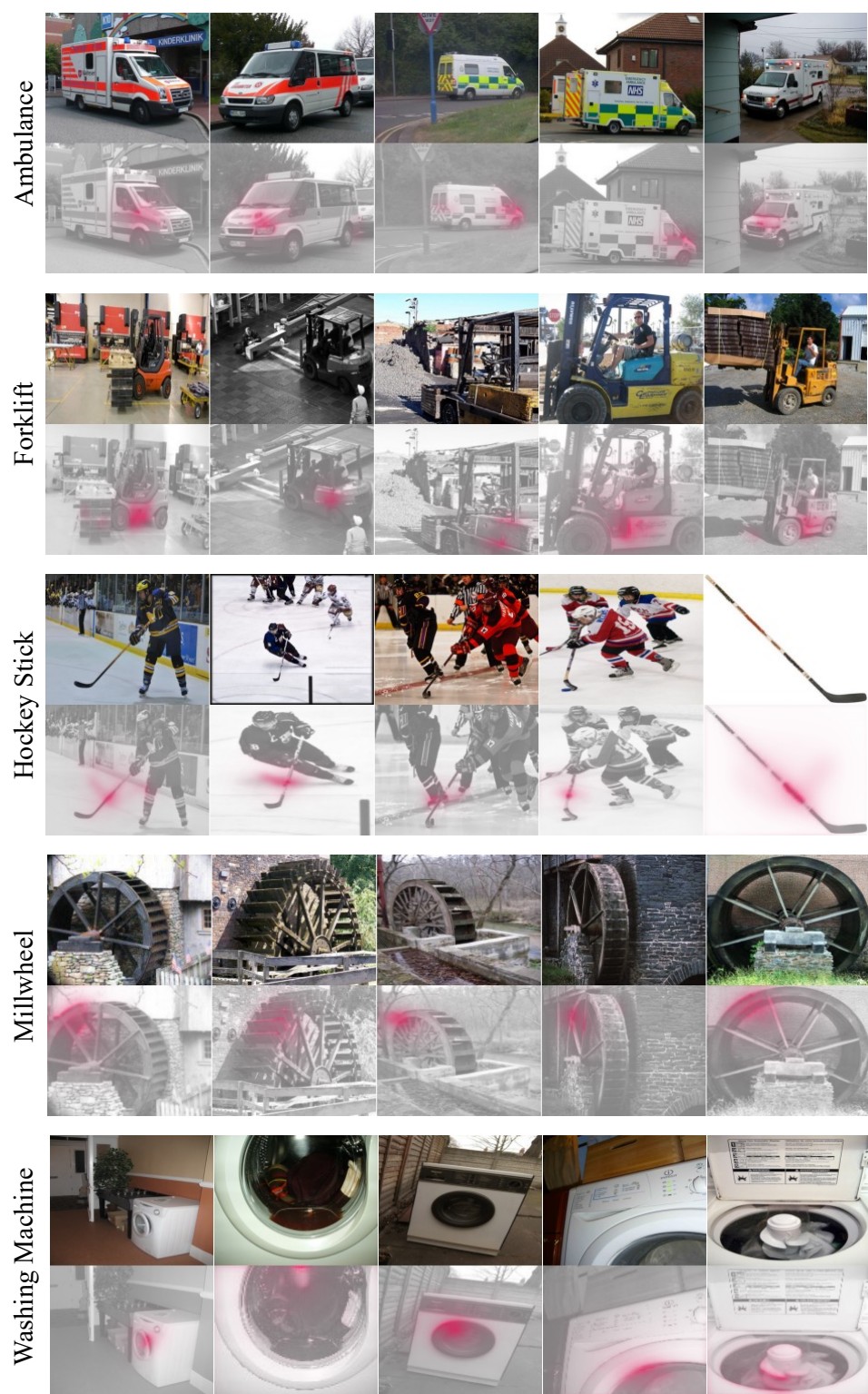

Figure L7: **CAVE (Ours) detects consistent concepts across different ImageNet3D classes**. For each class, the concept is *sampled randomly*, and the images associated with that concept are also *sampled randomly*, both with seed 42.

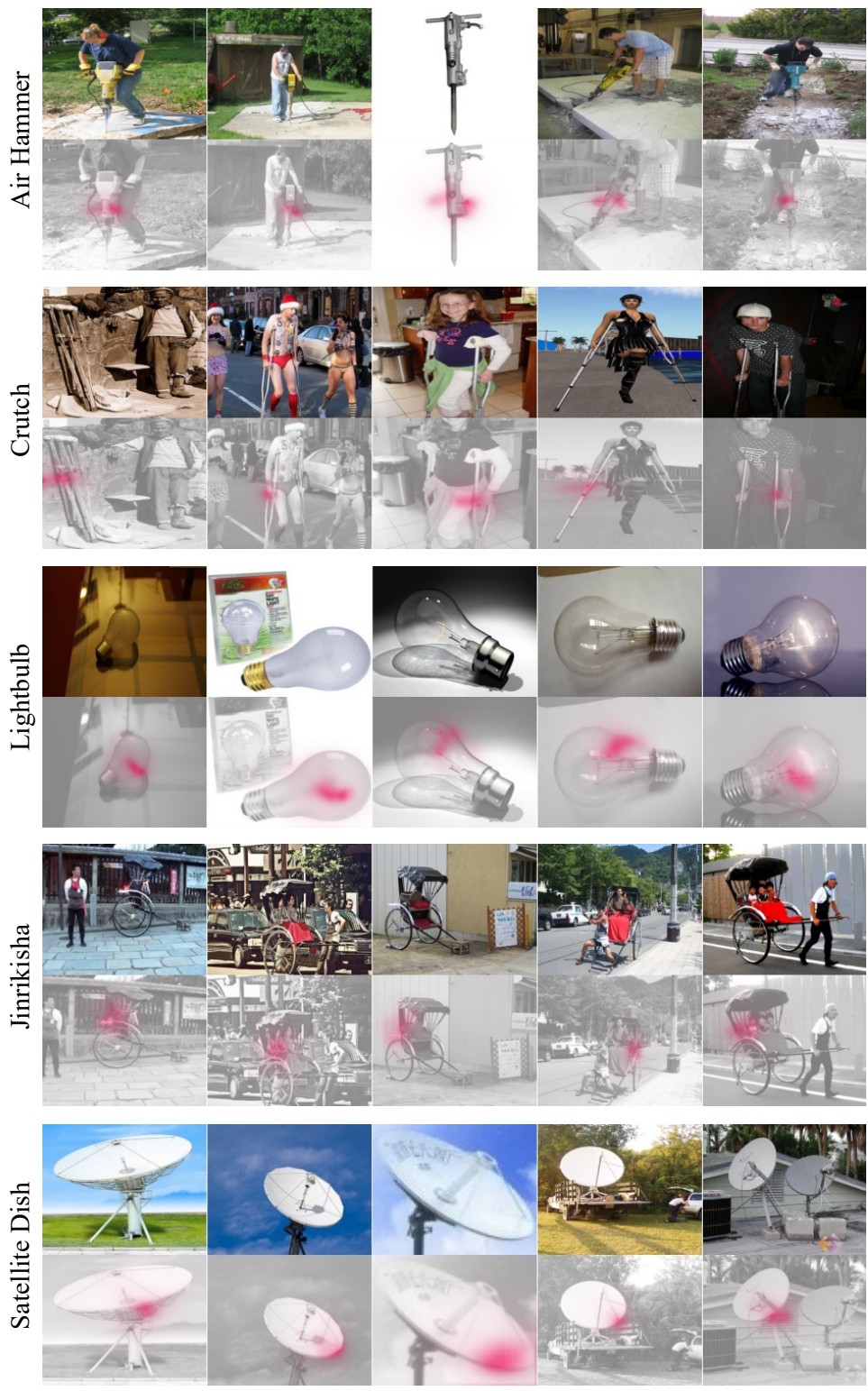

Figure L8: **CAVE (Ours) detects consistent concepts across different ImageNet3D classes**. For each class, the concept is *sampled randomly*, and the images associated with that concept are also *sampled randomly*, both with seed 42.

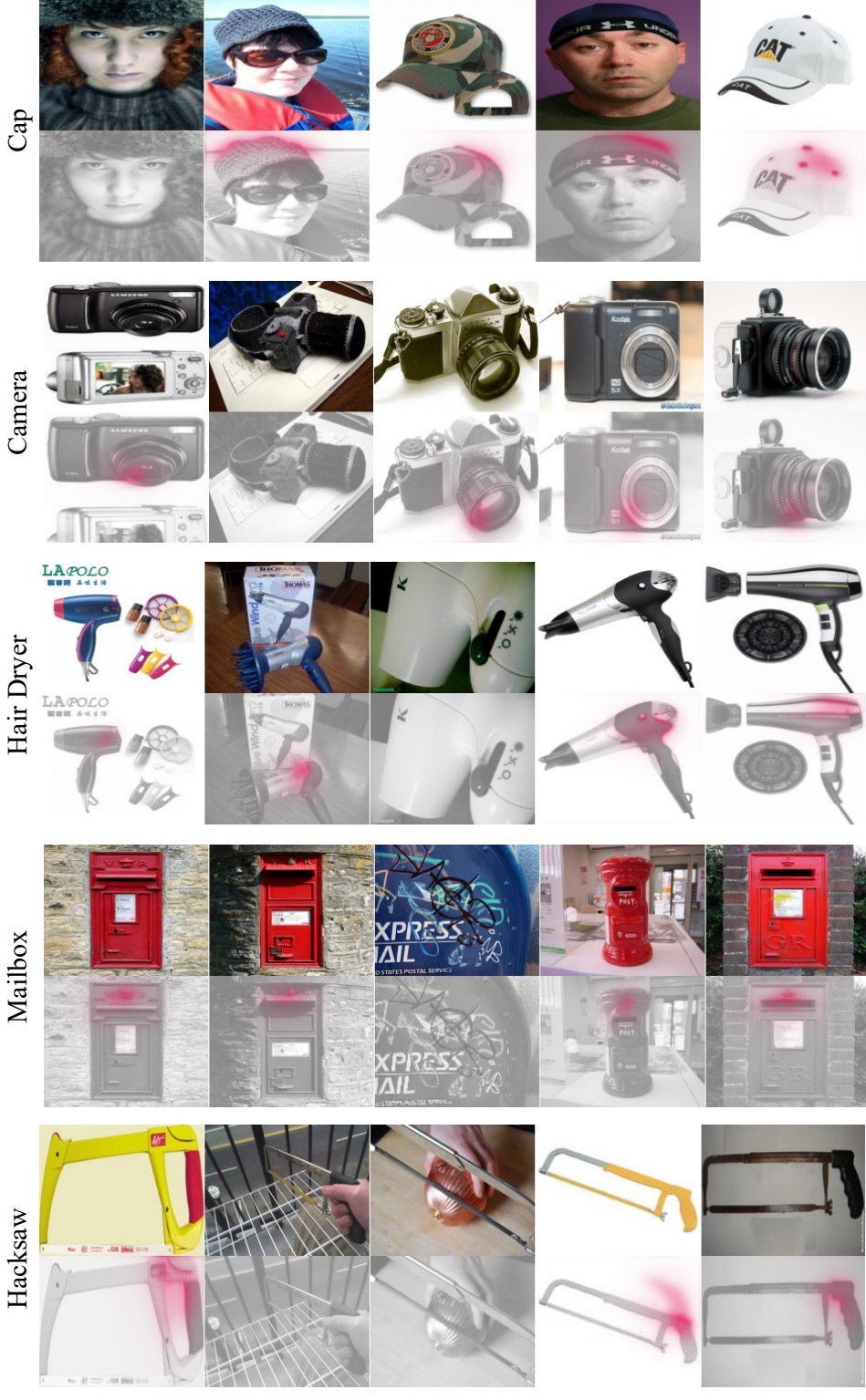

Figure L9: **CAVE (Ours) detects consistent concepts across different ImageNet3D classes and refrains from detection when the concepts are not visible in the image**. For each class, the concept is *sampled randomly*, and the images associated with that concept are also *sampled randomly*, both with seed 42.

## M  THE USE OF LARGE LANGUAGE MODELS

All scientific contributions, including the proposed method, experimental design, results, and analyses are solely our own work. We made minimal use of large language model, specifically ChatGPT, to only assist with the polishing of this paper, including:

  (i)  formatting tables and figures in latex (strictly for presentation), and

 (ii)  refining sentence structure for clarity and conciseness

