# OpenReview forum: "Interpretable 3D Neural Object Volumes for Robust Conceptual Reasoning"
_ICLR.cc/2026/Conference — ICLR 2026 Poster_

### Official Review · Reviewer_GACT · 2025-10-27

**Soundness:** 3
**Presentation:** 3
**Contribution:** 4
**Rating:** 8
**Confidence:** 3

**Summary:**

The paper is aim at finding a robust and explainable architecture for classification of images based on 3D representation of the object. The authors propose the CAVE method, which is aware of 3D representation of the object, and create an importance measure through it. The idea is to find the best correlation to a set of representations of given classes when generating the importance map. They also proposed LRP based method for faithfull attribution and 3D-C to evaluate stability.

**Strengths:**

* XAI is obviously one of the most challenging and important issues in nowadays AI era, in almost all fields, especially vision.
* The approach which the authors provide seems novel to me, and the idea to decompose each image to 3D concepts makes a lot of sense to me, and make it reasonable that the faithfulness of  the explanation will be better.
* The link between ellipsoid and Gaussianity, appear in almost all figures is super reasonable, and make things clearer and simpler. Evidence for the link of Gaussians and Ellipsoids, and how it is appear in CLIP you can find here [1].
* Impressive experimental section with convincing examples.


refs:

[1] Levi, Meir Yossef, and Guy Gilboa. "The Double-Ellipsoid Geometry of CLIP". ICML 25.

**Weaknesses:**

* One main concern is that this method is limited to rigid motion. Think of human body which can be transformed in a non-rigid motion then it is obvious that fixed set of dictionary representation cannot hold all variation of the body. So I find it as a limitation of the method.
* the 3D-C method is heavily relying on given set of meshes, where in the wild it is not given and may be hard to obtain especially for edge cases of objects (especially object with non-rigid movement abilities). I believe that the authors should elaborate on what are the options for a practitioner when this set is absent, or when he is interested in unknown set of classes.

Minor weaknesses:
* in the main figure on the first page, the left part is very intuitive and understandable, the right plot is much less. I didnt understand what the dashed lines stands for, moreover, it is better to put units or some explanation on the axes. I can get that the Y axis is accuracy in between 0-100 but what is 0.1 in spatial localization means?
* The citation of the CRAFT paper by Thomas Fel et. al is in German, replace it with the appropriate English one.

**Questions:**

* In line 51-85: "Notably, post-hoc methods... However such methods only approximate the model’s computations, and thus do not provide a faithful explanation". Seems to me like a very general and not accurate statement. Why did you claim that they do not provide faithful explanation? for the extreme case - a pure gradient from output to input of the entire architecture should ideally measure the impact of each input element on the output. Seems like very faithful explanation. I think that it should be clarify or backed somehow with experiments or references to other papers showing something like this.
* It is not clear in the preliminaries section whether the approach limited to embedders which produce feature vector for each pixel, since I think that most embedders squeeze the input information into much smaller dimension, ViT for example patchify the input. If it is the case, then it is a very profound limitation of your approach (or NOVUM you based on). If not - then I think that the notations of H and W are not clear.
* Im not sure how do the concepts being selected in space? why is bus contains the roof and car not (in Fig 4)?
* In many concept decomposition papers the sparsity is playing a vital role, the most basic example for this is the SAE (Sparse AutoEncoders). In several places in your paper you are trying to cinvince that sparsity is a good characteristic where it is almost trivial in concept decomposition.

Overall I find the ideas of the paper very intuitive, so the results seems reasonable to me. The idea to combine 3d-aware representation with XAI is very important, and the modeling through ellipsoids and Gaussians seems reasonable. I think that the paper is good and worth publishing.

---

> ### Author Response · Authors · 2025-11-20
> **Author response to Reviewer GACT (Part 1/2)**
>
> We thank Reviewer `GACT` for the very positive and constructive review of our paper. Below, we further address each point and describe the changes made in the revised manuscript.
>
> > **\[W1\]** One main concern is that this method is limited to rigid motion. Think of human body which can be transformed in a non-rigid motion then it is obvious that fixed set of dictionary representation cannot hold all variation of the body. So I find it as a limitation of the method.
>
> Indeed, our current formulation assumes a fixed canonical shape, which does not directly extend to highly non-rigid categories such as humans. This assumption is common across template-based correspondence methods [1]. At the same time, our approach is naturally compatible with articulated or deformable 3D models. Prior work [2] has already demonstrated that NOV-style representations can be extended to articulated templates (e.g., humans) by leveraging SMPL [3], enabling reasoning about pose on non-rigid bodies. This suggests that integrating deformable canonical models into our framework is not only feasible but a promising and realistic direction for future work. We are happy for your insightful question and added this discussion in section 7 to our revised manuscript.
>
> > **\[W2\]** 3D-C method is heavily relying on given set of meshes, where in the wild it is not given and may be hard to obtain especially for edge cases of objects (especially object with non-rigid movement abilities). I believe that the authors should elaborate on what are the options for a practitioner when this set is absent, or when he is interested in unknown set of classes.
>
> While 3D-C relies on a mesh when available, recent progress makes obtaining suitable meshes increasingly scalable in real-world use. Several directions exist for a practitioner: retrieving class-relevant meshes from large databases [4-8], with thousands to millions textured 3D objects across diverse categories, and mesh generation from images/texts [9-12] make it practical to generate high-quality mesh proxies even when CAD models are not provided. For non-rigid motions (e.g., humans), estimating 3D-C would require a deformable model (e.g., SMPL [3]) to estimate the instance pose, which is indeed an interesting future extension of our 3D-C metric. We happily added a brief discussion on this in section 7 of our revised manuscript.
>
> > **\[Minor W2\]** The citation of the CRAFT paper by Thomas Fel et. al is in German, replace it with the appropriate English one.
>
> Thank you for pointing this out. We apologise for the oversight. We have replaced the German citation of CRAFT with the correct English version in the revised manuscript.
>
>
> [1] Shtedritski, Aleksandar et al. "SHIC: Shape-Image Correspondences with no Keypoint Supervision". ECCV. 2024\.
> [2] Zhang, Yi et al. "3D-Aware Neural Body Fitting for Occlusion Robust 3D Human Pose Estimation" Proceedings of the IEEE/CVF International Conference on Computer Vision (ICCV). 2023\.
> [3] Loper, Matthew et al. "SMPL: : A Skinned Multi-Person Linear Model" ACM Trans. Graphics (Proc. SIGGRAPH Asia). 2015\.
> [4] Dong, Shaocong, et al. "From one to more: Contextual part latents for 3d generation." Proceedings of the IEEE/CVF International Conference on Computer Vision. 2025\.
> [5] Dong, Zhao, et al. "Digital twin catalog: A large-scale photorealistic 3d object digital twin dataset." Proceedings of the Computer Vision and Pattern Recognition Conference. 2025\.
> [6] Deitke, Matt, et al. "Objaverse-xl: A universe of 10m+ 3d objects." Advances in Neural Information Processing Systems 36 (2023): 35799-35813.
> [7] Deitke, Matt, et al. "Objaverse: A universe of annotated 3d objects." Proceedings of the IEEE/CVF conference on computer vision and pattern recognition. 2023
> [8] Chang, Angel X., et al. "Shapenet: An information-rich 3d model repository." arXiv preprint arXiv:1512.03012 (2015).
> [9] Chen, Sijin, et al. "Meshxl: Neural coordinate field for generative 3d foundation models." Advances in Neural Information Processing Systems 37 (2024): 97141-97166.
> [10] Siddiqui, Yawar, et al. "Meshgpt: Generating triangle meshes with decoder-only transformers." Proceedings of the IEEE/CVF conference on computer vision and pattern recognition. 2024\.
> [11] Yan, Xingguang, et al. "An Object is Worth 64× 64 Pixels: Generating 3D Object via Image Diffusion." 2025 International Conference on 3D Vision (3DV). IEEE, 2025\.
> [12] Chen, Yiwen, et al. "Meshanything v2: Artist-created mesh generation with adjacent mesh tokenization." Proceedings of the IEEE/CVF International Conference on Computer Vision. 2025\.

---

> ### Author Response · Authors · 2025-11-20
> **Author response to Reviewer GACT (Part 2/2)**
>
> > **\[Q1\]** In line 51-85: "Notably, post-hoc methods... However such methods only approximate the model’s computations, and thus do not provide a faithful explanation". Seems to me like a very general and not accurate statement. Why did you claim that they do not provide faithful explanation? for the extreme case \- a pure gradient from output to input of the entire architecture should ideally measure the impact of each input element on the output. Seems like very faithful explanation. I think that it should be clarify or backed somehow with experiments or references to other papers showing something like this.
>
> Thank you for pointing this out. We have clarified the text to distinguish between faithful-by-design (inherently interpretable) methods and post-hoc ones. Architectures such as CBMs and prototype networks provide faithful explanations because the prediction is **computed directly from interpretable units** (e.g., concepts/prototypes). In contrast, most post-hoc saliency or concept methods (including CRAFT, MCD, PCX, ICE) produce an **auxiliary explanation** from the intermediate activations that does not correspond to the model’s actual decision computation, i.e. it makes approximation errors which are left unexplained.
>
> Interestingly, the extreme case you mentioned is one that is highly discussed in the literature, with attributions from output to input being a **proxy** of how information is used for the classification differing significantly between approaches [1], and failing permutation-based sanity checks [2]. A recent line of work proposed a network architecture that produces faithful input-output attributions, i.e., the decision can be expressed directly through the units in the explanations [3].
>
> [1] S Rao et al. Better Understanding Differences in Attribution Methods via Systematic Evaluations. PAMI 2024\.
> [2] J. Adebayo et al. Sanity Checks for Saliency Maps.  NeurIPS 2018\.
> [3] M. Böhle et al. B-cos Networks: Alignment is All We Need for Interpretability. CVPR 2022\.
>
> > **\[Q2\]** It is not clear in the preliminaries section whether the approach limited to embedders which produce feature vector for each pixel, since I think that most embedders squeeze the input information into much smaller dimension, ViT for example patchify the input. If it is the case, then it is a very profound limitation of your approach (or NOVUM you based on). If not \- then I think that the notations of H and W are not clear.
>
> Sorry for the misunderstanding. Our approach is not limited to pixel-level embedders. We operate on latent feature maps $F_x \in \mathbb{R}^{H' \times W' \times C'}$ produced by the backbone, so the method only requires features at spatial locations $(H', W')$, not at the original pixel resolution $(H, W)$. The notation has been updated accordingly in lines 166-170.
>
> > **\[Q3\]** Im not sure how do the concepts being selected in space? why is bus contains the roof and car not (in Fig 4)?
>
> Thank you for the question\! In CAVE, concepts emerge by clustering the Gaussian features of the ellipsoid NOVs and these concepts correspond to spatial regions of the volume. Figure 4 is a sketch of CAVE to provide an  intuition of concept-annotated ellipsoidal neural volumes, with top-5 concepts for the shown image. The Car class does include a concept associated with its roof, but this concept is not among the top-5 activations for this particular input image and therefore does not appear in our illustration. Another point we want to highlight is that latent features from a car image can activate concepts belonging to another class with similar 3D geometry, such as Bus, which our Fig. 4 is intended to illustrate.
>
> > **\[Q4\]** Sparsity is a good characteristic and is almost trivial in concept decomposition.
>
> We agree that sparsity is a well-established and essential property in concept decomposition. Our intention was to make the connection explicit for readers who may be less familiar with this line of interpretability-related work. Emphasising sparsity helps clarify why a compact concept basis is important for interpretability within our 3D-aware setting.
>
>
> > **\[Minor W1\]** I didnt understand what the dashed lines stands for, moreover, it is better to put units or some explanation on the axes. [...] what is 0.1 in spatial localization means?\.
>
> The x-axis of Fig. 1 corresponds to Spatial Localisation (reported in Tab.1), which quantifies how well an explanation generated by a method overlaps with the ground-truth human-annotated object part in Pascal-Part. This metric is formally defined in Appendix Section G.2 and takes values in the range $[0, 1]$. A score of 0.1 means weak alignment with the human-annotated part, with higher values 0.25-0.30 indicating relatively stronger but still coarse localisation. We have now clarified this in Figure 1 caption.
>
> We would greatly appreciate clarification on which dashed line you are referring to, so that we can address it.

---

> > ### Comment · Reviewer_GACT · 2025-11-26
> >
> > My two main concerns where nicely addressed by the short discussion which you added. Moreover, the notations is clearer now after your revision. I still think that there are faithful post-hoc explanations and the claim was too decisive, but the added part made it more acceptable (at least by me). I liked the paper also in the initial iteration, and I still think it is good. Im happy to maintain my score. Best of luck.

---

> ### Author Response · Authors · 2025-11-26
>
> We sincerely thank you for your thoughtful feedback and for engaging with our responses! We appreciate your positive review of our work, and we’re very glad our responses addressed your concerns.

---

### Official Review · Reviewer_Kx36 · 2025-10-31

**Soundness:** 3
**Presentation:** 3
**Contribution:** 2
**Rating:** 6
**Confidence:** 2

**Summary:**

The author introduces a framework for interpretable and robust 3D-aware reasoning for image classification. The method is based on learning a sparse, interpretable dictionary of 3D object concepts and replaces dense 3D feature representations with these high level concepts. The approach is designed to enable model-faithful, localized, and consistent explanations for its predictions, even under OOD scenarios such as occlusion and context shifts. They use a recent method of Want et al (ICML 2025) for orientation estimation form 3D models. In addition, the paper proposes 3D Consistency (3D-C) metric that evaluates the spatial coherence of concepts using either ground-truth or estimated 3D meshes instead of human-defined parts. The paper presents an extensive experiment that benchmark their work against both post-hoc and inherently interpretable baselines, demonstrating advantages in robustness - interpretability trade offs, standard interpretability metrics, and classification accuracy on both in-distribution and OOD datasets.

**Strengths:**

1. Simplicity – instead of selecting dense 3D feature representations, it uses a sparse high level concepts instead.
2. Effectiveness - can operate both with and without ground-truth pose supervision leveraging Orient-Anything.
3. High accuracy – the authors demonstrates that CAVE achieves high accuracy and OOD robustness with far fewer parameters than dense NOV-based methods

**Weaknesses:**

1. 3D-C projects concept attributions onto a class CAD mesh. But if you do not have meshes for a class or the mesh is a poor proxy, you cannot score consistently well. This limits evaluation outside these datasets.
2. If the pose estimator struggles on certain data such as symmetric objects, CAVE accuracy and consistency will drop, making it a manageable weakness.
3. The method is limited to single objects scenes.
4. No reported stability across random seeds or concept count for K-Means.
5. No confidence intervals in the charts. Hard to judge the noise level.
6. Seems to be some missing refs and related work, such as:
a.	Xue, N., Tan, B., Xiao, Y. (2024): NEAT: Distilling 3D Wireframes from Neural Attraction Fields
b.	Liu, X., Zheng, C., Qian, M. (2024): Multi-View Attentive Contextualization for Multi-View 3D Object Detection
c.	Xue, N., Wu, T., Bai, S. (2023): Holistically-Attracted Wireframe Parsing
d.	Tan, B., Xue, N., Wu, T. (2023): NOPE-SAC: Neural One-Plane RANSAC for Sparse-View Planar 3D Reconstruction
e.	Xiao, Y., Xue, N., Wu, T. (2023): Level-S$^2$fM: Structure from Motion on Neural Level Set of Implicit Surfaces
f.	Grainger, R., Paniagua, T., Song, X. (2023): PaCa-ViT: Learning Patch-to-Cluster Attention in Vision Transformers
g.	Kashyap, P., Ravichandiran, P., Wang, L. (2023): Thermal Estimation for 3D-ICs Through Generative Networks
h.	Savadikar, C., Dai, M., Wu, T. (2023): Learning to Grow Artificial Hippocampi in Vision Transformers for Resilient Lifelong Learning

**Questions:**

1. Can the authors provide statistical variance of the results?
2. Can the authors present class-wise or attribute-wise breakdowns of accuracy/consistency, particularly in OOD or high-occlusion cases?
3. How can practitioners apply CAVE in unconstrained settings where 3D geometry is sparse or diverse?
4. Can the authors provide an accuracy and 3D Consistency as a function of pose error?

---

> ### Author Response · Authors · 2025-11-20
> **Author response to Reviewer Kx36  (Part 1/5)**
>
> We thank Reviewer `Kx36` for their thoughtful review of our paper. We address each of the raised concerns below.
>
> > **\[W1\]**. 3D-C projects concept attributions onto a class CAD mesh. But if you do not have meshes for a class or the mesh is a poor proxy, you cannot score consistently well. This limits evaluation outside these datasets.
>
> Thanks for pointing this out. Indeed, our 3D-C evaluation metric relies on the reference mesh with good fidelity to measure how consistently a concept localises to the same (3D) spatial region. In practice, there exist datasets with good annotation that can serve as benchmark data for this metric, some of which we have seen here [1-4]. Beyond these, we also anticipate that many more large scale mesh datasets get available over time, thanks to active advances in 3D reconstruction and generative modeling. Large-scale mesh datasets [5-8], with thousands to millions textured 3D objects across diverse categories, and recent works on mesh generation [9-12] make it practical to generate high-quality mesh proxies even when CAD models are not provided. We have added this discussion in section 7 of our revised manuscript.
>
> [1] Ma, Wufei, et al. "Imagenet3d: Towards general-purpose object-level 3d understanding." Advances in Neural Information Processing Systems 37\. 2024\.
> [2] Xiang, Yu, Roozbeh Mottaghi, and Silvio Savarese. "Beyond pascal: A benchmark for 3d object detection in the wild." IEEE winter conference on applications of computer vision\. IEEE, 2014\.
> [3] Zhao, Bingchen, et al. "Ood-cv: A benchmark for robustness to out-of-distribution shifts of individual nuisances in natural images." European conference on computer vision\. Cham: Springer Nature Switzerland, 2022\.
> [4] Wang, Angtian, et al. "Robust object detection under occlusion with context-aware compositionalnets." Proceedings of the IEEE/CVF conference on computer vision and pattern recognition\. 2020\.
> [5] Dong, Shaocong, et al. "From one to more: Contextual part latents for 3d generation." Proceedings of the IEEE/CVF International Conference on Computer Vision\. 2025\.
> [6] Dong, Zhao, et al. "Digital twin catalog: A large-scale photorealistic 3d object digital twin dataset." Proceedings of the Computer Vision and Pattern Recognition Conference\. 2025\.
> [7] Deitke, Matt, et al. "Objaverse-xl: A universe of 10m+ 3d objects." Advances in Neural Information Processing Systems 36 (2023): 35799-35813\.
> [8] Chang, Angel X., et al. "Shapenet: An information-rich 3d model repository." arXiv preprint arXiv:1512.03012 (2015)\.
> [9] Chen, Sijin, et al. "Meshxl: Neural coordinate field for generative 3d foundation models." Advances in Neural Information Processing Systems 37 (2024): 97141-97166\.
> [10] Siddiqui, Yawar, et al. "Meshgpt: Generating triangle meshes with decoder-only transformers." Proceedings of the IEEE/CVF conference on computer vision and pattern recognition\. 2024\.
> [11] Yan, Xingguang, et al. "An Object is Worth 64× 64 Pixels: Generating 3D Object via Image Diffusion." 2025 International Conference on 3D Vision (3DV)\. IEEE, 2025\.
> [12] Chen, Yiwen, et al. "Meshanything v2: Artist-created mesh generation with adjacent mesh tokenization." Proceedings of the IEEE/CVF International Conference on Computer Vision\. 2025\.

---

> ### Author Response · Authors · 2025-11-20
> **Author response to Reviewer Kx36  (Part 2/5)**
>
> > **\[W2, Q4\]**. If the pose estimator struggles on certain data such as symmetric objects, CAVE accuracy and consistency will drop, making it a manageable weakness.
>
> We appreciate reviewer `Kx36`’s intuition regarding symmetric objects; this is indeed a well-known source for pose ambiguity. Perhaps surprisingly, Orient Anything predictions on Pascal3D+ show that pose errors for several symmetric classes (e.g., sofa, TV monitor) are lower compared to less symmetric classes (e.g., aeroplane). The following Table (added as new Table A6 to the paper) provides mean pose error per class:
>
> **Table A6:** **Class-wise Mean Azimuth Pose Error (in degrees)** between Orient-Anything predictions and the ground-truth pose on Pascal3D+.
>
> | Metric | Aeroplane | Bicycle | Boat | Bottle | Bus | Car | Chair | Dining Table | Motorbike | Sofa | Train | TV Monitor |
> |--------|:-----------:|:---------:|:------:|:--------:|:-----:|:-----:|:--------:|:--------:|:-----------:|:------:|:-------|:-------------:|
> | **Azimuth error (°)** | 10.14 | 12.92 | 26.94 | 6.86 | 7.25 | 6.71 | 10.09 | 20.15 | 12.03 | 8.58 | 8.46 | 9.53 |
>
>
> We further added an analysis of accuracy and 3DC score with respect to pose error in the new **Fig. A4.** We find that
>
> * accuracy does not show a strong degradation for classes with pose ambiguity (e.g., dining table, boat with azimuth error \> 20 degree),
> * 3D-C shows a mild downward trend for larger azimuth errors, which is expected since 3D consistency directly reflects geometric alignment.
>
> We hypothesise that because CAVE is trained on Orient Anything’s estimated poses, it inherits any symmetric ambiguities in those estimates. At inference, a symmetric object (e.g., a left–right–symmetric boat) still activates the correct class concepts, even if they appear on the mirrored side. A concept trained to fire on the left may instead fire on the right, but this swap does not affect the final classification prediction. This "flip" hypothesis also aligns with the lower 3D-C scores.
>
>
> > **\[W3\]**. The method is limited to single object scenes.
>
> Thank you for highlighting this. Our work indeed focuses on object-centric classification. While our current experiments focus on single-object scenes, the approach is not inherently limited to this setting. Works in pose estimation in scenes follows an established approach of first *detecting a rough location of objects*, serving as a constrained to then fitting the object *to that local area* [1]. One can naturally extend CAVE to multi-object scenes following this established approach of first detecting object candidates and then applying our concept matching to each detected region. We see this as a promising and interesting direction for future work and are happily adding this to our discussion in section 7\.
> [1] Zhang, Jiyao et al. "Omni6DPose: A Benchmark and Model for Universal 6D Object Pose Estimation and Tracking". ECCV. 2024\.

---

> ### Author Response · Authors · 2025-11-20
> **Author response to Reviewer Kx36 (Part 3/5)**
>
> > **\[W4, W5, Q1\]**. No reported stability across random seeds or concept count for K-Means. No confidence intervals in the charts. Hard to judge the noise level.
>
> Thank you for the great suggestion. We have run additional repetitions of our experiments with varying seeds and included statistical variance across all key metrics of accuracy and interpretability in the updated draft. Specifically, we reported mean and standard deviation over **10 random seeds** for CAVE in new Tables 1, 2 and added confidence intervals for figure 7 in our revised manuscript. We provide these tables here for your convenience.
> **Table 1:** Concept interpretability evaluation.
>
> |                                | Faithful. | Spatial Localisation | Coverage
> |--------------------------------------|:-----------:|:------------:|:------------:
> | CAVE (Ours)                    | Yes   | 0.28 (&pm; 0.001) | 0.80 (&pm; 0.002)
> | CAVE (with full 3D supervision)| Yes   | 0.28 (&pm; 0.001)  | 0.87 (&pm; 0.002)
>
> | **3D Consistency (3D-C)** | Pascal3D+ | ImageNet3D | OccludedP3D+ | OOD-CV
> |--------------------------------------|:-----------:|:------------:|:------------:|:------------:
> | CAVE (Ours)                    |  0.40 (&pm; 0.001) | 0.40 (&pm; 0.001) | 0.23 (&pm; 0.006) | 0.24 (&pm; 0.002)
> | CAVE (with full 3D supervision)|  0.42 (&pm; 0.001) | 0.43 (&pm; 0.0003) | 0.23 (&pm; 0.010) | 0.26 (&pm; 0.001)
>
> **Table 2:** Accuracy on Pascal3D+ and ImageNet3D (in-distribution) and OccludedP3D+ and OOD-CV (out-of-distribution).
> | **Accuracy** |W/o Ground-truth 3D Pose| Pascal3D+ | ImageNet3D | OccludedP3D+ | OOD-CV
> |--------------------------------------|:-----------:|:-----------:|:------------:|:------------:|:------------:
> | CAVE (Ours)                    |**Yes**|  99.0 (&pm; 0.03) | 84.6 (&pm; 0.02) | 76.8 (&pm; 0.51) | 80.3 (&pm; 0.27)
> | CAVE (with full 3D supervision)|No | 99.4 (&pm; 0.02) | 88.5 (&pm; 0.03)  | 81.3 (&pm; 0.30)  | 84.0 (&pm; 0.21)
>
> We report concept-count ablation on in-distribution and OOD accuracy with confidence intervals in Fig.7. We additionally report the 3D-C scores across different numbers of concepts $D \in$ \{5, 10, 20, 40\} in new Table 3 (including the occlusion-wise breakdown). We provide the summary table below. We do not consider larger values of D, as increasing the number of learned concepts reduces sparsity, which is essential for maintaining interpretability in concept-based models [1], as also noted by reviewer `GACT`. We observe that our 3D-C scores improve with more concepts under heavy occlusion ([60-80%] occ.), but overall remain stable across concept counts in both in-distribution and OOD settings.
>
> **Table 3:** 3D Consistency (3D-C) of CAVE (w/o ground-truth pose) with different class concept counts.
> | Concept Count | Pascal3D+ | ImageNet3D | OccludedP3D+ | OOD-CV
> |--------------------------------------|:-----------:|:------------:|:------------:|:------------:
> | $D = 5$                    |  0.38 (&pm; 0.004) | 0.42 (&pm; 0.002) | 0.22 (&pm; 0.005) | 0.24 (&pm; 0.003)
> | $D = 10$                   |  0.39 (&pm; 0.003) | 0.41 (&pm; 0.001) | 0.23 (&pm; 0.007) | 0.24 (&pm; 0.001)
> | $D = 20$                   |  0.40 (&pm; 0.001) | 0.40 (&pm; 0.001) | 0.23 (&pm; 0.006) | 0.24 (&pm; 0.002)
> | $D = 40$                   |  0.41 (&pm; 0.004) | 0.40 (&pm; 0.001) | 0.24 (&pm; 0.003) | 0.23 (&pm; 0.001)
>
> We further report occlusion-wise breakdown in OccludedP3D+ of our CAVE's 3D-C with different class concept counts below.
> | Concept Count | [20-40%] occ. | [40-60%] occ. | [60-80%] occ.
> |--------------------------------------|:-----------:|:------------:|:------------:
> | $D = 5$                    |  0.27 (&pm; 0.002) | 0.22 (&pm; 0.01) | 0.17 (&pm; 0.01)
> | $D = 10$                   |  0.28 (&pm; 0.003) | 0.21 (&pm; 0.005) | 0.19 (&pm; 0.02)
> | $D = 20$                   |  0.29 (&pm; 0.002) | 0.21 (&pm; 0.002) | 0.20 (&pm; 0.02)
> | $D = 40$                   |  0.29 (&pm; 0.001) | 0.22 (&pm; 0.001) | 0.21 (&pm; 0.01)
>
> [1] Ramaswamy, Vikram V. et al. "Overlooked factors in concept-based explanations: Dataset choice, concept learnability, and human capability." Proceedings of the IEEE/CVF Conference on Computer Vision and Pattern Recognition. 2023\.
>
> > **\[W6\]**. Seems to be some missing refs and related work, such as: …
>
> We thank the reviewer for pointing out these related works. They indeed connect to the broader landscape of 3D geometry understanding, and we added the relevant suggested citations to the Related Work.

---

> ### Author Response · Authors · 2025-11-20
> **Author response to Reviewer Kx36 (Part 4/5)**
>
> > **\[Q2\]**. Can the authors present class-wise or attribute-wise breakdowns of accuracy/consistency, particularly in OOD or high-occlusion cases?
>
> ## Class-wise performance under different occlusion levels.
> We present these new results in Appendix A.1, table A1 for accuracy and A2 for 3D-C. We find that for individual classes, such as Boat, Bus or Train, the accuracy breaks down quicker than for others under increasing occlusion. Moreover, as expected, the 3D consistency is correlated with the trends in accuracy. For class-wise statistical variance across 10 random seeds, we report in Figures A1 and A2 of our revised manuscript.
>
> **Table A1:** **Class-wise Accuracy** on Pascal3D+ (in-distribution) and OccludedP3D+ (L1, L2, L3 occlusion).
> | Accuracy | Aeroplane | Bicycle | Boat | Bottle | Bus | Car | Chair | Dining Table | Motorbike | Sofa | Train | TV Monitor | **overall** |
> |-----------------|:-----------:|:---------:|:------:|:--------:|:-----:|:-----:|:--------:|:--------:|:-----------:|:------:|:-------:|:-------------:|:---------:|
> | L0 [0%] (in-distribution) | 99.67 | 99.86 | 99.26 | 99.85 | 98.48 | 99.72 | 91.78 | 99.53 | 98.79 | 99.57 | 95.90 | 100.00 | **99.0 (± 0.03)** |
> | L1 [20–40% occluded] | 96.78 | 98.20 | 92.41 | 98.06 | 86.32 | 97.99 | 85.58 | 96.48 | 94.45 | 98.32 | 78.58 | 99.55 | **94.8 (± 0.12)** |
> | L2 [40–60% occluded] | 83.51 | 90.44 | 73.30 | 93.88 | 62.38 | 87.63 | 71.49 | 85.08 | 84.02 | 94.89 | 57.10 | 91.03 | **82.8 (± 0.40)** |
> | L3 [60–80% occluded] | 40.60 | 62.60 | 41.05 | 81.35 | 21.44 | 52.81 | 44.81 | 60.24 | 52.41 | 83.71 | 32.35 | 57.41 | **52.7 (± 0.96)** |
>
> **Table A2:** **Class-wise 3D Consistency (3D-C)** on Pascal3D+ (in-distribution) and OccludedP3D+ (L1, L2, L3 occlusion). We denote --- for classes that do not have consistent concepts.
>
> | 3D-C | Aeroplane | Bicycle | Boat | Bottle | Bus | Car | Chair | Dining Table | Motorbike | Sofa | Train | TV | **overall** |
> |-----------------|:-----------:|:---------:|:------:|:--------:|:-----:|:-----:|:--------:|:--------:|:-----------:|:------:|:-------:|:-------------:|:---------:|
> | L0 [0%] (in-distribution)        | 0.220 | 0.235 | 0.174 | 0.694 | 0.343 | 0.299 | 0.430 | 0.456 | 0.248 | 0.561 | 0.385 | 0.799 | **0.404 (±0.0014)** |
> | L1 [20–40% occluded]     | 0.153 | 0.150 | 0.115 | 0.497 | 0.219 | 0.186 | 0.325 | 0.383 | 0.152 | 0.336 | 0.241 | 0.681 | **0.287 (±0.0018)** |
> | L2 [40–60% occluded]    | 0.105 | 0.104 | 0.085 | 0.380 | 0.149 | 0.117 | 0.255 | 0.288 | 0.094 | 0.204 | 0.169 | 0.581 | **0.211 (±0.0015)** |
> | L3 [60–80% occluded]     | ---   | 0.075 | ---   | 0.268 | ---   | 0.069 | 0.103 | 0.186 | 0.076 | 0.118 | ---   | 0.436 | **0.200 (±0.0166)** |

---

> ### Author Response · Authors · 2025-11-20
> **Author response to Reviewer Kx36 (Part 5/5)**
>
> ## OOD Attribute-wise performance.
> For OOD attributes, we find that *pose* is the most challenging attribute, followed by *weather*. We report these attribute-wise results in Tables A3, and the class-wise breakdown of each attribute in Tables A4 and A5, and class-wise statistical variance across 10 random seeds in Fig. A3 of our revised manuscript in Appendix A.2. We present the summary result here for your convenience.
>
> **Table A3:** OOD attribute-wise accuracy and 3D-C across 10 random seeds.
> | OOD Attribute | Context | Pose | Shape | Texture | Weather | **overall** |
> |--------------|:---------:|:------:|:-------:|:---------:|:---------:|:---------:|
> | **Accuracy** | 82.95 (± 0.20) | 72.77 (± 0.77) | 81.22 (± 0.65) | 83.27 (± 0.61) | 77.61 (± 0.39) | **80.3 (± 0.27)** |
> | **3D-C** | 0.236 (± 0.006) | 0.227 (± 0.018) | 0.239 (± 0.004) | 0.238 (± 0.004) | 0.234 (± 0.009) | **0.235 (± 0.002)** |
>
> Below, we further provide the class-wise accuracy and 3D-C scores for each OOD attribute to show how attribute sensitivity varies across object categories.
>
> **Table A4:** Class-wise accuracy per each OOD attribute (across 10 random seeds)
> | OOD Attribute | Aeroplane | Bicycle | Boat | Bus | Car | Chair | Dining Table | Motorbike | Sofa | Train | **overall** |
> |--------------|:-----------:|:---------:|:------:|:-----:|:-----:|:--------:|:--------:|:-----------:|:------:|:-------:|:---------:|
> | **Context** | 81.10 (± 1.06) | 88.92 (± 0.42) | 89.83 (± 1.34) | 66.40 (± 0.92) | 69.86 (± 0.91) | 74.05 (± 0.85) | 85.19 (± 0.67) | 84.66 (± 0.97) | 94.62 (± 0.43) | 75.81 (± 1.20) | **82.95 (± 0.20)** |
> | **Pose** | 87.59 (± 1.78) | 86.77 (± 1.38) | 79.39 (± 2.05) | 21.91 (± 1.75) | 65.56 (± 2.17) | 31.54 (± 2.43) | 33.33 (± 0.00) | 92.86 (± 0.00) | 75.00 (± 0.00) | 89.50 (± 3.69) | **72.77 (± 0.77)** |
> | **Shape** | 93.88 (± 0.57) | 90.89 (± 1.45) | 87.50 (± 1.96) | 90.00 (± 0.00) | 65.52 (± 2.30) | 71.17 (± 1.20) | 62.33 (± 2.59) | 93.49 (± 1.47) | 85.47 (± 0.44) | 63.33 (± 2.87) | **81.22 (± 0.65)** |
> | **Texture** | 97.43 (± 1.16) | 84.05 (± 1.00) | 89.19 (± 1.56) | 90.17 (± 0.91) | 90.62 (± 1.77) | 58.35 (± 0.89) | 83.68 (± 1.32) | 91.11 (± 0.65) | 88.62 (± 1.01) | 69.50 (± 4.08) | **83.27 (± 0.61)** |
> | **Weather** | 84.69 (± 0.85) | 92.63 (± 1.22) | 81.83 (± 1.62) | 55.82 (± 1.52) | 54.74 (± 1.25) | 87.14 (± 2.02) | 30.00 (± 0.00) | 91.28 (± 0.64) | 75.00 (± 0.00) | 84.31 (± 1.46) | **77.61 (± 0.39)** |
>
> **Table A5:** Class-wise 3D-C per each OOD attribute. For statistical variance across 10 random seeds, refer to Fig. A3.
> | OOD Attribute | Aeroplane | Bicycle | Boat | Bus | Car | Chair | Dining Table | Motorbike | Sofa | Train | **overall** |
> |--------------|:-----------:|:---------:|:------:|:-----:|:-----:|:--------:|:--------:|:-----------:|:------:|:-------:|:---------:|
> | **Context** | 0.152 | 0.159 | 0.129 | 0.235 | 0.164 | 0.239 | 0.394 | 0.175 | 0.401 | 0.309 | **0.236 (± 0.006)** |
> | **Pose** | 0.155 | 0.164 | 0.127 | 0.247 | 0.169 | 0.212 | 0.375 | 0.169 | 0.338 | 0.307 | **0.227 (± 0.018)** |
> | **Shape** | 0.157 | 0.162 | 0.131 | 0.246 | 0.148 | 0.242 | 0.392 | 0.176 | 0.422 | 0.310 | **0.239 (± 0.004)** |
> | **Texture** | 0.150 | 0.173 | 0.136 | 0.235 | 0.163 | 0.235 | 0.393 | 0.174 | 0.416 | 0.300 | **0.238 (± 0.004)** |
> | **Weather** | 0.153 | 0.156 | 0.131 | 0.228 | 0.156 | 0.229 | 0.411 | 0.176 | 0.387 | 0.305 | **0.234 (± 0.009)** |
>
>
>
> >**\[Q3\]**. How can practitioners apply CAVE in unconstrained settings where 3D geometry is sparse or diverse?
>
> Thank you for the question! CAVE can be applied in more unconstrained settings even when 3D geometry is sparse (e.g., bicycles) or highly diverse (e.g., aeroplanes). Our current experiments use a single canonical neural volume per class (ellipsoid), which works well for rigid and moderately varying classes. For categories with greater geometric diversity, we expect the performance to drop slightly because a single canonical geometry cannot fully capture intra-class shape variation.
> In practice, there are several natural extensions to CAVE for such settings:
>
> 1. Learning multiple neural volumes per class for different variants, allowing CAVE to select the best-fitting geometry for each image; or
> 2. Using a deformable template, where a canonical neural volume can smoothly adapt to variations (e.g., via learned deformations [1]).
>    We find these directions interesting and promising for future work.
>
> [1] Sommer, Leonhard et al. "Common3D: Self-Supervised Learning of 3D Morphable Models for Common Objects in Neural Feature Space". CVPR. 2025\.
>
> ---------
>
> We thank Reviewer `Kx36` once again for the thoughtful and constructive review, which helped us significantly improve the manuscript. **Should any further questions arise, we would be happy to address them.** We kindly ask Reviewer `Kx36` to consider a higher score in light of our detailed responses and revisions.

---

### Official Review · Reviewer_zP4N · 2025-10-31

**Soundness:** 2
**Presentation:** 3
**Contribution:** 2
**Rating:** 4
**Confidence:** 3

**Summary:**

This work introduces CAVE, a robust, interpretable image classifier based on 3-D object representations. To visualize pixel importance, the authors also extend LRP for the proposed model. Finally, a new metric called 3D consistency is introduced to measure whether learned concepts consistently match with the same part of the underlying 3D volume. Experiments show that CAVE produces better localization, coverage, and consistency than prior methods while achieving strong accuracy. Notably, in the absence of 3D annotations CAVE outperforms all other methods.

**Strengths:**

- The problem targeted by this paper is interesting and well motivated, while remaining reasonable in scope.
- The figures throughout the work are clear and well made.
- CAVE is, to my knowledge, the first model to combine neural object volumes with concept based explanations, providing a source of novelty.
- The numerical results achieved by CAVE are strong compared to other robust only or interpretable only methods.

**Weaknesses:**

- It is unclear to me why CAVE can be considered inherently interpretable/faithful. Each classification is a function of the dot product between each image patch and some concept; this dot product is not bounded, and does not reflect a meaningful distance/similarity measure. Even if it did, these concepts do not seem to have any semantic meaning -- they are freely learned parameters. As such, it may be inappropriate to compare with only interpretable/explainable models. If I am missing something fundamental, clarification would be appreciated. This is my primary concern.
- The clarity of this work suffers a bit from not being self-contained. In particular, it is unclear from this work how spatial location and distributional nature of NOV's actually matter. If a reader does not read NOVUM, each NOV simply seems like a free parameter vector in the embedding space of the backbone. Moreover, there are a couple minor issues with formalism that impact clarity:
    - Line 314: $\sum_{(i,j)} A^+(x, h) = 1$ <- i, j do not appear as indices inside the sum
    - The function $\Omega_y$ is never defined, including in the appendix

**Questions:**

- How can it be said that standalone NOVUM is not faithful, but CAVE is? The only difference between the two as predictive models seems to be the number of concepts.
- In the appendix, Spatial localisation is defined with respect to a single image-concept pair. How is this aggregated to a single value per model/dataset?
- It may be worth comparing to [1]; this is an extension of ProtoPNet that, while not targeting robustness, learns distributional concepts in 2D space.

[1] Wang, Chong, et al. "Mixture of gaussian-distributed prototypes with generative modelling for interpretable and trustworthy image recognition." IEEE Transactions on Pattern Analysis and Machine Intelligence (2025).

---

> ### Author Response · Authors · 2025-11-20
> **Author response to Reviewer zP4N (Part 1/3)**
>
> We thank Reviewer `zP4N` for their constructive feedback on our work. Below, we address each of the concerns raised.
> > **\[W1\]** It is unclear to me why CAVE can be considered inherently interpretable/faithful. Each classification is a function of the dot product between each image patch and some concept; this dot product is not bounded, and does not reflect a meaningful distance/similarity measure. Even if it did, these concepts do not seem to have any semantic meaning \-- they are freely learned parameters. As such, it may be inappropriate to compare with only interpretable/explainable models. If I am missing something fundamental, clarification would be appreciated. This is my primary concern.
>
> We apologise for the misunderstanding. We would like to clarify (1) why CAVE is inherently interpretable, (2) why the dot product in our concept matching is a bounded measure for latent image feature and volumetric concept alignment, (3) why concepts are often semantic-aligned in practice despite learned unsupervised, and (4) why our comparison to interpretable models is justified.
> ### (1) CAVE is inherently interpretable
> CAVE is inherently interpretable because its predictions are **computed entirely and exactly from activations of interpretable concepts**, following the same design principle as concept-based and prototype-based models [1-5]. These volumetric concepts are learned on pre-trained NOVs and then used as the sole units for classification, effectively re-parameterising ellipsoidal NOVUM so that its entire decision process operates only on a sparse set of concept units instead of thousands of opaque Gaussian features. This yields a **new, inherently interpretable architecture**. We revised this in lines 283–288.
>
> ### (2) The dot product is a measure for latent feature–concept alignment, and is bounded in $[-1, 1]$.
> In CAVE, the dot product is computed between L2-normalised latent feature vectors from the backbone and L2-normalised concept vectors (i.e., cluster centroids of Gaussian features). This effectively is a **cosine similarity** which is bounded in $[−1, 1]$. Cosine similarity is widely used in concept-based and prototype models [1-4], as well as in the additional suggested baseline [5]. We have updated the manuscript to further clarify this point in line 281-282\.
>
> ### (3) Semantic meaningfulness of concepts learned unsupervised.
> Indeed, our concepts are learned unsupervised (no concept labels), which is a desired feature to avoid costly and scarce annotation, in line with [1-5]. While none of these works can ensure that discovered concepts are semantically aligned, it has been observed in the literature that such approaches, including clustering, NMF, or autoencoder-based, yield highly interpretable features [6]. We also find that in practice our volumetric concepts show strong semantic alignment to human annotated parts (see e.g. Table 1 (Pascal-Part), Fig. 2, Fig. 6, and Appendix K). In our revised manuscript, we additionally report that these concepts are stable across random seeds (see Table 1), further supporting their semantic reliability.
>
> ### (4) Our comparison to interpretable models is justified.
> CAVE and all considered interpretable baselines LF-CBM, ProtoPNet, TesNet, PIP-Net, (new baseline) MGProto are inherently interpretable classifiers and learn concepts or prototypes without concept supervision, hence a comparison comes very natural.
>
> Combining (1), (2), (3), and (4), we hope this clarifies Reviewer `zP4N`’s concern.
>
> [1] Nauta, Meike et al. "Pip-net: Patch-based intuitive prototypes for interpretable image classification." Proceedings of the IEEE/CVF Conference on Computer Vision and Pattern Recognition. 2023\.
> [2] Wang, Jiaqi et al. "Interpretable image recognition by constructing transparent embedding space." Proceedings of the IEEE/CVF international conference on computer vision. 2021\.
> [3] Chen, Chaofan et al. "This looks like that: deep learning for interpretable image recognition." Advances in neural information processing systems 32\. 2019\.
> [4] Oikarinen, Tuomas et al. "Label-free Concept Bottleneck Models, International Conference on Learning Representations."  2023\.
> [5] Wang, Chong et al. "Mixture of gaussian-distributed prototypes with generative modelling for interpretable and trustworthy image recognition." IEEE Transactions on Pattern Analysis and Machine Intelligence (2025).
> [6] Fel, Thomas, et al. "A holistic approach to unifying automatic concept extraction and concept importance estimation." Advances in Neural Information Processing Systems 36 (2023): 54805-54818.

---

> ### Author Response · Authors · 2025-11-20
> **Author response to Reviewer zP4N (Part 2/3)**
>
> > **\[W2-a\]** The clarity of this work suffers a bit from not being self-contained. In particular, it is unclear from this work how spatial location and distributional nature of NOV actually matter. If a reader does not read NOVUM, each NOV simply seems like a free parameter vector in the embedding space of the backbone.
>
> We apologise for the missing details about NOVUM. NOVs are not free parameter vectors in feature space, but each NOV Gaussian is anchored to a fixed location on a canonical object surface. NOVUM contrastively trains these Gaussians to align with image features through a pose-consistency constraint. This means NOVUM learns how 2D latent features project onto a consistent 3D structure across views. The spatial arrangement and distribution of Gaussians therefore capture the object’s geometry. We have clarified this in the revised manuscript in lines 176, 182-184 and provided further clarification in Appendix E to aid the understanding of NOVUM.
>
> > **\[W2-b\]** Formalism issues in section 5 that impact clarity.
>
> We thank Reviewer `zP4N` for raising this point, which allowed us to further refine the clarity of our notation and definitions.
>
> **Regarding indices inside the sum $\sum\_{(i, j)} A^{+}(x, h) = 1$**:
>
> We denote by $(i, j)$ a pixel position in $A^{+}(x, h) \in \mathbb{R}^{H\times W}$ and by $A^{+}(x_{ij}, h)\in\mathbb{R}^{+}$ the positive attribution given to this pixel. We have updated the notation to $\sum_{(i,j)\in \\\{1,\ldots,H\\\}\times \\\{1,\ldots,W\\\}} A^{+}(x_{ij}, h) = 1$ (lines 362–363 of our revised manuscript).
>
> **Regarding the attribution-to-mesh mapping**, we now formally define the mapping $\mathcal{\Omega}\_{y}: \mathbb{R}^{H\times W} \rightarrow \mathbb{R}^{|\mathcal{Q}\_y|}$ where $\mathcal{Q}\_y$ denotes the set of triangles in the CAD model of class $y$ that represents the object’s surface. Given a concept $h$ and an input image $x$ of class $y$, $\Omega_{y}$ maps the positive attribution map $A^{+}(x, h) \in \mathbb{R}^{H\times W}$ of this concept onto the corresponding oriented CAD model and aggregates for each triangle $q \in \mathcal{Q}\_{y}$ the sum of all projected attributions falling onto it.  Concretely, we define the mapping component-wise as follows:
>
> $\\Omega_y(A^{+}(x, h))\_q := \\sum\_{(i,j) \in \\mathcal{P}^{(q)}\_y}  A^{+}(x\_{ij}, h),$
>
> where $\\mathcal{P}^{(q)}\_y$ is the set of pixel positions $(i,j) \in \\{1,\ldots,H\\}\times \\{1,\ldots,W\\}$ whose projection falls onto triangle $q \in \\mathcal{Q}\_y$ of the CAD model of class $y$. This is Eq. (3), lines 357-359 of our revised manuscript. Collecting all the components, we obtain the output vector of this mapping as
>
> $\\Omega\_y(A^{+}(x, h)) = (\\Omega_y(A^{+}(x, h))_q)\_{q \in \\mathcal{Q}\_y} \in \\mathbb{R}^{|\\mathcal{Q}\_y|}$
>
> > **\[Q1\]** How can it be said that standalone NOVUM is not faithful, but CAVE is? The only difference between the two as predictive models seems to be the number of concepts.
>
> We thank Reviewer `zP4N` for raising this point. We agree that NOVUM is (of course) mathematically consistent with its own computation. In concept-based interpretability, **an explanation is considered model-faithful if it is expressed in terms of the interpretable units (e.g., a small set of prototypes or a concept bottleneck) the model itself uses for classification** [1-6]. As such, model faithfulness is not applicable to opaque models. We realise that the red $\textcolor{red}{✘}$ for NOVUM in Table 1 may have caused confusion by unintentionally suggesting "not faithful." We apologise for this and have replaced it with "–" in our revision to clearly indicate *not applicable*.
>
> [1] Rudin, Cynthia. "Stop explaining black box machine learning models for high stakes decisions and use interpretable models instead". Nature machine intelligence, 1(5):206–215, 2019\.
> [2] Wang, Bor-Shiun et al. "Mcpnet: An interpretable classifier via multi-level concept prototypes". In Proceedings of the IEEE/CVF Conference on Computer Vision and Pattern Recognition, pp. 10885–10894, 2024\.
> [3] Koh, Pang Wei et al. "Concept bottleneck models." International conference on machine learning. PMLR, 2020\.
> [4] Oikarinen, Tuomas, et al. "Label-free Concept Bottleneck Models, International Conference on Learning Representations."  2023.
> [5] Böhle, Moritz et al. "B-cos networks: Alignment is all we need for interpretability." Proceedings of the IEEE/CVF Conference on Computer Vision and Pattern Recognition. 2022\.
> [6] Chen, Chaofan et al. "This looks like that: deep learning for interpretable image recognition." Advances in neural information processing systems 32\. 2019\.

---

> ### Author Response · Authors · 2025-11-20
> **Author response to Reviewer zP4N (Part 3/3)**
>
> > **\[Q2\]** Spatial localisation is defined with respect to a single image-concept pair. How is this aggregated to a single value per model/dataset?
>
> Our goal is to evaluate the spatial localisation of concepts w.r.t human-annotated object parts, so **for each model**, the metric is first computed for each image–concept pair and then aggregated across images, concepts, and classes as follows:
>
> - **Per-concept score**: For each image $x$ and each concept $h$ of the same class, we compute **per image–concept pair score** $SL_h(x) = \underset{k}{\max} SL_{h,k}(x)$, i.e., the ground-truth part $b_k$ in $x$ with which the concept has the highest alignment, and $SL_{h,k}(x)$ as defined in our Appendix Equation (F.1), line 1517-1519. Then **per-concept** score for concept $h$ is computed as the average across test images in which the concept $h$ is active.
> - **Per-class score**: for each class, we average across concept localisation scores across all concepts for that class.
> - **Dataset-level score** on Pascal-Part: finally, we report the mean of these per-class scores as the overall spatial localisation score (which we report in Table 1).
>   We added this clarification in lines 1524 \- 1532\.
>
> > **\[Q3\]** Comparison with MGProto.
>
> Thank you for this suggestion. MGProto [1] is indeed a relevant and strong baseline that we were not aware of before. Following your recommendation, we have added MGProto in our evaluation. MGProto performs competitively with existing baselines across several metrics, including spatial localisation and OOD accuracy. CAVE outperforms MGProto across metrics and datasets. The updated results are shown in Tables 1 and 2 in the revised manuscript and also report here below for your convenience. For all metrics, higher is better, with results for CAVE reported with statistical variance across 10 random seeds.
>
> **Table 1:** Concept interpretability evaluation of MGProto [1] vs. CAVE (Ours)
> |                                | Faithful. | Spatial Localisation | Coverage
> |--------------------------------------|:-----------:|------------|------------
> | MGProto                          | **Yes**   | 0.25   | 0.35
> | CAVE (Ours)                    | **Yes**   | **0.28** (&pm; 0.001) | **0.80** (&pm; 0.002)
>
> | **3D Consistency (3D-C)** | Pascal3D+ | ImageNet3D | OccludedP3D+ | OOD-CV
> |--------------------------------------|-----------|------------|------------|------------
> | MGProto                          | 0.19 | 0.16 | 0.16 | 0.07
> | CAVE (Ours)                    |  **0.40** (&pm; 0.001) | **0.40** (&pm; 0.001) | **0.23** (&pm; 0.006) | **0.24** (&pm; 0.002)
>
> **Table 2:** Accuracy of MGProto [1] vs. CAVE (ours) on Pascal3D+ and ImageNet3D (in-distribution) and OccludedP3D+ and OOD-CV (out-of-distribution).
> | **Accuracy** |W/o Ground-truth 3D Pose| Pascal3D+ | ImageNet3D | OccludedP3D+ | OOD-CV
> |--------------------------------------|:-----------:|-----------|------------|------------|------------
> | MGProto                         |**Yes** | 97.2 | 64.2 | 73.8 | 72.3
> | CAVE (Ours)                    |**Yes**|  **99.0** (&pm; 0.03) | **84.6** (&pm; 0.02) | **76.8** (&pm; 0.51) | **80.3** (&pm; 0.27)
>
>
> [1] Wang, Chong et al. "Mixture of gaussian-distributed prototypes with generative modelling for interpretable and trustworthy image recognition." IEEE Transactions on Pattern Analysis and Machine Intelligence (2025).
>
> -----------
> We sincerely thank Reviewer `zP4N` once again for their constructive feedback. Your comments have helped us substantially improve the clarity of our manuscript. We have addressed each concern in detail and revised the paper accordingly.  **If there are any additional concerns, we would be happy to address them. We hope that our clarifications, additional discussions, and new results sufficiently resolve the issues and convey the contributions of our work more clearly.** We kindly ask reviewer zP4N to consider a higher rating in light of the revisions and responses provided.

---

> > ### Comment · Reviewer_zP4N · 2025-11-21
> >
> > I'd like to start by thanking the authors for their thorough response. I am satisfied with their answers to my second and and third questions, and believe the clarity has been improved by addressing my second weakness.
> >
> > I still find the claim that "CAVE is inherently interpretable because its predictions are computed entirely and exactly from activations of interpretable concepts" unconvincing, largely because I see it as inconsistent with the discussion of NOVUM. The key contention here is to justify that these concepts are indeed interpretable.
> >
> > The concepts in ProtoPNet-style methods ([1-3, 5], from the authors' first response) can be considered interpretable/meaningful because the network is trained to form well separated latent clusters (see the clst and sep loss of [3], which are used in most followup work). They are additionally projected onto the nearest input image in most ProtoPNet work, which yields a pixel space representation of each prototype. These steps are not applied in CAVE.
> >
> > The concepts in concept-bottleneck-style methods ([4, 6], from the authors' first response) can be considered interpretable because they are explicitly trained to encode some semantic concept, independent of the classification objective. This yields a natural set of visualizations for each concept -- the concept set. This does not apply to the NOVs of CAVE.
> >
> > It is true that the predictions in CAVE are formed using a cosine similarity to a set of NOVs (thank you for the clarification on this point). However, something similar can be said of the final convolutional layer in any black box CNN: predictions are formed using the dot product between a set of learned convolutional filters and an input. It is generally accepted that convolutional filters are not interpretable. Simply normalizing the feature representation and convolutional filter would not change that.
> >
> > As such, I remain unconvinced that the NOV's in CAVE are a "human-interpretable set of concept units" while those in NOVUM are "opaque Gaussian features". The concept units themselves are identical, unless I'm mistaken. It's their sparsity that is changed, not the nature of the NOVs.
> >
> > All of this said, I still find the numerical results compelling. I would simply like to see the claims around faithfulness/inherent interpretability adjusted for the reasons described above.

---

> ### Author Response · Authors · 2025-11-25
> **Author response to Reviewer zP4N (Part 1/2)**
>
> We sincerely thank Reviewer zP4N for staying so engaged in the discussion. Your follow-up clarification helped us understand the central issue much more clearly which we address as follows, along the lines of [Q1], [W1]. We now clearly (1) distinguish CAVE's implicit interpretability from the explicit interpretability of prototype-based models and CBMs, (2) recognise both NOVUM and CAVE are faithful, with CAVE adding implicit (emergent) interpretability through sparse, region-level concepts (constrained complexity).
>
> > The concepts in ProtoPNet-style methods can be considered interpretable/meaningful because the network is trained to form well separated latent clusters, additionally projected onto the nearest input image. The concepts in concept-bottleneck-style methods can be considered interpretable because they are explicitly trained to encode some semantic concept. These are not applied in CAVE.
>
> **[W1]** We agree with Reviewer zP4N that the nature of concepts in CAVE differs from those in, for example, prototype-based networks and CBMs. Prototype-based approaches (e.g., ProtoPNet and its variants) provide explicit visual interpretability through a dedicated prototype layer and prototype projection, while CBMs provide explicit semantic interpretability by supervising concepts with semantic labels. To make this distinction clear, we refer to these models collectively as **explicitly interpretable**, in contrast to CAVE and recent concept-based methods [2–4] that are **implicitly interpretable** (unsupervised concept learning, with emergent semantics).
>
> To contextualise CAVE’s interpretability within the literature, we draw on Molnar’s taxonomy of intrinsic (inherent) interpretability. Molnar highlights that models can be inherently interpretable even when they *“mix both interpretability by design and post-hoc interpretability”*, with his examples being models whose structure makes their computation transparent and complexity appropriately constrained, even if individual components require post-hoc visualisation to be inspected [1]. CAVE thus fits under the same paradigm of inherent interpretability: its prediction is transparent by design, decomposing exactly over a small dictionary of region-level concepts derived from Gaussian features, while our adapted LRP is used for post-hoc concept visualisation.
>
> While these approaches differ in whether interpretability is explicit (prototype-based, CBMs) or implicit (CAVE), all of them are model-faithful and share the scope of interpreting predictions. We are grateful to the reviewer for highlighting this important conceptual distinction. **We refined our framing accordingly in lines 143–155, and 281–284, and added a small discussion paragraph (481-485) in Section 7.**
>
> > Something similar can be said of the final convolutional layer in any black box CNN: predictions are formed using the dot product between a set of learned convolutional filters and an input. It is generally accepted that convolutional filters are not interpretable. Simply normalizing the feature representation and convolutional filter would not change that.
>
> We agree that individual filters are typically not interpretable due to polysemanticity and superposition, but backbone features **spanning all channels/filters** (i.e., represent activation patterns among populations of filter responses) encode semantics [5–7]. CAVE does not attempt to interpret single filters; instead, its concept vectors are cluster centroids of Gaussian features trained to align with these backbone features. Therefore, CAVE is not only geometry-grounded but also operates at the same representational level as concept-based methods, such as CRAFT [2], ICE [3], and MCD [4] which recover meaningful concept directions unsupervised and are regarded as interpretable. We clarify in our revision that this interpretability is, however, implicit.

---

> ### Author Response · Authors · 2025-11-25
> **Author response to Reviewer zP4N (Part 2/2)**
>
> > As such, I remain unconvinced that the NOV's in CAVE are a "human-interpretable set of concept units" while those in NOVUM are "opaque Gaussian features". The concept units themselves are identical, unless I'm mistaken. It's their sparsity that is changed, not the nature of the NOVs.
>
> **[Q1]** We thank the reviewer for this crucial clarification. We apologise for the ambiguity between faithfulness with interpretability in our original framing. Both NOVUM and CAVE are faithful-by-design, as their predictions decompose exactly over their respective decision units.
>
> Both models share the underlying NOV feature space, but differ in how the decision units are defined. NOVUM treats each individual Gaussian feature as its own decision unit, resulting in thousands of fine-grained, highly localised vectors. Importantly, CAVE **does not simply reuse these units**; it replaces them with newly learned cluster centroids, which act as distinct, coarse region-level decision units. These centroids represent **entire regions** of NOVUM’s Gaussian features. At inference, backbone features are mapped to this much smaller, sparser dictionary, creating a coarse-grained many-to-one correspondence. We emphasise that a small number of centroids is sufficient to summarise NOVUM’s densely distributed Gaussian features without sacrificing accuracy or robustness (Tab. 2). This shift from dense, local Gaussians to region-level representative concepts yields implicit interpretability. We find that CAVE’s centroids, i.e., our concepts, empirically align with semantically meaningful object parts (Tab. 1, Fig. 6, App. L). Hence, the key change is not only in sparsity but also in the *nature/granularity* of the units (coarse region-level centroids vs. fine-grained Gaussians).
>
> Nevertheless, neither set of units is explicitly human-interpretable, as the reviewer correctly noted. We revised the manuscript to remove wording that could imply NOVUM is unfaithful and to clarify in lines 280–284 and 420–423 that CAVE preserves NOVUM’s faithfulness while adding **implicit** interpretability through clustering Gaussian features into sparse, region-level concepts.
>
>
> [1]  Molnar, C. (2025). Interpretable Machine Learning: A Guide for Making Black Box Models Explainable (3rd ed.). christophm.github.io/interpretable-ml-book/, Chapter 4.
> [2] Fel, Thomas, et al. "Craft: Concept recursive activation factorization for explainability." Proceedings of the IEEE/CVF Conference on Computer Vision and Pattern Recognition. 2023.
> [3] Zhang, Ruihan, et al. "Invertible concept-based explanations for cnn models with non-negative concept activation vectors." Proceedings of the AAAI Conference on Artificial Intelligence. Vol. 35. No. 13. 2021.
> [4] Vielhaben, Johanna, Stefan Bluecher, and Nils Strodthoff. "Multi-dimensional concept discovery (MCD): A unifying framework with completeness guarantees." Transactions on Machine Learning Research. 2023.
> [5] Scherlis, Adam, et al. "Polysemanticity and capacity in neural networks." 2022.
> [6] Bau, David, et al. "Network dissection: Quantifying interpretability of deep visual representations." Proceedings of the IEEE conference on computer vision and pattern recognition. 2017.
> [7] Olah, et al., "Zoom In: An Introduction to Circuits", Distill, 2020.
>
> --------------------------------------------------------------
> We have also added a concise summary of the key points from this discussion in Appendix B. We thank Reviewer zP4N once again for this thoughtful and constructive exchange. Your feedback has helped us clarify several conceptual nuances that are often blurred in the interpretability literature. **We hope that our revisions clarify these points, and we welcome further questions or feedback**.

---

> > ### Comment · Reviewer_zP4N · 2025-11-25
> >
> > Thank you for the continued productive discussion on this point. The clarification is helpful, and the additional notes positioning CAVE in the literature are appropriate.
> >
> > Given that both NOVUM and CAVE are faithful-by-design, I would recommend removing the "Faithful" column from Table 1. However, this is a fairly minor nitpick at this point.
> >
> > I greatly appreciate the authors' thoughtful responses, and I am happy to raise my score to recommend acceptance at this point.

---

> > > ### Author Response · Authors · 2025-11-25
> > >
> > > Thank you for the constructive feedback and the positive assessment. We appreciate the suggestion regarding Table 1 and have updated the table accordingly. We are grateful for your thoughtful consideration of our work and are glad to hear that the clarifications were helpful.

---

### Author Response · Authors · 2025-11-20
**Author response summary**

We sincerely thank all reviewers for their thoughtful and constructive feedback, their positive remarks, and their clear appreciation of our contributions.
We thank Reviewers `GACT` and `zP4N` for recognising the **novelty** of our method, noting that ***"CAVE is the first model to combine neural object volumes with concept-based explanations"*** (`zP4N`). Both reviewers also emphasised the importance and motivation of the problem, describing it as ***"interesting and well motivated"*** (`zP4N`) and ***"one of the most challenging and important issues in nowadays AI era"*** (`GACT`).
Reviewer `zP4N` highlighted our **figure clarity and quality**, and that ***"the numerical results achieved by CAVE are strong compared to other robust only or interpretable only methods"***.
Reviewer `Kx36` emphasised the ***"simplicity"*** and ***"effectiveness"*** of our approach while reaching ***"high accuracy and OOD robustness"***.
Finally, Reviewer `GACT` remarked that our **experimental results as** ***"impressive with convincing examples"***, and that our consistent ***"link between ellipsoids and Gaussianity"*** throughout our figures is ***"super reasonable"***, making the method ***"clearer"***.
We greatly value all these positive assessments.

During the rebuttal, we made substantial improvements inspired by all reviewers’ constructive feedback. In particular, we

1) Added **MGProto as an additional baseline**. In updated Tables 1 and 2, we show that CAVE also outperforms this new method.
2) Added **statistical variance to validate the stability** of our results. We now report mean (&pm; std) across 10 random seeds for CAVE with and without 3D ground-truth poses.
3)  Added a **concept-count ablation for 3D consistency** (new Table 3), **class-wise analysis across occlusion levels** (new Appendix A1, Tables A1 and A2, Figures A1 and A2), **OOD-attribute-wise analysis** (new Appendix A.2, Tables A3, A4, and A5, Figure A3) and **pose error analysis** (new Appendix A3, Table A6 and Figure A4).
4) **Clarified inherent interpretability and concept matching in CAVE** (new Appendix B): We added an explicit explanation of why CAVE is inherently interpretable, clarified that concept matching uses a bounded cosine similarity, and improved the description of NOV-based concept discovery.
5) **Strengthened discussion and related work**. We expanded the discussion on multi-object scenes, mesh availability, and non-rigid categories, and integrated related work suggested by the reviewers.

Our updates are highlighted in purple in the revised pdf. We thank the reviewers again for their constructive feedback, which directly contributes to a stronger paper. We look forward to addressing any further questions during the discussion phase.

---

### Author Response · Authors · 2025-12-02

We thank ACs and all reviewers for their time, especially in this exceptional circumstance. We are grateful for the opportunity to summarise our rebuttal and subsequent discussions with reviewers. In addition to our previous overview of manuscript revisions, we include below a brief context summary of points highlighted by reviewers, followed by summaries of each reviewer exchange and the current points of agreement. We do our best to remain neutral and strictly factual. Our goal is to accurately reflect the reviewers’ concerns and our responses without re-arguing the case, overstating consensus, or introducing additional interpretations.

------------------------
### Consistent strengths across reviews

**Novelty and Conceptual Contribution**: the proposed CAVE is described as “novel” (`GACT`, `zP4N`), “first model to combine neural object volumes with concept-based explanations” (`zP4N`). Reviewers describe our research problem as “interesting and well motivated” (`zP4N`) and our idea as “intuitive” (`GACT`), noting our consistent link between ellipsoid and Gaussianity is “super reasonable” (`GACT`).

**Empirical Strength and Comprehensive Evaluation**: reviewers agree that CAVE achieves “high accuracy and OOD robustness” with “extensive experiments” (`Kx36`), “impressive experimental section with convincing examples” (`GACT`) and that its numerical results is “strong compared to other robust-only or interpretable-only methods” (`zP4N`).

**Simplicity, Effectiveness and Clarity**: CAVE is described as “simple” and “effective”, and “can operate both with and without ground-truth pose supervision” (`Kx36`), and “in the absence of 3D annotations CAVE outperforms all other methods” (`zP4N`). Reviewers further praised the clarity and quality of our figures and visual explanations (`zP4N`, `GACT`).

------------------------
### Reviewer positions after discussion
**Reviewer `GACT` (initial rating: 8, later maintained score)**:
- Reviewer `GACT` asked us to clarify key assumptions (rigid motion, mesh dependence, per-pixel features), and to justify our claim about post-hoc faithfulness, while overall finding our paper idea intuitive and our paper worthy of publication.
- We addressed these by explaining compatibility with deformable models (e.g. SMPL [1]), outlining practical mesh-retrieval/generation options, clarifying that per-pixel features are not required, and adding references to support our discussion of post-hoc methods being unfaithful to model computation. In their final comment, reviewer `GACT` noted that these points were “nicely addressed”, found our revised faithfulness claim “more acceptable”, and stated that they were “happy to maintain [their] score”.

**Reviewer `zP4N` (initial rating: 4, later recommended acceptance)**:
- Reviewer `zP4N` asked us to clarify our claims about CAVE’s inherent interpretability and faithfulness (relative to NOVUM), and requested clearer exposition of NOVs, formal definitions, metric aggregation, and a baseline comparison to MGProto [2].
- We clarified CAVE’s implicit, emergent interpretability and situated its inherent interpretability within Molnar’s taxonomy [3], refined our framing of faithfulness, expanded the NOV explanation and formalism, and added MGProto as a baseline. In their final comment, reviewer `zP4N` found the clarification “helpful”, the positioning of CAVE’s inherent interpretability “appropriate”, and was “happy to raise [their] score to recommend acceptance”.

**Reviewer `Kx36` (initial rating: 6, no further discussion)**:
- Reviewer `Kx36` requested additional stability/variance analyses, pose-error analysis, class-wise and OOD attribute-wise breakdowns, and practical guidance for more unconstrained 3D settings, and cases when CAD meshes are not immediately available.
- We added full variance analyses (across 10 random seeds), stability across concept counts with occlusion-wise breakdown, detailed pose-error, class-wise, and OOD attribute-wise analyses, clarified how CAVE can be applied when 3D geometry is sparse or diverse (e.g., using multiple neural volumes per class or deformable templates), and outlined practical mesh-retrieval/generation options when they are not immediately available; we were waiting for the reviewer’s response to these additional results.

We sincerely appreciate the reviewers’ constructive and thoughtful feedback in the discussion phase, which has meaningfully improved the clarity and quality of our paper. We also thank ACs in advance for their time and attention in reviewing our submission in this cycle.

[1] Loper et al., SMPL, SIGGRAPH Asia 2015.
[2] Wang et al., MGProto, TPAMI 2025.
[3] Molnar, Interpretable Machine Learning, 3rd ed., 2025.

---

### Meta-Review · Area_Chair_Mq24 · 2026-01-06

**Summary:**

Reviewers primarily questioned the validity of the authors' claims regarding "inherent interpretability" and "faithfulness," specifically challenging whether the learned volumetric concepts—derived from unsupervised clustering of Gaussian features—could be considered semantically meaningful or comparable to explicit prototypes (zP4N). Technical concerns focused on the method's limitation to rigid objects and the dependency of the 3D Consistency (3D-C) metric on available CAD meshes, which might limit applicability in unconstrained or non-rigid settings (GACT, Kx36). Additionally, reviewers requested more rigorous empirical validation, including comparisons to strong baselines like MGProto, statistical variance reporting to ensure stability, and granular breakdowns of performance across different occlusion levels and OOD attributes (Kx36, zP4N).

**Reviewer Concerns:**

The authors successfully addressed the majority of these concerns through extensive revisions and active discussion. The theoretical debate on interpretability was resolved by distinguishing CAVE's "implicit" interpretability from explicit prototype methods, leading Reviewer zP4N to recommend acceptance, while the addition of the MGProto baseline and statistical variance tables directly satisfied specific requests for empirical rigor. While the limitation regarding non-rigid objects (e.g., humans) remains inherent to the current formulation, the authors' explanation of compatibility with deformable templates like SMPL satisfied Reviewer GACT. The concerns raised by Reviewer Kx36 regarding stability and granular performance were met with new appendix data (Tables 1-3, A1-A6), though this reviewer did not explicitly confirm these additions resolved their hesitation.

**Reviewer Scores:**

Reviewer zP4N has already actively participated and raised their score to an Accept, and Reviewer GACT most likely will maintain their score of 8. Reviewer Kx36, who initially gave a "marginally above threshold" score of 6, would likely have raised their score had they reviewed the new data; their primary critique was the lack of stability analysis and granular breakdowns, both of which the authors provided comprehensively in the revision (showing low variance and detailed OOD performance). Consequently, the paper likely commands a consensus for acceptance among all experts.

---

### Decision · Program_Chairs · 2026-01-26

Accept (Poster)